# ICLASS 1.1, a variational Inverse modelling framework for the Chemistry Land-surface Atmosphere Soil Slab model: description, validation and application

Peter J.M. Bosman[1] and Maarten C. Krol[1,2]

[1]Meteorology and Air Quality Group, Wageningen University, Wageningen, the Netherlands
[2]Institute for Marine and Atmospheric Research, Utrecht University, Utrecht, the Netherlands

**Correspondence:** Peter Bosman (peter.bosman.publicaddress@gmail.com)

**Abstract.** This paper provides a description of ICLASS 1.1: a variational Inverse modelling framework for the Chemistry Land-surface Atmosphere Soil Slab model. This framework can be used to study the atmospheric boundary layer, surface layer or the exchange of gases, moisture, heat and momentum between the land surface and the lower atmosphere. The general aim of the framework is to allow to assimilate various streams of observations (fluxes, mixing ratios at multiple heights, ...) to estimate model parameters, thereby obtaining a physical model that is consistent with a diverse set of observations. The framework allows to retrieve parameters in an objective manner, and enables the estimation of information that is difficult to obtain directly by observations, for example free-tropospheric mixing ratios or stomatal conductances. Furthermore it allows to estimate possible biases in observations. Modelling the carbon cycle at ecosystem level is one of the main intended fields of application. The physical model around which the framework is constructed is relatively simple, yet contains the core physics to simulate the essentials of a well-mixed boundary layer and of land–atmosphere exchange. The model includes an explicit description of the atmospheric surface layer, a region where scalars show relatively large gradients with height. An important challenge is the strong non-linearity of the model, which complicates the estimation of best parameter values. The constructed adjoint of the tangent linear model can be used to mitigate this challenge. The adjoint allows for an analytical gradient of the objective cost function, used for minimisation of this function. An implemented Monte-Carlo way of running ICLASS can further help to handle non-linearity, and provides posterior statistics on the estimated parameters. The paper provides a technical description of the framework, includes a validation of the adjoint code, as well as tests for the full inverse modelling framework and a successful example application for a grassland in the Netherlands.

## 1 Introduction

Exchanges of heat, mass and momentum between the land surface and the atmosphere play an essential role for weather, climate, air quality and biogeochemical cycles. Surface heating under sunny daytime conditions usually leads to the growth of a relatively well-mixed layer close to the land surface, the convective boundary layer (CBL). This layer is directly impacted by exchange processes with the land surface and is also a layer where humans live in. Modelling the composition and thermodynamic state of the CBL in its interaction with the land surface is the target of the Chemistry Land-surface Atmosphere Soil

Slab model (CLASS; Vilà-Guerau De Arellano et al., 2015). This and similar models have been applied frequently, e.g. for understanding the daily cycle of evapotranspiration (van Heerwaarden et al., 2010), studying the effects of aerosols on boundary layer dynamics (Barbaro et al., 2014), studying the effects of elevated $CO_2$ on boundary layer clouds (Vilà-Guerau De Arellano et al., 2012) or for studying the ammonia budget (Schulte et al., 2021). Next to a representation of the CBL, the CLASS model includes a simple representation for the exchange of gases, heat, moisture and momentum between the land surface and the lower atmosphere. The model explicitly accounts for the surface layer, which is, under sunny daytime conditions, a layer within the CBL close to the surface with relatively strong vertical gradients of scalars (e.g. specific humidity and temperature) and momentum (Stull, 1988). The best model performance is during the convective daytime period. Since the CBL-model physics are relatively simple and only include the essential boundary layer processes, the model performs best on what might be called "golden days". Those are days in which advection is either absent or uniform in time and space, deep convection and precipitation are absent, and sufficient incoming shortwave radiation heats the surface allowing for the formation of a prototypical convective boundary layer. When these assumptions are met, the evolution of the budgets of heat, moisture, and gases is to a large extent determined by local land–atmosphere interactions. The aforementioned assumptions should ideally be valid for the whole modelled period. They should ideally hold on a spatial scale large enough that violations of the assumptions in the region do not influence the model simulation location. In practice, days are often not "ideal", e.g. a time-varying advection can be present. This does not necessarily mean the model cannot be applied to that day, but, performance is likely to be worse.

To further our understanding of land–atmosphere exchange, tall tower observational sites have been established, for instance at Cabauw, the Netherlands (Bosveld et al., 2020; Vermeulen et al., 2011); Hyytiälä, Finland (Vesala et al., 2005); and Harvard Forest, USA (Commane et al., 2015). These observational sites provide time series of different types of measurements (observation streams). Even so, many studies only use a (small) fraction of the different streams of observations available for a specific day and location (e.g. Vilà-Guerau De Arellano et al., 2012). A model like CLASS, containing both a mixed-layer and land-surface part, can be used to fit an extensive set of observation streams simultaneously. When model results are consistent with a diverse set of measurements, this gives more confidence that the internal physics are robust and the model has been adequately parameterised to reliably simulate reality. However, an important difficulty in the application of a model like CLASS concerns parameter tuning to obtain a good fit to observations. Some parameters can be obtained quite directly from observations (for instance initial mixed-layer humidity), but, for example, estimating free-tropospheric lapse rates or certain land-surface parameters is often more challenging. When many parameters need to be determined, the feasible parameter space becomes vast. If this vast parameter space is not properly explored, the obtained parameters can be subjective and sub-optimal. The estimation of parameters is further complicated by possible overfitting and the problem of parameter equifinality (Tang and Zhuang, 2008), the latter especially in case not enough types of observations are used. Next to that, some of the available ecosystem/CBL-level observations may suffer from biases. An example is the closure of the surface energy balance, where the available energy is often larger than the sum of the latent and sensible turbulent heat fluxes (Foken, 2008). This energy balance closure problem is a known issue with eddy-covariance observations (Foken, 2008; Oncley et al., 2007; Renner et al., 2019), and various explanations have been suggested (Foken, 2008).

The above text illustrates the need for an objective optimisation framework, capable of correcting observations for biases. We therefore present here a description of ICLASS, an inverse modelling framework built around the CLASS model, including a bias-correction scheme for specific bias patterns. This framework can estimate model parameters, by minimising an objective cost function using a variational (Chevallier et al., 2005) framework. ICLASS uses a Bayesian approach, in the sense that it combines information, both from observations and from prior knowledge about the parameters, to come to a solution with a reduced uncertainty in the optimised parameters. A major strength of this framework is that it allows to incorporate several streams of observations, for instance, chemical fluxes, mixing ratios, temperatures at multiple heights, and radiosonde observations of the boundary-layer height. By optimising a number of predefined key parameters of the model, we aim to obtain a diurnal simulation that is consistent with a diverse set of measurements. Additionally, error statistics that are estimated provide information about the constraints the measurements place on the model parameters. Modelling the carbon cycle at ecosystem level is one of the main intended fields of application. As an example, with some extensions to the framework, ICLASS could be applied to ecosystem observations of the coupled exchange of $CO_2$ and carbonyl sulfide (a tracer for obtaining stomatal conductance, Whelan et al., 2018).

Besides ICLASS, there have been extensive earlier efforts in literature to estimate parameters in land-atmosphere exchange models. For example, Bastrikov et al. (2018) and Mäkelä et al. (2019) optimised parameters of the land-surface models OR-CHIDEE and JSBACH respectively. These models simulate additional processes not included in the CLASS model, which enables to calculate additional land-surface variables. For example, in contrast to ORCHIDEE, CLASS cannot simulate leaf phenology or the allocation of carbon to different biomass pools. However, a distinct advantage of the CLASS model is the coupling of a land-surface model to a mixed-layer model. This facilitates the inclusion of atmospheric observations such as mixing ratios in the optimisation of land-atmosphere exchange parameters. Next to that, simple models like CLASS have the advantage of requiring less computation time, and the output might be more easily understood. Kaminski et al. (2012) and Schürmann et al. (2016) also assimilate both land-surface-related and atmosphere-related observations. In those studies a land-surface model is coupled to an atmospheric transport model. Meteorology is not simulated in those studies. In ICLASS, meteorology adds an additional set of observation streams, that can be used to optimise land-surface-related parameters that are linked both to gas fluxes and meteorology.

An important challenge for the optimisation framework is the strong non-linearity of the model. As an example, the change in mixed-layer specific humidity ($q$) with time is a function of $q$ itself: a stronger evapotranspiration flux leads to an increased specific humidity in the mixed layer, which in turn reduces the evapotranspiration flux again (van Heerwaarden et al., 2009). The non-linearity causes numerically calculated cost function gradients to deviate from the true analytical gradients, since the cost function can vary irregularly with a changing model parameter value. This is hampering proper minimisation of the cost function when using numerically calculated gradients. An *adjoint* has been used in the past to optimise parameters, e.g. for land-surface models (Raoult et al., 2016; Ziehn et al., 2012). Constructing the adjoint of the tangent linear model is a way to obtain more accurate gradient calculations, as the adjoint provides a locally exact analytical gradient of the cost function at the locations where the function is differentiable. This approach furthermore allows to efficiently retrieve the sensitivity of model output to model parameters. Also, using an analytical gradient is generally computationally less expensive compared to using

a numerical gradient (Doicu et al., 2010, p17). Margulis and Entekhabi (2001a) constructed an adjoint model framework of a coupled land-surface boundary-layer model, which they used to study differences in daytime sensitivity of surface turbulent fluxes for the same model in coupled and uncoupled modes (Margulis and Entekhabi, 2001b). However, their CBL model did not include carbon dioxide nor does it allow to model scalars at specific heights in the surface layer. We expect these to be important for our framework that aims to make optimal use of several information streams. For doing gradient calculations within the ICLASS framework, we constructed the adjoint of CLASS.

The paper is structured as follows: First we give some information on the (slightly adapted) forward model CLASS (Sect. 2). The inverse modelling framework built around CLASS is described in Sect. 3. Information on how error statistics are employed and produced follows in Sect. 4, after which we provide a description of the model output (Sect. 5) and technical details of the code (Sect. 6). Afterwards we present the adjoint and gradient tests that serve as validation for the constructed adjoint model (Sect. 7). Further information about the adjoint model is available in the supplementary material. In Sect. 8 we perform observation system simulation experiments that validate the full inverse modelling framework. In the last section before the concluding discussion we present an example application, for a grassland site in the Netherlands (Cabauw), where a very comprehensive meteorological dataset is complemented with detailed measurements of $CO_2$ mixing ratios and surface fluxes.

## 2 Forward model

The employed forward model in our inverse modelling framework is the (slightly adapted) Chemistry Land-surface Atmosphere Soil Slab model (CLASS; Vilà-Guerau De Arellano et al., 2015). The model code is freely available on GitHub (https://classmodel.github.io/). We made use of the Python version of CLASS to construct our inverse modelling framework. We will shortly describe the essentials of the model which are relevant for the inverse modelling framework.

The model consists of several parts, namely the mixed layer, the surface layer, and the land surface, including the soil (Fig. 1). It is a conceptual model that uses a relatively small set of differential equations (Wouters et al., 2019). The core of the model is a box-model representation of an atmospheric mixed layer. Therefore an essential assumption of the model is that during daytime turbulence is strong enough to maintain well-mixed conditions in this layer (Ouwersloot et al., 2012). The mixed-layer tendency equation for any scalar (e.g. $CO_2$, heat) is:

$$\text{tendency} = \frac{(\text{surface flux} + \text{entrainment flux})}{\text{mixed layer height}} + \text{advection} \tag{1}$$

The surface flux is the exchange flux with the land surface (including vegetation and soil). The entrainment flux is the exchange flux between the mixed layer and the overlying free troposphere. For moisture and chemical species, a cloud mass flux can also be included in the equation. The mixed layer height is dynamic during the day and evolves under the driving force of the surface heat fluxes and large scale subsidence. Cloud effects on the boundary-layer height and growth due to mechanical turbulence can also be accounted for.

Above the mixed layer a discontinuity occurs in the scalar quantities, representing an infinitely small inversion layer. Above the inversion, the scalars are assumed to follow a linear profile with height in the free troposphere (Fig. 1). The entrainment

fluxes are calculated as follows: First, the buoyancy entrainment flux is taken as a fixed fraction of the surface flux of this quantity (Stull, 1988, p 478), to which entrainment driven by shear can optionally be added. From this virtual heat entrainment flux, an entrainment velocity is calculated. The entrainment flux for a specific scalar (e.g. $CO_2$) is then obtained by multiplying the entrainment velocity with the value of the (inversion-layer) discontinuity for the respective scalar.

The surface layer is defined in the model as the lowest 10% of the boundary layer. In this (optional) layer, Monin-Obukhov similarity theory (Monin and Obukhov, 1954; Stull, 1988) is employed. In the original CLASS surface layer, scalars such as temperature are evaluated at $2\,m$ height. For some scalars, we have extended this to multiple user-specified heights. This allows to compare observations of chemical mixing ratios and temperatures at different heights (e.g. along a tower) to model output. Since the steepness of vertical profiles depends on wind speed and roughness of the surface, these gradients reflect information
about these quantities.

    The (optional) land surface includes a simple soil representation as well as an a-gs module. This a-gs module (Jacobs, 1994; Ronda et al., 2001) is a big-leaf method (Friend, 2001) for calculating the exchange of $CO_2$ between atmosphere and biosphere, and the stomatal resistance. The latter is used for calculating $H_2O$ exchange. As an alternative for a-gs, a Jarvis-Stewart approach (Jarvis, 1976; Stewart, 1988) can also be used in the calculation of $H_2O$ exchange. The latter approach is
more simple, herein, stomatal conductance consists of a maximum conductance multiplied with a set of factors between 0 and 1 (Jacobs, 1994). In CLASS, there are 4 factors included, which represent limitations due to the amount of incoming light, temperature, vapour pressure deficit and soil moisture. The land-surface part is responsible for calculating the exchange fluxes of sensible heat, latent heat and $CO_2$ between the mixed layer and the land surface. The model has a module for calculating long- and shortwave radiation dynamically. In this module, shortwave radiation is calculated using the date and time, cloud
cover and albedo. For longwave radiation, surface temperature and the temperature at the top of the surface layer are used. The resulting net radiation is used implicitly in the calculation of the heat fluxes, thereby obtaining a closed energy balance in the model (simplified calculation). Soil temperature and moisture are also simulated, based on a force-restore model. The soil heat flux to the atmosphere is calculated based on the gradient between soil and surface temperature, the latter is obtained from a simplified energy balance calculation.
More details on the equations in the model can be found in Vilà-Guerau De Arellano et al. (2015). Note that some relatively small changes with respect to the original CLASS model have been implemented, as documented in the ICLASS manual, which is part of the material that can be downloaded via the Zenodo link in the "Code and data availability" section.

## 3   Inverse modelling framework

### 3.1   General

Inverse modelling is based on using observations and, ideally, prior information to statistically optimise a set of variables driving a physical system (Brasseur and Jacob, 2017). The $n$ variables to be optimised are contained in a state vector $x$. In our framework, this vector can be subdivided in two vectors. Those are $x_m$, containing state variables belonging to the input of CLASS (e.g. $CO_2$ advection, albedo,..), and $x_b$, containing state variables belonging to our bias-correction scheme (see Sect.

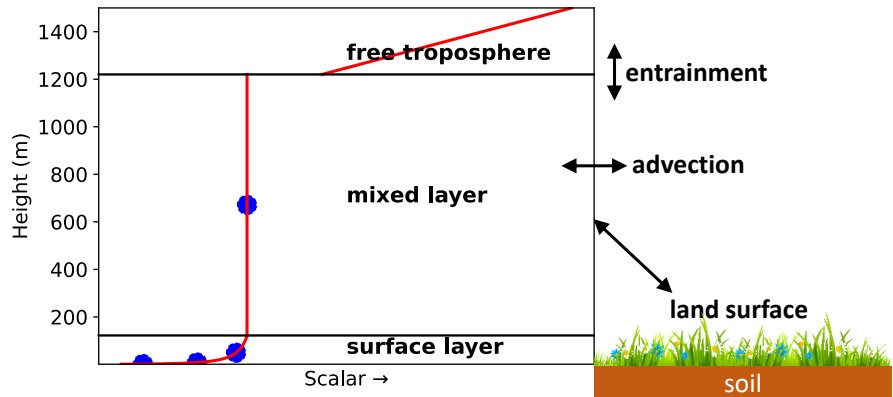

**Figure 1.** Sketch of the employed forward model: the (slightly adapted) CLASS model. The blue dots in the surface layer represent user-specified heights where the model calculates scalars, e.g. the $CO_2$ mixing ratio. The mixed layer is represented by a single bulk value in the model (blue dot in mixed layer). At the top of the mixed layer, a discontinuity (jump) occurs in the profiles. The free troposphere is not explicitly modelled, but is taken into account for the exchange with the mixed layer (entrainment). The slope of the free troposphere line is the free troposheric lapse rate. A constant advection can be taken into account as a source/sink.

3.2). The forward-model $H$ (CLASS) projects vector $\boldsymbol{x_m}$ to provide model output that can be compared to observations, e.g.
temperatures at different heights (full list in ICLASS manual). This output does not only depend on $\boldsymbol{x_m}$, but also on model parameters that are not part of the state. Those are contained in a vector $\boldsymbol{p}$. The result is contained in vector $\boldsymbol{H}(\boldsymbol{x_m}, \boldsymbol{p})$. Note that an overview of all inverse-modelling variables defined in this section is given in Table B1, including dimensions and units. We initially define a cost function $J\,(-)$ as (Brasseur and Jacob, 2017):

$$J(\boldsymbol{x}) = (\boldsymbol{x} - \boldsymbol{x_A})^T \mathbf{S_A}^{-1}(\boldsymbol{x} - \boldsymbol{x_A}) + (\boldsymbol{y} - \boldsymbol{H}(\boldsymbol{x_m}, \boldsymbol{p}))^T \mathbf{S_O}^{-1}(\boldsymbol{y} - \boldsymbol{H}(\boldsymbol{x_m}, \boldsymbol{p})) \tag{2}$$

where $\boldsymbol{x_A}$ represents the a-priori estimate of the state vector, $\boldsymbol{y}$ is the vector of observations used within the modelled time window, and $\boldsymbol{H}(\boldsymbol{x_m}, \boldsymbol{p})$ is the vector of model results at the times of the observations. The latter vector is the model equivalent of the observation vector $\boldsymbol{y}$. The superscript $T$ means transpose, $\mathbf{S_A}$ represents the a-priori error covariance matrix (defined in supplementary material) and $\mathbf{S_O}$ represents the matrix of observational error covariances. This cost function quantifies two aspects, namely the fit between model output and observations as well as how well the posterior state matches with prior information about the state. Regarding the observational error covariance matrix $\mathbf{S_O}$ (fully defined in Brasseur and Jacob, 2017), we assume for simplicity the observational errors to be uncorrelated (as in e.g. McNorton et al., 2018; Chevallier et al., 2010; Ma et al., 2021). This simplifies the matrix $\mathbf{S_O}$ to a diagonal matrix, with observational error variances as diagonal elements. This way Eq. (2) simplifies to:

$$J(\boldsymbol{x}) = (\boldsymbol{x} - \boldsymbol{x_A})^T \mathbf{S_A}^{-1}(\boldsymbol{x} - \boldsymbol{x_A}) + \sum_{i=1}^{m} \frac{(H(\boldsymbol{x_m}, \boldsymbol{p})_i - y_i)^2}{\sigma_{O,i}^2} \tag{3}$$

Here, $\sigma^2_{O,i}$ is the $i^{\text{th}}$ diagonal element of $\mathbf{S_O}$, $m$ is the number of observations. It is customary to refer to the first term of this cost function as the background part and to the second part of this function as the data part. Note that if also the a-priori errors are uncorrelated, the first term in the equation can be simplified in a similar way as the second term. The observational error variance linked to the $i^{\text{th}}$ observation ($\sigma^2_{O,i}$) can be further split up as follows:

$$\sigma^2_{O,i} = \sigma^2_{I,i} + \sigma^2_{M,i} + \sigma^2_{R,i} \tag{4}$$

where $\sigma^2_{I,i}$ is the instrument (measurement) error variance, $\sigma^2_{M,i}$ the model error variance and $\sigma^2_{R,i}$ the representation error variance (see Brasseur and Jacob, 2017). These errors are assumed to be independent of each other and normally distributed.

At this point, we introduce two extra features to the cost function. Firstly we allow the user to specify a weight for each individual observation, in case some observations are deemed less important than others. In principle, the observational error variances could also be adapted for this purpose, but by using weights we can keep realistic error estimations (important for Sect. 4.2). Those weights can also be used to manipulate the relative importance of the background term and the data term. This is similar to the "regularisation factor" explained in Brasseur and Jacob (2017). Secondly, we introduce part of our bias-correction scheme in the data part of the cost function, namely scaling factors for observations. These factors can also be optimised. With the additions mentioned above, the cost function as given in Eq. (3) modifies to:

$$J(\boldsymbol{x}) = (\boldsymbol{x} - \boldsymbol{x_A})^T \mathbf{S_A}^{-1}(\boldsymbol{x} - \boldsymbol{x_A}) + \sum_{i=1}^{m} w_i \frac{(H(\boldsymbol{x_m}, \boldsymbol{p})_i - s_i\, y_i)^2}{\sigma^2_{O,i}} \tag{5}$$

where $w_i$ $(-)$ is a weight for each individual observation in the cost function. $s_i$ $(-)$ is a scaling factor for observation $y_i$, identical for each timestep, but allowed to differ between each observation stream. The background term in the cost function can be left out if the user desires so. The introduction of the scaling factors means we need to adapt the observational error variances as well, the $i^{\text{th}}$ observational error variance is now given by:

$$\sigma^2_{O,i} = \text{var}(s_i^{\{t\}} y_i - H(\boldsymbol{x_m^{\{t\}}}, \boldsymbol{p})_i) \tag{6}$$

where $\boldsymbol{x_m^{\{t\}}}$ is the unknown vector of "true" values of the model parameters in the state vector, and $s_i^{\{t\}}$ is the "true" value of the scaling factor for the $i^{\text{th}}$ observation. The decomposition from Eq. (4) remains valid.

In the statistical optimisation, we attempt to find the values of the state vector x such that the function in Eq. (5) reaches its absolute minimum. This is done starting from an initial guess ($\boldsymbol{x} = \boldsymbol{x_A}$), after which the state vector is improved iteratively. The cost function and the gradient of the cost function (derivatives with respect to all parameters) are computed for different combinations of parameters in the state vector (Fig. 2). The framework uses by default a truncated Newton method, the *tnc* algorithm (The SciPy community; Nash, 2000), for the optimisations. Truncated Newton methods are suitable for non-linear optimisation problems (Nash, 2000). The chosen algorithm allows for specifying hard bounds on the state vector parameters, preventing unphysical parameter values for individual parameters in the state vector. Raoult et al. (2016) used similar constraints in their inverse modelling system, Bastrikov et al. (2018) used bound constraints as well. The analytical cost-function-gradient calculations are described in Sect. 3.4, a basic numerical derivative option (Sect. 3.5) is available as well, although we expect this in general to be outperformed by the analytical derivative.

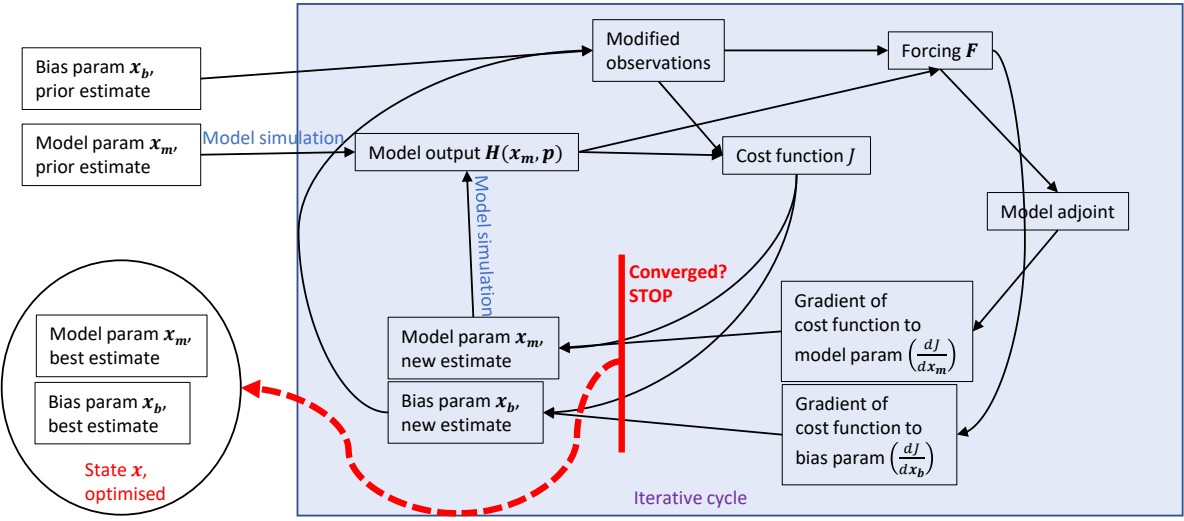

**Figure 2.** Slightly simplified sketch of the workflow of the inverse modelling framework, when using the adjoint model for the derivatives with respect to model parameters. Note that, for clarity of the figure, direct arrows between the parameters and the cost function and its gradients are not drawn. These arrows arise via the background part of the cost function (see equations in text). Everything within the shaded rectangle is part of the iterative cycle of optimisation. Model parameters that are not optimised are in vector $p$, this vector is used together with $x_m$ in every model simulation. In case ICLASS is run in Monte-Carlo mode (Sect. 3.6 and Sect. 4.2), this figure applies to the individual ensemble members.

## 3.2 State-vector parameters

As mentioned before, the state vector can be decomposed into a vector $x_m$ and a vector $x_b$. Vector $x_m$ contains state variables related to the input of CLASS, such as initial conditions (e.g. initial mixed-layer potential temperature), boundary conditions (e.g. $CO_2$ advection), and uncertain model constants (e.g. roughness length for heat). The full list of model parameters that can be optimised is given in the ICLASS manual.

Vector $x_b$ contains parameters belonging to our bias-correction scheme in the data part of the cost function. There are two ways of bias-correcting, the first one is by using observation scaling factors for observation streams. These scaling factors ($s_i$) have been introduced in Eq. (5). As an example, in the state vector to be optimised, one can include a single scaling factor for all $CO_2$ surface flux observations. The second possible method of bias correcting (Sect. 3.3) is implemented specifically for the energy balance closure problem (Foken, 2008; Oncley et al., 2007; Renner et al., 2019), it involves a parameter "Frac$_H$" ($-$) that can be optimised.

### 3.3 Bias correction for energy balance closure

We first define an observational energy balance closure residual (Foken, 2008):

$$\varepsilon_{\text{eb}} = \boldsymbol{R_n} - (\boldsymbol{H}_{\text{orig}} + \boldsymbol{LE}_{\text{orig}} + \boldsymbol{G}) \tag{7}$$

where $\boldsymbol{R_n}$ is the time series of net radiation measurements, $\boldsymbol{H}_{\text{orig}}$ and $\boldsymbol{LE}_{\text{orig}}$ are the measured sensible and latent heat fluxes and $\boldsymbol{G}$ is the measured soil heat flux. The difference between measured net radiation and the sum of measured heat fluxes is calculated for every time step and this represents the energy balance closure residual. If desired, the user can easily specify an expression of their own for $\varepsilon_{\text{eb}}$. Subsequently, the observations for the sensible and latent heat flux are adapted as follows:

$$\boldsymbol{y_H} = \boldsymbol{H}_{\text{orig}} + \text{Frac}_H\, \varepsilon_{\text{eb}} \tag{8}$$

$$\boldsymbol{y_{LE}} = \boldsymbol{LE}_{\text{orig}} + (1 - \text{Frac}_H)\varepsilon_{\text{eb}} \tag{9}$$

This implies that the energy balance closure residual is added partly to the sensible, partly to the latent heat flux, thereby closing the energy balance in the observations. This partitioning is determined by parameter $\text{Frac}_H$, which is taken to be constant during the modelled period. This approach of closing the energy balance is similar to Renner et al. (2019), but we optimise a parameter, instead of using the evaporative fraction, for partitioning $\varepsilon_{\text{eb}}$. Limitations of this approach are that we assume the radiation and soil heat flux measurements to be bias-free, and the $\text{Frac}_H$ parameter constant.

### 3.4 Analytical derivative

For the optimisations, we do not only compute a cost function, but we also use the gradient of the cost function with respect to the state vector elements. This informs us on the direction in which the cost function lowers. How the gradient with respect to an individual element is calculated depends on which state vector element is considered. In case the $i^{\text{th}}$ state vector element is a model-input parameter, the gradient with respect to this element is computed as (similar to Brasseur and Jacob, 2017, their Eq. 11.105):

$$\frac{\partial J}{\partial x_i} = 2(\mathbf{S_A}^{-1}(\boldsymbol{x} - \boldsymbol{x_A}))_i + 2((\boldsymbol{\nabla}_{\boldsymbol{x_m}}\boldsymbol{H}(\boldsymbol{x_m}, \boldsymbol{p}))^T \boldsymbol{F})_{i-l} \tag{10}$$

Where $\boldsymbol{\nabla}_{\boldsymbol{x_m}}\boldsymbol{H}(\boldsymbol{x_m}, \boldsymbol{p})$ is the local Jacobian matrix of $H$, $l$ is the number of non-model-input parameters in the state $\boldsymbol{x}$ occurring before $x_i$, and $\boldsymbol{F}$ is the forcing vector, with elements $F_k$ defined as:

$$F_k = w_k \frac{(H(\boldsymbol{x_m}, \boldsymbol{p})_k - s_k\, y_k)}{\sigma_{O,k}^2} \tag{11}$$

There is one forcing element in the vector for every observation that is used, $F_k$ is the forcing related to the observation $y_k$. For calculating the model-input part of the analytical derivative we constructed the adjoint of the model, $(\boldsymbol{\nabla}_{\boldsymbol{x_m}}\boldsymbol{H}(\boldsymbol{x_m}, \boldsymbol{p}))^T$, which is used to obtain a locally exact analytical gradient (in specific cases not exact, see section "If-statements" in supplementary material). More information on the adjoint is given in the supplementary material.

In case the $i^{\text{th}}$ state vector element is an observation scaling factor (Eq. 5) for observation stream $j$, the gradient of the cost function with respect to the $i^{\text{th}}$ state vector element is computed as:

$$\frac{\partial J}{\partial x_i} = 2(\mathbf{S_A}^{-1}(\boldsymbol{x} - \boldsymbol{x_A}))_i + \sum_{k=1}^{m_j} -2F_k\, y_{j,k} \tag{12}$$

where $m_j$ is the number of observations of the type (stream) where the observation scaling factor is applied to, e.g. if the observation scale is a scaling factor for surface $CO_2$ flux observations, $m_j$ is the number of surface $CO_2$ flux observations. $y_{j,k}$ is the $k^{\text{th}}$ observation of this observation stream. $F_k$ is the forcing related to the observation $y_{j,k}$.

Finally, if the $i^{\text{th}}$ state vector element is $\text{Frac}_H$, the gradient is calculated as

$$\frac{\partial J}{\partial x_i} = 2(\mathbf{S_A}^{-1}(\boldsymbol{x} - \boldsymbol{x_A}))_i - 2\left(\boldsymbol{F_H} \cdot \frac{d\boldsymbol{y_H}}{d\text{Frac}_H} + \boldsymbol{F_{LE}} \cdot \frac{d\boldsymbol{y_{LE}}}{d\text{Frac}_H}\right) \tag{13}$$

Where $\boldsymbol{F_H}$ and $\boldsymbol{F_{LE}}$ are the forcing vectors for the sensible and latent heat flux respectively, $\boldsymbol{y_H}$ and $\boldsymbol{y_{LE}}$ are the observation vectors for the sensible and latent heat flux respectively and "$\cdot$" is the Euclidian inner product. Note that when the $\text{Frac}_H$ parameter is included in the state, the observation scaling factors for sensible and latent heat flux observations will be set equal to 1, and are not allowed to be included in the state. The terms $\frac{d\boldsymbol{y_H}}{d\text{Frac}_H}$ and $\frac{d\boldsymbol{y_{LE}}}{d\text{Frac}_H}$ represent respectively the derivatives of the sensible and latent heat flux observations to the $\text{Frac}_H$ parameter, which follow from equations 8 and 9 as follows:

$$\frac{d\boldsymbol{y_H}}{d\text{Frac}_H} = \varepsilon_{\text{eb}} \tag{14}$$

$$\frac{d\boldsymbol{y_{LE}}}{d\text{Frac}_H} = -\varepsilon_{\text{eb}} \tag{15}$$

Note that equations 10, 12 and 13 all have the same first term that originates from the background part of the cost function.

## 3.5  Numerical derivative

A simple numerical derivative is available as alternative to the analytical gradient. The derivative of the cost function to the $i^{\text{th}}$ state element is numerically calculated as

$$\frac{\partial J}{\partial x_i} = \frac{J(x_i + \alpha) - J(x_i - \alpha)}{2\alpha} \tag{16}$$

where $\alpha$ is a very small perturbation to state parameter $x_i$, with a default value of $10^{-6}$, and has the units of $x_i$.

## 3.6  Handling convergence challenges

The highly non-linear nature of the optimisation problem can cause the optimisation to get stuck in a local minimum of the cost function (Santaren et al., 2014; Bastrikov et al., 2018; Ziehn et al., 2012). This means that the resulting posterior state vector can depend on the prior starting point (Raoult et al., 2016), and the resulting posterior state can remain far from optimal. In the worst case, the non-linearity of the model can even lead to a crash of the forward model. This happens with certain combinations of input parameters, that lead to unphysical situations or undesired numerical behaviour. After starting from a

user-specified prior state vector, the *tnc* algorithm autonomously decides which parameter values are tested during the rest of the optimisation. It is possible to place hard bounds on individual parameters when using the *tnc* algorithm, but this does not

always prevent all possible problematic combinations of parameters from being evaluated.

To obtain information about the uncertainty of the posterior solution, and to deal with the challenges described above, the framework also allows a Monte-Carlo approach (Tarantola, 2005). This entails that the framework does not start at a single state vector with prior estimates, but instead uses an *ensemble* of prior state vectors $x_A$, leading to an ensemble of posterior parameter estimates. The ensemble of optimisations can be executed in parallel on multiple processors, thereby reducing the

time it takes to perform the total optimisation. More details on the Monte-Carlo mode of ICLASS are given in the Sect. 4.2.

As an additional way to improve the posterior solution, we have implemented a "restart" algorithm. If the optimisation results in a cost function that is higher than a user-specified number, the framework will restart the optimisation from the best state reached so far. This "fresh start" of the *tnc* algorithm, whereby the algorithm's memory is cleared, often leads to a further lowering of the cost function. The maximum number of restarts (>=0) is specified by the user. If an ensemble is used, every

individual member with a too high posterior cost function will be restarted.

## 4  Error statistics

### 4.1  Prior and observations

From a Bayesian point of view, ICLASS can combine information, both from observations and from prior knowledge about the state vector, to come to a solution with a reduced uncertainty in the state parameters. In the derivation of Eq. (2), it is

assumed that both prior and observational errors follow a (multivariate) normal distribution (Tarantola, 2005). However, some prior input parameters are bounded (Sect. 3.6), e.g. the albedo cannot be negative. Dealing with bounds is a known challenge in inverse modelling, several approaches can be followed (Miller et al., 2014). As an example, Bergamaschi et al. (2009) had to deal with methane emissions in the state vector becoming negative. Their solution was to make the emissions a function of an emission parameter that is being optimised, instead of optimising the emissions themselves. By their choice of function, the

emissions cannot become negative, even though the emission parameter is unbounded. In our case such an approach is more difficult, as we have a diverse set of bounded parameters.

Our approach is to enforce hard bounds for state values via the *tnc* algorithm, it however means that the normality assumption will be violated to some extent, as the normal distribution (for which the user specifies the variance) for prior parameter values then becomes a truncated normal distribution. For a parameter following a truncated normal prior distribution, the prior variance

used in the cost function is not (fully) equal to the variance of the actual prior distribution. The extent to which this is the case, depends on the degree of truncation.

Our system also allows for specifying covariances between state elements in $\mathbf{S_A}$. We assume observational errors to be uncorrelated, see Sect. 3.1. Equation 4 states that the observational error variance consists of an instrument error, a model error and a representation error variance. The instrument and representation error standard deviation are taken from user input,

the model error standard deviation can either be specified by the user or estimated from a sensitivity analysis. In the latter

case, an ensemble of forward-model runs is performed. For each of these runs, a set of parameters, not belonging to the state vector, is perturbed (so parameters part of vector $p$, not from $x_m$, see Sect. 3.1). These perturbed parameters are used together with the unperturbed prior vector $x_m$ as model input. The user should specify which parameters should be perturbed, and the distribution from which random numbers will be sampled to add to the default (prior) model parameters. For the latter, there is the choice between a "normal", "bounded normal", "uniform" or "triangular" distribution. The model error standard deviation for each observation stream at each observation time, is then obtained from the spread in the ensemble of model output for the observation stream at that time.

## 4.2 Ensemble mode to estimate posterior uncertainty

When ICLASS is run in Monte-Carlo (ensemble) mode, it allows for estimating the uncertainty of the posterior state. The principle for obtaining posterior error statistics bears strong similarity to what has been done in Chevallier et al. (2007), it will be shortly explained here. We start with the prior state specified by the user. Then, for each ensemble member and for every state parameter, a random number is drawn from a normal distribution with variance of the respective parameter specified by the user, and mean 0. When also covariances are specified for the prior parameters, a multivariate normal distribution can be used to sample from. The sampled numbers are then added to the state vector. In case any non-zero covariances are used to perturb, then, if the new state vector falls out of bounds (if specified), it is discarded and the procedure repeated for that ensemble member. In case no covariance structure is used, only the parameter which falls out of bounds has to be replaced. It has to be noted that this effectively leads to sampling from a truncated (multivariate) normal distribution.

Similarly, for every ensemble member a random number drawn from a normal distribution (without bounds) is added to the (scaled) observations. This modifies the cost function of an individual ensemble member (Eq. 5) as follows:

$$J(\boldsymbol{x}) = \left(\boldsymbol{x} - \boldsymbol{x}_{\boldsymbol{A}}^{\{\boldsymbol{p}\}}\right)^T \mathbf{S_A}^{-1} \left(\boldsymbol{x} - \boldsymbol{x}_{\boldsymbol{A}}^{\{\boldsymbol{p}\}}\right) + \sum_{i=1}^{m} w_i \frac{(H(\boldsymbol{x_m}, \boldsymbol{p})_i - s_i\, y_i + \varepsilon_i)^2}{\sigma_{O,i}^2} \tag{17}$$

$\boldsymbol{x}_{\boldsymbol{A}}^{\{\boldsymbol{p}\}}$ is the perturbed prior state vector. The random numbers $\varepsilon_i$ in the data part of the cost function are sampled from normal distributions with mean 0 and standard deviation $\sigma_{O,i}$. This can be seen as either a perturbation of the scaled observations or a perturbation of the model-data mismatch. As a consequence of the change in the cost function equation, the equation for the forcing vector (Eq. 11) is updated as well:

$$F_k = w_k \frac{(H(\boldsymbol{x_m}, \boldsymbol{p})_k - s_k\, y_k + \varepsilon_k)}{\sigma_{O,k}^2} \tag{18}$$

Furthermore, in equations 10, 12 and 13, for perturbed ensemble members, $\boldsymbol{x}_{\boldsymbol{A}}^{\{\boldsymbol{p}\}}$ is used instead of $\boldsymbol{x_A}$. The vector $\boldsymbol{\varepsilon}_{\mathrm{eb}}$ (Eq. 7) is not perturbed and kept identical for all members. For every ensemble member an optimisation is performed. Each optimisation is classified as either successful or not successful. Here, successful is defined as having a final reduced chi-squared (see Sect. 5, Eq. 19) smaller or equal than a user-specified number. The posterior state parameters for the successful members are considered a sample of the posterior state space, and are saved in a matrix. Those parameters can be used to estimate the true posterior pdfs. A covariance and a correlation matrix is then constructed using the matrix of posterior state parameters. These covariance

and correlation matrices are used as an estimate of the true posterior state covariance and correlation matrices, respectively. If the user prefers so, also parameters not belonging to the state can be perturbed in the ensemble, in a similar way as outlined in Sect. 4.1.

Note that, next to the ensemble, there will also be a run with an unperturbed prior, just as is the case when no ensemble is used. We refer to this run as member 0. This member is however not included in the calculation of ensemble-based error statistics (e.g. correlation matrix), as the choice of prior values is not random for this member.

## 5    Output

ICLASS can write several output files. The output includes the obtained parameters, as well as the posterior reduced chi-squared
statistic and the posterior and prior cost function. The reduced chi-squared goodness-of-fit statistic is defined in ICLASS as:

$$\chi_r^2 = \frac{J}{\sum_{i=1}^{m}(w_i) + n} \tag{19}$$

In this equation, $m$ is the number of observations and $n$ the number of state parameters (see Sect. 3 for the other symbols). Note that if all the weights are set to 1, the denominator simplifies to $m+n$. The $n$ will only be included in the denominator when the user chooses to include a background part in the cost function (default). In a simple case where the prior errors are
uncorrelated, this $\chi_r^2$ statistic for the posterior solution should be around 1 when the optimisation converges well and priors and errors are properly specified (e.g. Michalak et al., 2005). In a case where all weights are 1, this can be understood as follows: The average value of the $i^{\text{th}}$ posterior observation residual squared, $(H(\boldsymbol{x_{m}}_{,\text{post}}, \boldsymbol{p})_i - s_i\, y_i)^2$, should be close to $\sigma_{O,i}^2$, and the average value of the $i^{\text{th}}$ posterior data residual squared, $(x_{\text{post,i}} - x_{\text{A,i}})^2$, should be close to the $i^{\text{th}}$ diagonal element of the a-priori error covariance matrix when the optimisation converges well and errors and prior parameters are properly specified. We have
$m$ observation residuals, and $n$ data residuals (if background part included). In this example with a diagonal $\mathbf{S_A}$ matrix, the residuals are assumed to be independent of each other. Each squared residual contributes on average a value of approximately 1 to the cost function, summing to approximately $m+n$, and thus $\chi_r^2 \approx 1$. Note however that the $\chi_r^2$ statistic can be misleading, in particular when observational errors are correlated (Chevallier, 2007). Furthermore, as mentioned in Sect. 4.1, prior parameters can follow a truncated normal distribution, violating the normality assumption. The impact of this depends on the degree of
truncation, but also on the number of observations etc. It can lead to an ideal $\chi_r^2$ value diverting from 1.

ICLASS also calculates a prior and a posterior partial cost function and a posterior $\chi_r^2$ statistic for individual observation streams and for the correspondence to prior information (background). The partial cost function for observation stream $j$ is calculated as:

$$J_j = \sum_{i=1}^{m_j} w_i \frac{(H(\boldsymbol{x_m}, \boldsymbol{p})_i - s_j\, y_{j,i} + \varepsilon_i)^2}{\sigma_{O,i}^2} \tag{20}$$

where $m_j$ is the number of observations of stream $j$, $y_{j,i}$ is the $i^{\text{th}}$ observation of stream $j$, and $H(\boldsymbol{x_m}, \boldsymbol{p})_i$ is the model output corresponding with $y_{j,i}$. In case scaled observations are perturbed in the ensemble (Sect. 4.2), $\varepsilon_i$ is a random number, otherwise

it is 0. The $\chi_r^2$ statistic for observation stream $j$ is defined here as:

$$\chi_{r,j}^2 = \frac{J_j}{\sum_{i=1}^{m_j}(w_i)} \tag{21}$$

This equation resembles Eq. (13) of Meirink et al. (2008), but note that variable $\frac{\chi^2}{n_s}$ in their paper is similar to $\chi_{r,j}^2$ in our paper. The $\chi_r^2$ statistic for the background part is calculated as (similar to Eq. 26 of Michalak et al., 2005):

$$\chi_{r,b}^2 = \frac{J_b}{n} \tag{22}$$

Where $J_b$ is the background part of the cost function, i.e. the first term from Eq. (5). The mean bias error, root mean squared error and the ratio of model and observation variance for every observation stream is also calculated, both for the prior and posterior state. The mean bias error for the $j^{\text{th}}$ observation stream is defined here as:

$$\varepsilon_{\text{bias},j} = \frac{1}{m_j} \sum_{i=1}^{m_j}(H(\boldsymbol{x_m},\boldsymbol{p})_{j,i} - s_j\, y_{j,i}) \tag{23}$$

In this equation, $m_j$ is the number of observations of type (stream) $j$, $H(\boldsymbol{x_m},\boldsymbol{p})_{j,i}$ is the model output for observation stream $j$ at time index $i$, $y_{j,i}$ is an observation of stream $j$ at time index $i$ and $s_j$ is an observation scaling factor for obs of stream $j$. Note that even when the scaled observations are perturbed, we do not include those perturbations here (compare Eq. 20). In case the $\text{Frac}_H$ parameter is used, the energy balance corrected observations (equations 8 and 9) will be used in the equation above. For the root mean squared error and the ratio of model and observation variance, the observation scales and energy balance corrected observations are used as well. ICLASS also calculates the normalised deviation of the posterior from the prior for every state parameter $x_i$:

$$\delta_{\text{nor},i} = \frac{x_{\text{post},i} - x_{A,i}}{\sigma_{A,i}} \tag{24}$$

where $\sigma_{A,i}$ is the square root of the prior variance of parameter $i$, i.e. the square root of diagonal element $i$ from matrix $\mathbf{S_A}$. Note that, also in case of an ensemble, we use the unperturbed prior, i.e. the prior of member 0.

If run in ensemble mode, we additionally store estimates of the posterior error covariance matrix (Sect. 4), the posterior error correlation matrix, the posterior/prior variance ratio for each of the state parameters, the mean posterior state, and the optimised and prior states for every single ensemble member. When non-state parameters are perturbed in the ensemble, these parameters are part of the output, also the correlation of the posterior state parameters with these non-state parameters can be written as output. When the model error standard deviations are estimated by ICLASS (Sect. 4.1), there is a separate file containing statistics on the estimated error standard deviations. Finally, there are additional output files with information about the optimisation process. For every model simulation (and for every ensemble member) in the iterative optimisation process, one can find the parameter values used in this simulation as well as the value of the cost function, split up into a data and a background part. For the gradient calculations, one can find the parameter values used, as well as the the derivatives of the cost function with respect to every state parameter. The derivatives of the background part are also provided separately. More details on the output of ICLASS can be found in a separate section of the manual. An overview of the output variables defined in this section is given in Table B2.

## 6  Technical details of the code

We have written the entire code in Python 3 (https://www.python.org). Using the framework requires a Python 3 installation, including the NumPy (van der Walt et al. (2011); https://numpy.org), SciPy (https://scipy.org) and Matplotlib (Hunter (2007); https://matplotlib.org) libraries. The code is operating-system independent and consists of three main files:

- "forwardmodel.py", containing the adapted CLASS model.

- "inverse_modelling.py", containing most of the inverse modelling framework, such as the function to be minimised in the optimisation and the model adjoint.

- "optimisation.py", the file to be run by the user for performing an actual optimisation, in this file observations are loaded, the state vector defined, etc. The input paragraphs in this file should be adapted by the user to the optimisation performed.

There are three additional files, the first is "optimisation_OSSE.py", and is similar to optimisation.py, but is meant specifically for observation system simulation experiments (also in this file, the user should adapt the input paragraphs to the optimisation to be performed). The second file is "testing.py", containing tests for the code as described in Sect. 7. The file "postprocessing.py" is a script that can be run after the optimisations are finished, for post-processing output data, e.g. to plot a colored matrix of correlations. Using the Pickle module, the optimisation can store variables on the disk near the end of the optimisation, and these variables can be read in again in the "postprocessing.py" script. This has the advantage that e.g. the formatting of plots can easily be adapted without redoing the optimisation. This script should be adapted by the user to the optimisation performed and the output desired.

## 7  Adjoint model validation

When using the adjoint modelling technique, an extensive testing system is required to make sure that the analytical gradient of the cost function computed by the adjoint model is correct. There are two tests that are essential, which are described below.

The gradient test (for the tangent linear model) is a test to determine whether the derivatives with respect to the model input parameters in the tangent linear model are constructed correctly. The construction of the adjoint is based on the tangent linear model, so errors in the tangent linear propagate to the adjoint model. In the gradient test, model input state variables are perturbed, which leads to a change in model output. The change in model output when employing the tangent linear model is compared to the change in model output when a numerical finite difference approximation is employed. Mathematically, the test for the $i^{\text{th}}$ model output element can be written as (similar to Honnorat et al., 2007; Elizondo et al., 2000):

$$\frac{dH(\boldsymbol{x_m}, \boldsymbol{p})_i}{d\boldsymbol{x_m}} \cdot \boldsymbol{\Delta x_m} \approx \frac{H(\boldsymbol{x_m} + \alpha \boldsymbol{\Delta x_m}, \boldsymbol{p})_i - H(\boldsymbol{x_m}, \boldsymbol{p})_i}{\alpha} \tag{25}$$

where $\boldsymbol{\Delta x_m}$ is a vector of ones, with the same length as vector $\boldsymbol{x_m}$. $\alpha$ is a small positive number, $H$ represents the forward-model operator, $H(\boldsymbol{x_m}, \boldsymbol{p})_i$ is the $i^{\text{th}}$ model output element, $\boldsymbol{x_m}$ is the vector of model input variables to be tested (model-parameters part of state), "$\cdot$" is the Euclidian inner product, and vector $\boldsymbol{p}$ is the set of non-state parameters used by the model.

**Table 1.** A simple gradient test example involving the derivative of the 2 m temperature with respect to the roughness length for heat ($z_{0h}$). The right column gives, for different values of perturbation $\alpha$ (m), the result of 1 - RHS/LHS, whereby RHS is the right hand side of Eq. 25, and LHS the left hand side of the same equation. Note that only the RHS of the equation is influenced by $\alpha$. In this example we have only perturbed parameter $z_{0h}$, but multiple parameters can be perturbed within one test.

| $\alpha$(m) | 1 - ratio RHS and LHS $(-)$ |
|---|---|
| 0.5 | $4.7 \times 10^{-1}$ |
| 0.2 | $3.2 \times 10^{-1}$ |
| 0.1 | $2.2 \times 10^{-1}$ |
| $1 \times 10^{-2}$ | $3.4 \times 10^{-2}$ |
| $1 \times 10^{-3}$ | $3.6 \times 10^{-3}$ |
| $1 \times 10^{-4}$ | $3.7 \times 10^{-4}$ |
| $1 \times 10^{-5}$ | $3.7 \times 10^{-5}$ |
| $1 \times 10^{-6}$ | $4.2 \times 10^{-6}$ |
| $1 \times 10^{-7}$ | $2.6 \times 10^{-6}$ |
| $1 \times 10^{-8}$ | $-8.2 \times 10^{-6}$ |
| $1 \times 10^{-9}$ | $5.9 \times 10^{-4}$ |
| $1 \times 10^{-12}$ | $-8.2 \times 10^{-2}$ |

Several increasingly smaller values are tested for $\alpha$. When the tangent linear model is correct, the right-hand side of the equation will converge to the left-hand side when $\alpha$ gets progressively smaller, although for too small $\alpha$ numerical errors start to arise (Elizondo et al., 2000). Instead of using the full tangent linear model, individual model statements can also be checked. In this case $\boldsymbol{x_m}$ and $H(\boldsymbol{x_m}, \boldsymbol{p})_i$ can contain intermediate (not part of model input or output) model variables. The gradient test is considered successful if the ratio of the left- and right-hand sides of the equation lies in the interval [0.999 - 1.001]. The results of a simple example of a gradient test, involving the derivative of the 2 m temperature with respect to the roughness length for heat ($z_{0h}$), is shown in Table 1.

The dot-product (adjoint) test checks whether the adjoint model code is correct for the given tangent linear code. It tests whether the identity

$$\langle \mathbf{H_L}\boldsymbol{x_m}, \boldsymbol{y} \rangle = \langle \boldsymbol{x_m}, \mathbf{H_L^T}\boldsymbol{y} \rangle \tag{26}$$

holds (Krol et al., 2008; Meirink et al., 2008; Honnorat et al., 2007; Claerbout, 1992). The equation can also be written as

$$\frac{\langle \mathbf{H_L}\boldsymbol{x_m}, \boldsymbol{y} \rangle - \langle \boldsymbol{x_m}, \mathbf{H_L^T}\boldsymbol{y} \rangle}{\langle \mathbf{H_L}\boldsymbol{x_m}, \boldsymbol{y} \rangle} = 0 \tag{27}$$

In this equation, $\mathbf{H_L} = \boldsymbol{\nabla}_{\boldsymbol{x_m}} \boldsymbol{H}(\boldsymbol{x_m}, \boldsymbol{p})$ represents the tangent linear model operator, i.e. a matrix (the local Jacobian of $H$) with the element on the $i^{\text{th}}$ row and $j^{\text{th}}$ column given by $\frac{dH(\boldsymbol{x_m}, \boldsymbol{p})_i}{dx_{m,j}}$. $\boldsymbol{x_m}$ is in this equation a vector with perturbations to the model state, $\mathbf{H_L^T}$ represents the adjoint model operator and "$\langle \rangle$" is the Euclidian inner product. A very small deviation

from 0 is acceptable due to machine rounding errors (Claerbout, 1992). Our criterion for passing the test, when using 64-bit floating-point calculations, is that the absolute value of the left-hand side of Eq. 27 should be $<= 5 \times 10^{-13}$. This test can also be applied to individual statements or a block of statements in the adjoint model code. In this case the definitions of the variables in Eq. 27 slightly change, e.g. $\mathbf{H_L^T}$ then represents a part of the adjoint model. The adjoint test will be illustrated by a (slightly simplified) example from the model code, the well-known Stefan–Boltzmann's law for outgoing longwave radiation. The forward-model code reads:

```
Lwout = bolz * Ts ** 4.
```

The tangent linear model code for this statement reads:

```
dLwout = bolz * 4 * Ts ** 3. * dTs
```

And the corresponding adjoint code:

```
adTs += bolz * 4 * Ts ** 3. * adLwout
adLwout = 0
```

In this case $\boldsymbol{x_m}$ corresponds to *dTs*, the $\mathbf{H_L}\boldsymbol{x_m}$ variable is *dLwout*. The vector $\boldsymbol{y}$ is *adLwout* and $\mathbf{H_L^T}\boldsymbol{y}$ is *adTs*. In the test $\boldsymbol{x_m}$ and $\boldsymbol{y}$ are assigned random numbers. When we evaluated Eq. (27) on this part of the code, the result was less than $1 \times 10^{-15}$ (which corresponds to approximately $5 \times$ machine precision), meaning that the test passes.

We have constructed a separate script that performs a vast amount of gradient tests and adjoint tests and informs the user whenever a test fails. This file (testing.py) is included with the code files. The number of time steps we specified for testing is small, as the computational burden increases with the amount of time steps. The model passes the vast majority of the tests that are executed in this file. Closer inspection of the tests that fail reveals that numerical noise is a likely explanation for the small fraction of tests that are labelled as failing. Further validation of the inverse modelling framework follows in the next section.

## 8 Inverse modelling validation: Observation System Simulation Experiments

To test the ability of the system to properly estimate model parameters, we show in this chapter the results of observation system simulation experiments with artificial data. This type of experiments is classic to test the ability of the system to properly estimate model parameters. For example, similar tests have been performed by Henze et al. (2007), for their adjoint of a chemical transport model. In our experiments we simulate a growing convective boundary layer for a location at mid-latitudes from 10–14 h, including surface layer calculations. In the chosen setup, the land surface is coupled to the boundary layer. The land surface provides heat and moisture, and exchanges $CO_2$ with the CBL.

### 8.1 Parameter estimation

We perform in this section a total of five main experiments of varying complexity. The things that differ among experiments, are the choice of observations and state vectors. The procedure for the first four experiments is as follows. We first run the

model with chosen values of a set of parameters we want to optimise. A set of model output data from this simulation then serves as the observations, while the parameters used to create these observations are referred to as the "true" parameters.

Then we perform an optimisation using these observations, starting from a perturbed prior state vector. In the cost function, we do not include the background part, to make sure that it is possible to find back the "true" parameters. This is because the background part of the cost function implies a "penalty" for deviating from the prior state. This penalty implies that, when the model is run with the true parameters, the cost function would still not be 0. Next to that, the minimum of the cost function is (generally) shifted. When leaving out the background part, the framework should, if converging properly, be able to find back

the parameter values that were used for creating the observations, starting from any other prior state vector.

Since real-life observations usually contain noise, the last experiment follows the same procedure, except that the observations are also perturbed. This is done by adding white Gaussian noise to the observations. For each observation $y_i$, a random number drawn from a normal distribution with mean 0 and standard deviation equal to a specified $\sigma_{I,i}$ (Table 3), is added. The model and representation errors are set to 0 in all experiments. All weights for the cost function were set to 1. Table 2 list for

every experiment the prior, the "true" and the optimised state variables. The complexity of the experimental setup increases from the top to the bottom of the table.

For the first experiment, we perturbed the initial boundary-layer height and albedo. The rest of the parameters are left unchanged compared to the true parameters. The observation streams we used are specific humidity and boundary-layer height. The specific humidity is expected to be influenced by the albedo, as the amount of available net radiation is relevant for

the amount of evapotranspiration, which influences the specific humidity. The boundary-layer height is relevant for specific humidity as well, as it determines the size of the mixing volume in which evaporated water is distributed. The state parameters are thus expected to have a profound influence on the cost function.

The optimised model shows a very good fit to the observations (Fig. 3), and the original parameter values are found back with a precision of at least five decimal places (Table 2, no five decimal places shown). This result indicates the optimisations

work in such simple situations.

To test the system for a more complex problem, we change from two parameters in the state to five (Table 2, second block), while keeping the same observations. Again, based on physical reasoning, the added parameters are expected to influence the cost function. However, one might expect that the observations might not be able to uniquely constrain the state, i.e. parameter interdependency issues might arise for this setup (equifinality). As an example, a reduction in soil moisture might influence

the observations in a similar way as a reduction in stomatal conductance. Mathematically, this means that multiple nearly equivalent local minima might be present in the cost function.

Also for this more complex case, we manage to get a very good fit (Fig. 3, c and d). However, inspecting the obtained parameter values (Table 2, second block, column "2 obs streams") we see that the optimisation does not fully converge to the "true" state. This is likely caused by parameter interdependency, as described earlier. To solve this, extra observation

streams need to be added that allow to disentangle the effects of the different parameters in the state. We have tested this in the third experiment, by adding the sensible and latent heat flux, the temperature at 2 m height and the surface $CO_2$ flux to the observations. With this more complete set of observations, the true parameter values are found back with a precision of

**Table 2.** Parameter values for the main observation system simulation experiments of Sect. 8.1. The observation streams used in the 2-obs-streams experiments are $q$ and $h$, for the 6 obs streams experiment $H$, $LE$, $T_2$ and $F_{CO2}$ are added, for the 7-obs-streams experiments additionally $CO2_{20}$ is added. The value of e.g. parameter $z_{0m}$ (only a state element in the 10-parameter-state experiments) is in the 2-parameter-state and 5-parameter-state experiments equal to the true value of $z_{0m}$ in the 10-parameter state experiments. See Table A1 for a description of the parameters, and Table A2 for a description of the observation streams.

| Parameter | True | Prior | Optimised | |
|---|---|---|---|---|
| | 2-parameter state | | 2 obs streams | |
| $h$ (m) | 350.0 | 650.0 | 350.0 | |
| $\alpha_{\mathrm{rad}}$ (−) | 0.200 | 0.450 | 0.200 | |
| | 5-parameter state | | 2 obs streams | 6 obs streams |
| $h$ (m) | 350.0 | 650.0 | 350.0 | 350.0 |
| $\alpha_{\mathrm{rad}}$ (−) | 0.200 | 0.450 | 0.200 | 0.200 |
| $\alpha_{\mathrm{sto}}$ (−) | 1.00 | 0.50 | 0.96 | 1.00 |
| $w_g$ (−) | 0.270 | 0.140 | 0.306 | 0.270 |
| $\gamma_\theta$ (K m$^{-1}$) | 0.0030 | 0.0050 | 0.0030 | 0.0030 |
| | 10-parameter state | | 7 obs streams | |
| | | | no noise | noise |
| $h$ (m) | 350.0 | 650.0 | 350.0 | 344.4 |
| $\alpha_{\mathrm{rad}}$ (−) | 0.200 | 0.450 | 0.200 | 0.207 |
| $\alpha_{\mathrm{sto}}$ (−) | 1.00 | 0.50 | 1.00 | 1.01 |
| $w_g$ (−) | 0.270 | 0.140 | 0.270 | 0.238 |
| $\gamma_\theta$ (K m$^{-1}$) | $3.0 \times 10^{-3}$ | $5.0 \times 10^{-3}$ | $3.0 \times 10^{-3}$ | $3.1 \times 10^{-3}$ |
| $\gamma_q$ (kg kg$^{-1}$m$^{-1}$) | $-1.0 \times 10^{-6}$ | $-3.0 \times 10^{-6}$ | $-1.0 \times 10^{-6}$ | $-1.1 \times 10^{-6}$ |
| $CO_2$ (ppm) | 422.0 | 380.0 | 422.0 | 423.2 |
| $adv_{CO2}$ (ppm s$^{-1}$) | 0.0 | $6.0 \times 10^{-3}$ | $9.8 \times 10^{-8}$ | $-1.4 \times 10^{-4}$ |
| $z_{0m}$ (m) | 0.020 | 0.100 | 0.020 | 0.048 |
| $z_{0h}$ (m) | 0.020 | 0.010 | 0.020 | 0.013 |

at least four decimal places (column "6 obs streams" in Table 2). As a side experiment, not included in Table 2, we have run the same experiment as before, but using the numerical derivative described in Sect. 3.5. In this case, convergence is notably slower, e.g. more than 6 times as many iterations were needed to reduce the cost function to less than 0.1 % of its prior value. This indicates that, at least for this more complex non-linear optimisation case, an analytical gradient outperforms our simple numerical gradient calculation.

In the fourth main experiment, we increase our number of state parameters to 10 (Table 2). To further constrain the parameters, we expand our set of observation streams with the $CO_2$ mixing ratio measured at 20 m height. Also for this complex

**Table 3.** The root mean squared error (RMSE) and the specified measurement error standard deviation ($\sigma_I$) for the observation streams in the 10-parameter-state OSSE with perturbed observations. In all our OSSEs, the measurement error standard deviations are chosen to be constant with time. Note that the measurement error equals the observational error here, since the model and representation errors were set to 0 for simplicity. The $\sigma_I$ are identical for all OSSEs in Table 2, but not all observation streams are used in each OSSE.

| Observation stream | RMSE prior | RMSE optimised | $\sigma_I$ |
|---|---|---|---|
| $q$ (kg kg$^{-1}$) | $4.5 \times 10^{-4}$ | $3.6 \times 10^{-4}$ | $4.0 \times 10^{-4}$ |
| $h$ (m) | 365.2 | 91.9 | 100.0 |
| $H$ (W m$^{-2}$) | 91.7 | 22.8 | 25.0 |
| $LE$ (W m$^{-2}$) | 128.6 | 22.7 | 25.0 |
| $T_2$ (K) | 1.50 | 0.62 | 0.65 |
| $F_{CO2}$ (mg CO2 m$^{-2}$s$^{-1}$) | $42 \times 10^{-2}$ | $3.6 \times 10^{-2}$ | $4.0 \times 10^{-2}$ |
| $CO2_{20}$ (ppm) | 30.5 | 1.7 | 2.0 |

setup, the framework managed to find the true values back, with a precision of one decimal place for initial boundary-layer height, and more for the other parameters.

These tests show that (at least for the range of tests performed) the framework is able to find the minimum of the cost function well. Additionally, we performed another observation system simulation experiment to test the framework's usefulness in a more realistic situation with noise on the observations. The result can be seen in Figure 3 e–h, the framework finds a good fit to the observations, even though it has now become impossible to fit every single (perturbed) observation. The posterior root mean squared error is about the same size (slightly smaller) as the observational error standard deviation for all observation streams (Table 3). This confirms the visual good fit from Fig. 3 e–h, given that these observational error standard deviations were used to create the random perturbations for the observations. The $\chi_r^2$ statistic of the optimisation has become close to 1 (0.83), indicating no strong over- or under-fitting of the observations.

## 8.2 Posterior uncertainties and bias correction

In this section, we describe two additional experiments that focus on the validation of the two bias-correction methods (Sect. 3), and the posterior uncertainty of the optimised parameters. As a starting point, we take the 10-parameter-state experiment with unperturbed observations from the previous section. We add two additional state parameters related to bias correction (Table 4). We have tested that for this setup, ICLASS is able to retrieve the true parameters well (Table 4, column "Optimised unp."), including the parameters related to bias correction ($Frac_H$ and $s_{CO2}$). We now perturb the "true" observations, by adding to each observation $y_i$ a random number from a normal distribution with mean 0 and standard deviation 1.5 $\sigma_{O,i}$. The factor 1.5 means that we mis-specify the observational errors in the framework, as can happen in real situations as well. In this OSSE we will not simply attempt to fit the observations as good as possible, but we want to employ both observations and prior information. We therefore include the background part of the cost function. The prior error standard deviations are given

in Table 4, the a-priori error covariance matrix was chosen to be diagonal. As we want an estimate of the posterior parameter uncertainty, we ran ICLASS using an ensemble of 100 perturbed members. For simplicity, we take the total observational error equal to the measurement error, as for the previous OSSEs.

From the 101 members in the ensemble, the unperturbed optimisation has the lowest posterior cost function. Figure 4 indicates that the optimised solution shows a strongly improved fit with the (adapted) observations compared to the prior fit. As expected, due to the perturbations on the observations and the background part of the cost function, the true parameters are not found back anymore. For slightly more than half of the state parameters, the optimised parameter value is located less than 1 posterior standard deviation away from the true value (Table 4, rightmost column). The largest relative deviation from the truth is for $w_g$, the posterior value is approximately 2 standard deviations away from the true value. The parameter uncertainty, as expressed by the standard deviations of the state parameters in Table 4, is reduced for every parameter after the optimisation. In addition, the posterior root mean squared error (RMSE) is reduced for every observation stream after optimisation (Table 5). We see that the posterior RMSE now approaches 1.5 times $\sigma_O$, which can be explained by the intentional mis-specification of errors. For the same reason, the values of the partial reduced chi-squared statistics ($\chi^2_{r,j}$) should be near 2.25 (1.5 squared), for all observation streams except $CO2_{20}$. The latter is more complex due to the presence of the bias-correction scaling factor ($s_{CO2}$). Indeed, we find $\chi^2_{r,j}$ values ranging between 2.09 and 2.17 for all streams except $CO2_{20}$ (Table 5). Note that for the analysis of the posterior uncertainty, we have used only members with a posterior $\chi^2_r$ equal or lower than 4.5, only 38 of the 100 perturbed members were therefore used. We conclude here that ICLASS performs well in all our tests with artificial data, we present an application with real observations in the next section (more extensive).

# 9 Application example: CO$_2$ and boundary layer meteorology at a Dutch grassland

## 9.1 Setup

For this example of how the framework can be used, we used data obtained at Cabauw, the Netherlands from 25 September 2003. A 213 m high measurement tower is present at this location (Bosveld et al., 2020). This day is chosen here since it has been used in several studies before (Vilà-Guerau De Arellano et al., 2012; Casso-Torralba et al., 2008) and we consider it as a "golden day" (Sect. 1), for which the CLASS model is expected to perform at its best.

The set of observation streams we used is given in Table 6, and a description of these streams is given in Table A2. Some of the observations we used are not directly part of the Cabauw dataset, we have derived them from other observations in the same dataset. The parameters we optimise (the state) are given in Table 7, and a description of these parameters can be found in Table A1. The model settings, prior state and non-state parameters are to a large extent based on Vilà-Guerau De Arellano et al. (2012). The modelled period is from 9–15 UTC, and we activated the surface layer option in the model. We ran ICLASS in ensemble mode with 175 members, an ensemble member was considered successful when having a final $\chi^2_r <= 2.0$ (Eq. 19). The prior error standard deviations are given in Table 7, the a-priori error covariance matrix was assumed to be diagonal. The representation errors were set to 0, the chosen weights and model and measurement error standard deviations are given

**Table 4.** Parameter values for the bias-correction OSSEs. The used observation streams are $q$, $h$, $H$, $LE$, $T_2$, $F_{CO2}$ and $CO2_{20}$. "$\sigma_A$" and "$\sigma_{\text{post}}$" are respectively the prior and posterior standard deviation of the state parameters. The rightmost column lists the optimised parameter value minus the true value, normalised with the posterior standard deviation of the respective parameter. Column "Optimised unp." lists the optimised parameter values of the experiment without perturbed observations. The four columns at the right-hand side only apply to the OSSE with perturbed observations. See Table A1 for a description of the parameters, and Table A2 for a description of the observation streams.

| Parameter | True | Prior | Optimised unp. | Optimised | $\sigma_A$ | $\sigma_{\text{post}}$ | $\frac{\text{opt. - true}}{\sigma_{\text{post}}}$ |
|---|---|---|---|---|---|---|---|
| $h$ (m) | 350.0 | 650.0 | 350.0 | 379.1 | 200.0 | 53.3 | 0.545 |
| $\alpha_{\text{rad}}$ $(-)$ | 0.200 | 0.450 | 0.200 | 0.188 | 0.150 | 0.010 | -1.209 |
| $\alpha_{\text{sto}}$ $(-)$ | 1.00 | 0.50 | 1.00 | 1.12 | 1.50 | 0.12 | 0.928 |
| $w_g$ $(-)$ | 0.270 | 0.140 | 0.270 | 0.101 | 0.150 | 0.084 | -2.006 |
| $\gamma_\theta$ (K m$^{-1}$) | $3.0 \times 10^{-3}$ | $5.0 \times 10^{-3}$ | $3.0 \times 10^{-3}$ | $3.1 \times 10^{-3}$ | $3.0 \times 10^{-3}$ | $0.3 \times 10^{-3}$ | 0.198 |
| $\gamma_q$ (kg kg$^{-1}$m$^{-1}$) | $-1.0 \times 10^{-6}$ | $-3.0 \times 10^{-6}$ | $-1.0 \times 10^{-6}$ | $-0.8 \times 10^{-6}$ | $3.0 \times 10^{-6}$ | $0.5 \times 10^{-6}$ | 0.362 |
| $CO_2$ (ppm) | 422.0 | 380.0 | 421.9 | 357.8 | 50.0 | 38.3 | -1.675 |
| $adv_{CO2}$ (ppm s$^{-1}$) | 0.0 | $6.0 \times 10^{-3}$ | $4.3 \times 10^{-7}$ | $0.4 \times 10^{-3}$ | $5.6 \times 10^{-3}$ | $0.2 \times 10^{-3}$ | 1.768 |
| $z_{0m}$ (m) | 0.020 | 0.100 | 0.020 | 0.081 | 0.100 | 0.074 | 0.821 |
| $z_{0h}$ (m) | 0.020 | 0.010 | 0.020 | 0.013 | 0.010 | 0.008 | -0.947 |
| $Frac_H$ $(-)$ | 0.350 | 0.600 | 0.350 | 0.392 | 0.200 | 0.087 | 0.479 |
| $s_{CO2}$ $(-)$ | 1.400 | 1.000 | 1.400 | 1.179 | 0.300 | 0.132 | -1.670 |

**Table 5.** The prior and posterior root mean squared error (RMSE), the specified measurement error standard deviation ($\sigma_I$) and the partial reduced chi-squared statistic ($\chi^2_{r,j}$) for the observation streams in the bias-correction OSSE with perturbed observations. The RMSE and $\chi^2_{r,j}$ values are for the member with the lowest posterior cost function, i.e. the member with unperturbed prior. Note that the measurement error equals the observational error here, since the model and representation errors were set to 0 for simplicity. All weights for the cost function were set to 1. We used a total of 280 individual (artificial) observations in the cost function. See Table A2 for a description of the observation streams.

| Observation stream | RMSE prior | RMSE optimised | $\sigma_I$ | $\chi^2_{r,j}$ |
|---|---|---|---|---|
| $q$ (kg kg$^{-1}$) | $6.1 \times 10^{-4}$ | $5.8 \times 10^{-4}$ | $4.0 \times 10^{-4}$ | 2.11 |
| $h$ (m) | 415.6 | 147.2 | 100.0 | 2.17 |
| $H$ (W m$^{-2}$) | 140.4 | 36.5 | 25.0 | 2.13 |
| $LE$ (W m$^{-2}$) | 101.9 | 36.4 | 25.0 | 2.12 |
| $T_2$ (K) | 1.82 | 0.94 | 0.65 | 2.09 |
| $F_{CO2}$ (mg CO$_2$ m$^{-2}$s$^{-1}$) | $40.7 \times 10^{-2}$ | $5.8 \times 10^{-2}$ | $4.0 \times 10^{-2}$ | 2.12 |
| $CO2_{20}$ (ppm) | 127.9 | 3.5 | 2.0 | 3.10 |

in the supplementary material, further detailed settings of the optimisation can be found via the Zenodo link in the "Code and data availability" section.

## 9.2 Computational costs

To give an idea of the computational costs involved, we added a timer to an optimisation. An optimisation like the one described in this section, but without the use of an ensemble, took in our specific case about 1000 times the time of an individual model simulation using the original CLASS model. However, in our test an individual simulation with this model took less than 1 second using a single CPU, so the computational cost is still relatively small. The total time it takes to perform an optimisation is dependent on how well the optimisation converges and on the configuration. Using an ensemble increases the computational

costs, but multiple ensemble members can be run in parallel on multiprocessor computing systems. An optimisation with an ensemble, like the one described in this section took in our specific case approximately 2 hours, using 32 cores of a high-performance computing cluster. The total summed CPU-time was about 45 hours.

## 9.3 Model fit

Figure 5 shows that the optimised run shows a much better fit than the prior run to the subset of observations present in this

figure. For temperature, a total of seven observation heights are included, but we only show three for brevity. The cost function is reduced from 4702 to 126. The $\chi_r^2$ goodness of fit statistic has a value of 0.80 for the optimised run, indicating a slight overfitting (or non-optimal error specifications). For all observation streams, the root mean squared error (Sect. 5, Table 6) and the absolute value of the mean bias error (Eq. 23, not shown) are reduced after optimisation. The ratio of model and observation variance becomes closer to 1 for the vast majority of observation streams, except for three of the specific humidity streams and

the sensible heat flux.

## 9.4 Bias Correction

Let us now turn to the energy balance (Fig. 6). As described in Sect. 3.3, we forced the energy balance in the observations to close, by partitioning the energy balance gap partly to $H$, and partly to $LE$. From Fig. 6, it is clear that in the original observations, the sum of $H$ and $LE$ is generally lower than in both the prior and posterior model runs. Both the model and the

590 "corrected" observations take the energy balance into account. When the difference in $R_n - G$ between model and observations is small (Eq. 7), we might normally expect the model to more easily fit the corrected observations than the original ones. Also in this example, the posterior fit is improved compared to the prior, especially for the sensible heat flux (Table 6). We notice that most of the additional energy is partitioned to $LE$ (Table 7 and Fig. 6). The energy balance closure residual (LHS Eq. 7) is important to account for in this case, the mean value of the residuals is only 11 % smaller as the mean of the measured (without

applying Eq. 8) sensible heat flux. Note that for some of the data points around noon, the energy-balance correction tends to decrease the heat fluxes (Fig. 6). Inspection of a satellite image of that day revealed that this is likely caused by the presence of

high clouds, causing a fast drop in net radiation. The measured heat fluxes tend to react slower however, leading to a negative value for $\varepsilon_{\text{eb}}$ in Eq. (7).

As is the case for the surface energy balance closure problem, there can also be biases in the $CO_2$ flux eddy-covariance observations (Liu et al., 2006). Here we have neglected this for simplicity.

## 9.5 Optimised parameters

The optimised state is shown in Table 7. Advection of heat remains relatively close to 0 after optimisation ($-\,0.13 \pm 0.06$ $K\,h^{-1}$, where 0.06 is $\sigma_{\text{post}}$). This is slightly outside the range of 0.1–0.3 $K\,h^{-1}$ found by Casso-Torralba et al. (2008), who analysed the same day (using a longer time period). The result can be considered in agreement with Bosveld et al. (2004), who concluded large scale heat advection to be negligible for this day. For $CO_2$ however, we find an advection of $5.8 \pm 1.9$ $ppm\,h^{-1}$, which is higher than what was found by Casso-Torralba et al. (2008, their Fig. 9). We shortly return to this in Sect. 9.6. The two parameters that deviate the most from their prior values (normalised with the prior standard deviation, Table 7) are $\gamma_{CO2}$ and $R_{10}$. Both parameters are linked to the $CO_2$ budget, $\gamma_{CO2}$ influences the amount of entrained $CO_2$ from the free troposphere, while $R_{10}$ influences the amount of $CO_2$ entering the mixed layer via respiration. From Fig. 5c, it is clear that the prior run has a way too strong net surface $CO_2$ flux compared to the observations, which explains why $R_{10}$ is strongly increased in the optimised state. Two main pathways in which the $CO_2$ flux can decrease (in magnitude) is either by increasing $R_{10}$ or by decreasing the conductance (via $\alpha_{\text{sto}}$). However, the latter also impacts the model fit of the latent heat flux. Unfortunately, separate measurements of GPP and respiration (e.g. derived from carbonyl sulfide observations, Whelan et al., 2018) are not available in our dataset. These could help to further constrain the $CO_2$-related parameters. In a more detailed study, the parameter estimates might still be improved by e.g. better estimating vector $\boldsymbol{p}$ and the observational errors.

## 9.6 Posterior uncertainty and correlations

The discussion above illustrates that not all of the parameters in the state may be assumed to be fully independent. To analyse the posterior correlations, we have constructed a correlation matrix (Sect. 4.2), shown in Fig. 7. From the 174 perturbed ensemble members, 150 were successful, and were used for the calculation of statistics. As expected, the correlation between $\alpha_{\text{sto}}$ and $R_{10}$ is very strongly positive (0.94) and this can likely be explained by their opposite effects on the $CO_2$ flux, as explained in the previous paragraph. Some of the correlations between parameters can be relatively complex, e.g. a correlation between two posterior parameters might involve a third parameter that correlates with both.

Coming back to the discrepancy in $CO_2$ advection between our analysis and what was found by Casso-Torralba et al. (2008), it can be noted that the $adv_{CO2}$ parameter is relatively strongly correlated with both the $\Delta_{CO2}$ and $\gamma_{CO2}$ parameters (Fig. 7: corr. = –0.65 and –0.80 respectively). This can indicate that entrainment from the free troposphere is hard to disentangle from advection in shaping the $CO_2$ budget, with the current set of observations we incorporate. Differences in how entrainment is handled by Casso-Torralba et al. (2008) might explain part of the difference in estimated $CO_2$ advection.

To check to what extent the obtained correlations are robust and independent of the selected number of ensemble members, we reconstructed the correlation matrix using only half of the successful perturbed members (75 of the 150). The average

absolute value of difference between the non-diagonal matrix entries when using the subsample and the non-diagonal matrix entries when using the full successful perturbed ensemble amounts to 0.05, with a maximum of 0.23 for one entry. This suggests that using 150 members in the correlation analysis leads to a reasonably robust estimate of the posterior correlations.

Another option we explore is to analyse the posterior probability density functions (pdfs) for the successful perturbed members in the ensemble. As an example, we show the pdfs for two parameters, $adv_\theta$ and $\gamma_q$ (Fig. 8). For $adv_\theta$, the posterior uncertainty is clearly reduced compared to the prior, as the posterior pdf is markedly narrower. For $\gamma_q$ (the free-tropospheric specific-humidity lapse rate) however, there is no clear reduction in uncertainty. The wide posterior pdf implies that similar results can be obtained over a relatively wide range of $\gamma_q$, possibly by perturbing other parameters with a similar effect. To further constrain this parameter, more specific observation streams would need to be added, possibly from radiosondes. The use of ICLASS in e.g. the planning of observational campaigns can therefore help to determine beforehand what type of observations are needed to better constrain the processes represented in the model. This is done through the use of observation system simulation experiments, similar to e.g. Ye et al. (2022).

## 10   Concluding discussion

We have presented here a description of ICLASS, a variational Inverse modelling framework for the Chemistry Land-surface Atmosphere Soil Slab model. This framework serves as a tool to study the atmospheric boundary layer and/or land–atmosphere exchange. It avoids the need of manual trial-and-error in choosing parameter values for the model when fitting observations, thereby providing more objectivity. Some extent of subjectivity however remains, as the proper specification of errors is not always simple (e.g. Rödenbeck et al., 2003). The use of more advanced error estimation methods can mitigate this. The very non-linear model around which the framework is built, makes the optimisation challenging. In ICLASS, two main ways in which this challenge is tackled are:

– The use of an analytical gradient of the cost function, involving the model adjoint, allowing for more precise gradient calculations

– The possibility of running ICLASS in a Monte-Carlo way. This involves perturbing the prior state vector and the (scaled) observations. When a single optimisation does not converge, the use of an ensemble can provide a solution.

The latter way of running ICLASS also has the advantage that posterior error statistics can be obtained, which is of paramount importance in inverse modelling.

The model is relatively simple, yet contains the physics to model the essentials of the convective boundary layer and of land surface–atmosphere exchange. Its simplicity however means that the model does not perform well in every situation, the best performance can be expected for days when the boundary layer resembles a prototypical convective boundary layer. We have shown the usefulness of ICLASS by applying it to a "golden day" at a Dutch grassland site where extensive data is available. The fit to several streams of observations simultaneously was greatly improved in the posterior compared to the prior. We have to keep in mind, however, that we cannot expect a relatively simple model to capture all small-scale processes playing a role in

the convective boundary layer and in land surface–atmosphere exchange (e.g. heterogeneous surface heating and evaporation, influence of individual thermals, ...).

Key strengths of this framework are that observations from several information streams can be used simultaneously, and
665 surface layer profiles can be taken into account. The framework allows to integrate knowledge from ecosystem-level studies (fluxes) and global studies (mixing ratios). It can be seen as a tool that maximises the use of available observational data at CBL/ecosystem level (e.g. along tall towers like the Cabauw tower). Observation system simulation experiments using ICLASS can also help in the planning of observational campaigns, to determine in advance which observation streams are needed to better constrain model processes. Another feature of the framework is the capacity of correcting observations for
biases. A specific bias-correction system for the energy balance closure problem is implemented. The energy balance residual was shown to be substantial at the Dutch grassland site in our application example. Correcting for biases is critical in inverse modelling, to prevent bias errors to propagate in parameter estimates. The correction of biases is however a very complex topic. There are limitations to the level of complexity that our bias-correction methods can handle, ICLASS cannot be expected to deal completely with all bias issues.

ICLASS is computationally relatively cheap to run and can be extended in the future by e.g. incorporating a more detailed representation of the vegetation. This extension can further improve the capabilities to fit sets of observations at locations with a more complex vegetation structure.

*Code and data availability.* The code is hosted at GitHub, the current version of ICLASS is available from https://github.com/PBosmanatm/ICLASS under a GNU General Public License, v3.0 or later. The adapted forward model is also included via the GitHub link, as well as the ICLASS
manual. The GitHub repository is linked to Zenodo, which provides DOIs for released software. The release on which this reference paper is based can be found at https://doi.org/10.5281/zenodo.7239147 (Bosman and Krol, 2022) (GNU General Public License, v3.0 or later). The code and the data used for creating the plots with optimisation results in this paper, as well as for the contents of the tables, are part of the downloadable material. The data used in this paper can somewhat differ from the most recent version of these data. Boundary layer height data were provided by Henk Klein Baltink (KNMI). This data (and all other used data) can be found via the Zenodo link. The newest version
of the Cabauw data (except boundary-layer heights) can be found at the following locations: The $CO_2$ mixing ratios can be found at the ICOS (https://www.icos-cp.eu/data-products/ERE9-9D85) and ObsPack (https://gml.noaa.gov/ccgg/obspack/) websites. Temperature, heat fluxes etc. can be found at https://dataplatform.knmi.nl/dataset/?tags=Insitu&tags=CESAR.

## Appendix A: Description of used parameters and observation streams

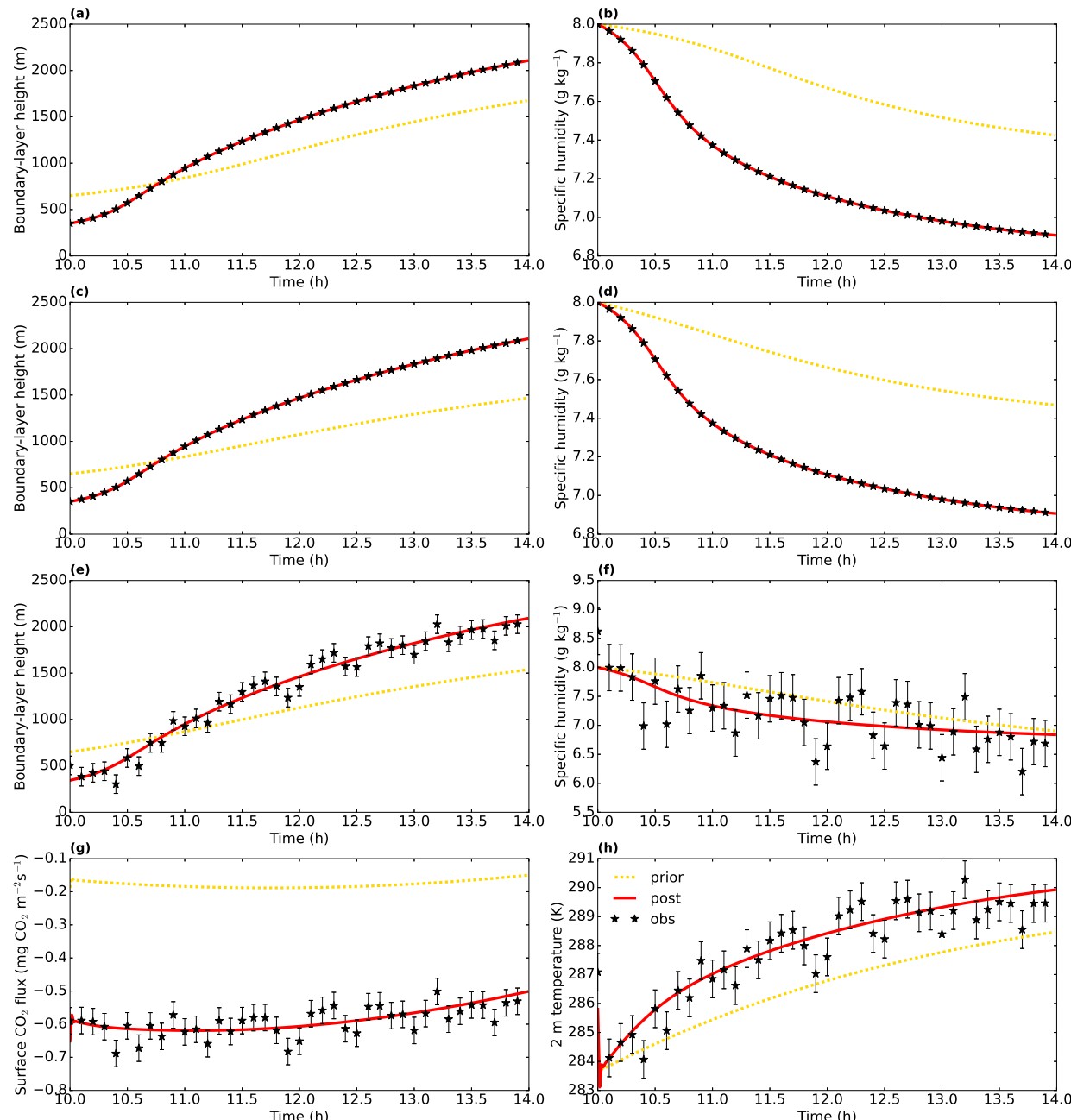

**Figure 3.** Model output from three observation system simulation experiments from Sect. 8.1: 2-parameter state experiment with albedo and initial boundary-layer height (top row), 5-parameter state experiment with two observation streams (second row) and experiment with perturbed observations (third and fourth rows). "post" means posterior, i.e. the optimised simulation. Even though the posterior lines look very similar for the first and second rows, the model parameters are not identical. The specific humidity is from the mixed layer. The error bars show the observational error standard deviations $\sigma_O$, here equal to $\sigma_I$.

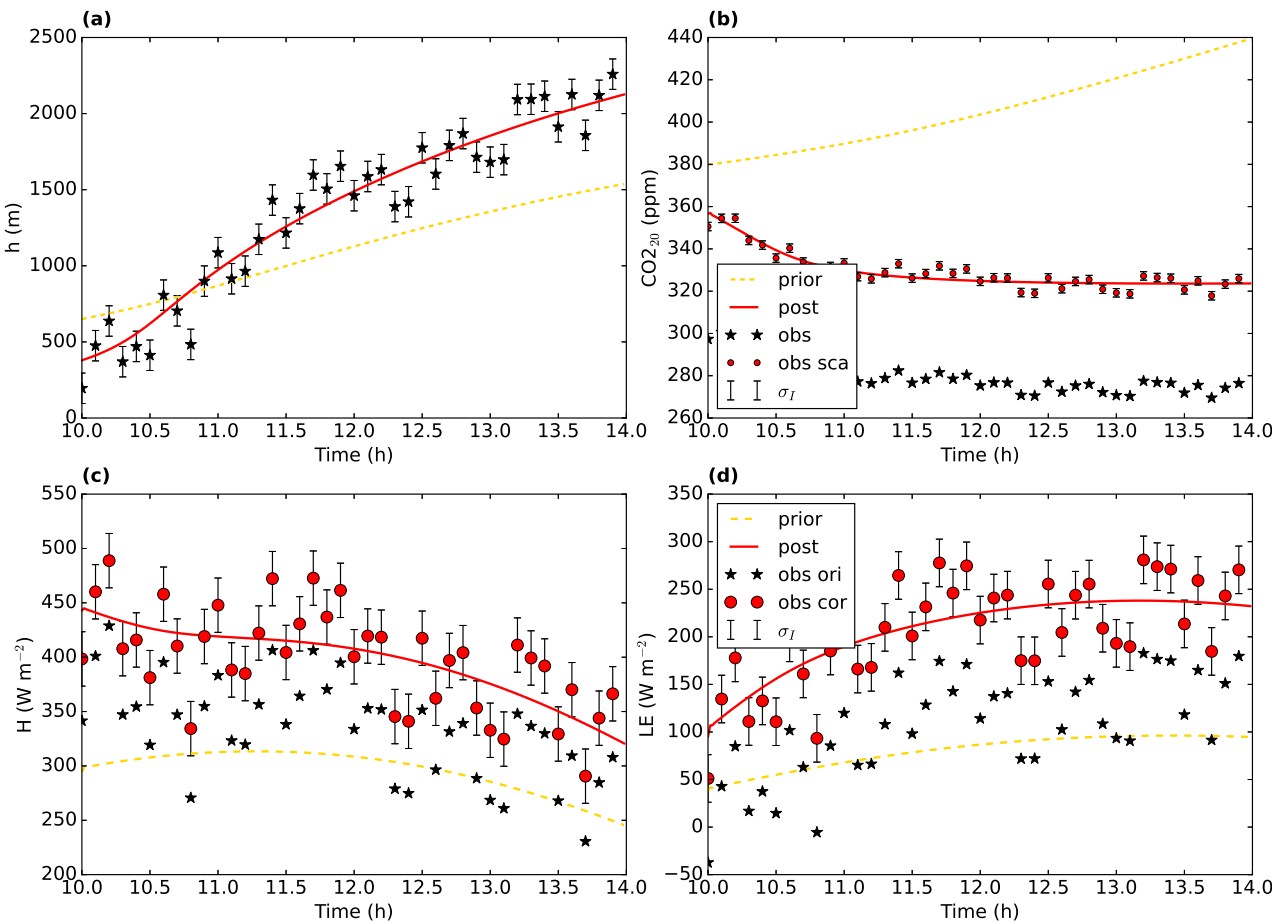

**Figure 4.** Model output and observations for the bias-correction OSSE with perturbed observations. The full red line shows the optimised model ("post" meaning posterior), the dashed yellow line is the prior model. The (uncorrected) observations are shown by a black star, the error bars show the measurement error standard deviation $\sigma_I$ (here equal to $\sigma_O$) on the observations. In (b), the red-filled circles are the scaled observations, i.e. $s_{CO2} \times \boldsymbol{y}$. In (c) and (d), the black stars are the original (although perturbed) observations, the red dots the observations after applying Eq. 8 and 9.

**Table 6.** Observation fit statistics for the Cabauw optimisation. The left columns show the root mean squared error (RMSE) for the prior and posterior state. The two right columns show the ratio between model and observation variance for the prior and posterior state. Only model data points for times at which we have an observation available are taken into account. We used a total of 670 individual observations in the cost function. The chosen cost-function weights and model and measurement error standard deviations are given in the supplementary material. The sensible and latent heat flux observations used are corrected for the energy balance gap, see Eq. (8) and Eq. (9). The observation streams are described in Table A2.

| Observation stream | RMSE prior | RMSE post. | Var. ratio prior | Var. ratio post. |
|---|---|---|---|---|
| $T_{200}$ (K) | 0.828 | 0.286 | 1.456 | 0.763 |
| $T_{140}$ (K) | 0.868 | 0.296 | 1.449 | 0.759 |
| $T_{80}$ (K) | 0.751 | 0.259 | 1.422 | 0.753 |
| $T_{40}$ (K) | 0.741 | 0.242 | 1.426 | 0.773 |
| $T_{20}$ (K) | 0.827 | 0.294 | 1.440 | 0.786 |
| $T_{10}$ (K) | 0.781 | 0.282 | 1.434 | 0.789 |
| $T_2{}^1$ (K) | 0.815 | 0.307 | 1.505 | 0.842 |
| $q_{200}$ (kg kg$^{-1}$) | $9.22 \times 10^{-4}$ | $1.24 \times 10^{-4}$ | 0.353 | 0.753 |
| $q_{140}$ (kg kg$^{-1}$) | $9.59 \times 10^{-4}$ | $1.24 \times 10^{-4}$ | 0.332 | 0.708 |
| $q_{80}$ (kg kg$^{-1}$) | $9.49 \times 10^{-4}$ | $1.33 \times 10^{-4}$ | 0.328 | 0.623 |
| $q_{40}$ (kg kg$^{-1}$) | $8.70 \times 10^{-4}$ | $1.71 \times 10^{-4}$ | 0.351 | 0.488 |
| $q_{20}$ (kg kg$^{-1}$) | $8.87 \times 10^{-4}$ | $1.74 \times 10^{-4}$ | 0.434 | 0.406 |
| $q_{10}$ (kg kg$^{-1}$) | $9.16 \times 10^{-4}$ | $1.67 \times 10^{-4}$ | 0.523 | 0.383 |
| $q_2{}^1$ (kg kg$^{-1}$) | $8.90 \times 10^{-4}$ | $2.55 \times 10^{-4}$ | 0.514 | 0.219 |
| $CO2_{207}$ (ppm) | 19.693 | 0.712 | 5.728 | 0.866 |
| $CO2_{127}$ (ppm) | 21.528 | 2.778 | 3.754 | 0.568 |
| $CO2_{67}$ (ppm) | 20.774 | 2.088 | 3.840 | 0.578 |
| $CO2_{27}$ (ppm) | 19.506 | 1.342 | 4.548 | 0.660 |
| $h$ (m) | 270.5 | 131.5 | 0.524 | 1.107 |
| $LE$ (W m$^{-2}$) | 22.28 | 18.47 | 0.691 | 0.986 |
| $H$ (W m$^{-2}$) | 33.41 | 13.05 | 0.997 | 1.316 |
| $F_{CO2}$ (mg $CO_2$ m$^{-2}$ s$^{-1}$) | 0.494 | 0.102 | 0.342 | 0.514 |
| $S_{out}$ (W m$^{-2}$) | 18.83 | 10.21 | 2.507 | 2.095 |

[1]The 2 m temperature and 2 m dewpoint observations (the latter used for deriving $q_2$) were actually taken at 1.5 m. We make our calculations as if the observations were taken at 2 m. This rounding might have introduced some bias to the $T_2$ and $q_2$ observations. We tested the impact of this by performing a corrected optimisation (without ensemble). The state parameters resulting from this optimisation were all within the 2 standard deviation posterior uncertainty range of the uncorrected optimisation ($x \pm 2\sigma_{post}$), and 9 out of 14 parameters where within the 1 standard deviation uncertainty range ($x \pm \sigma_{post}$).

**Table 7.** State parameters for the Cabauw optimisation. $\delta_{\mathrm{nor}}$ is the normalised deviation, i.e. the posterior minus the prior, normalised with the square root of the prior variance (Eq. 24). "$\sigma_A$" and "$\sigma_{\mathrm{post}}$" are respectively the prior and posterior standard deviation of the state parameters.

| Parameter | Prior | Posterior | $\delta_{\mathrm{nor}}$ | $\sigma_A$ | $\sigma_{\mathrm{post}}$ |
|---|---|---|---|---|---|
| $adv_\theta$ $(\mathrm{K\,s^{-1}})$ | 0.000 | $-3.684 \times 10^{-5}$ | $-6.632 \times 10^{-2}$ | $5.556 \times 10^{-4}$ | $1.731 \times 10^{-5}$ |
| $adv_q$ $(\mathrm{kg\,kg^{-1}s^{-1}})$ | 0.000 | $1.086 \times 10^{-8}$ | $1.955 \times 10^{-2}$ | $5.556 \times 10^{-7}$ | $4.415 \times 10^{-8}$ |
| $adv_{CO2}$ $(\mathrm{ppm\,s^{-1}})$ | 0.000 | $1.602 \times 10^{-3}$ | 0.3845 | $4.167 \times 10^{-3}$ | $5.204 \times 10^{-4}$ |
| $\Delta_\theta$ $(\mathrm{K})$ | 4.200 | 3.121 | -0.7191 | 1.500 | 0.486 |
| $\gamma_\theta$ $(\mathrm{K\,m^{-1}})$ | $3.600 \times 10^{-3}$ | $2.849 \times 10^{-3}$ | -0.2502 | $3.00 \times 10^{-3}$ | $6.10 \times 10^{-4}$ |
| $\Delta_q$ $(\mathrm{kg\,kg^{-1}})$ | $-8.000 \times 10^{-4}$ | $-2.138 \times 10^{-3}$ | -0.6692 | $2.00 \times 10^{-3}$ | $5.59 \times 10^{-4}$ |
| $\gamma_q$ $(\mathrm{kg\,kg^{-1}\,m^{-1}})$ | $-1.200 \times 10^{-6}$ | $-1.207 \times 10^{-7}$ | 0.5397 | $2.0000 \times 10^{-6}$ | $1.8409 \times 10^{-6}$ |
| $\Delta_{CO2}$ $(\mathrm{ppm})$ | -44.00 | -22.42 | 0.8632 | 25.00 | 7.00 |
| $\gamma_{CO2}$ $(\mathrm{ppm\,m^{-1}})$ | 0.000 | $-6.389 \times 10^{-2}$ | -2.130 | $3.000 \times 10^{-2}$ | $2.040 \times 10^{-2}$ |
| $\alpha_{\mathrm{sto}}$ $(-)$ | 1.000 | 0.8990 | -0.4038 | 0.2500 | 0.0640 |
| $\alpha_{\mathrm{rad}}$ $(-)$ | 0.2500 | 0.2285 | -0.2146 | 0.1000 | 0.0014 |
| $\mathrm{Frac}_H$ $(-)$ | 0.6000 | 0.3474 | -0.8420 | 0.3000 | 0.0391 |
| $w_g$ $(-)$ | 0.4800 | 0.5153 | 0.2355 | 0.1500 | 0.0667 |
| $R_{10}$ $(\mathrm{mg\,CO_2\,m^{-2}\,s^{-1}})$ | 0.2300 | 0.6393 | 1.364 | 0.3000 | 0.0346 |

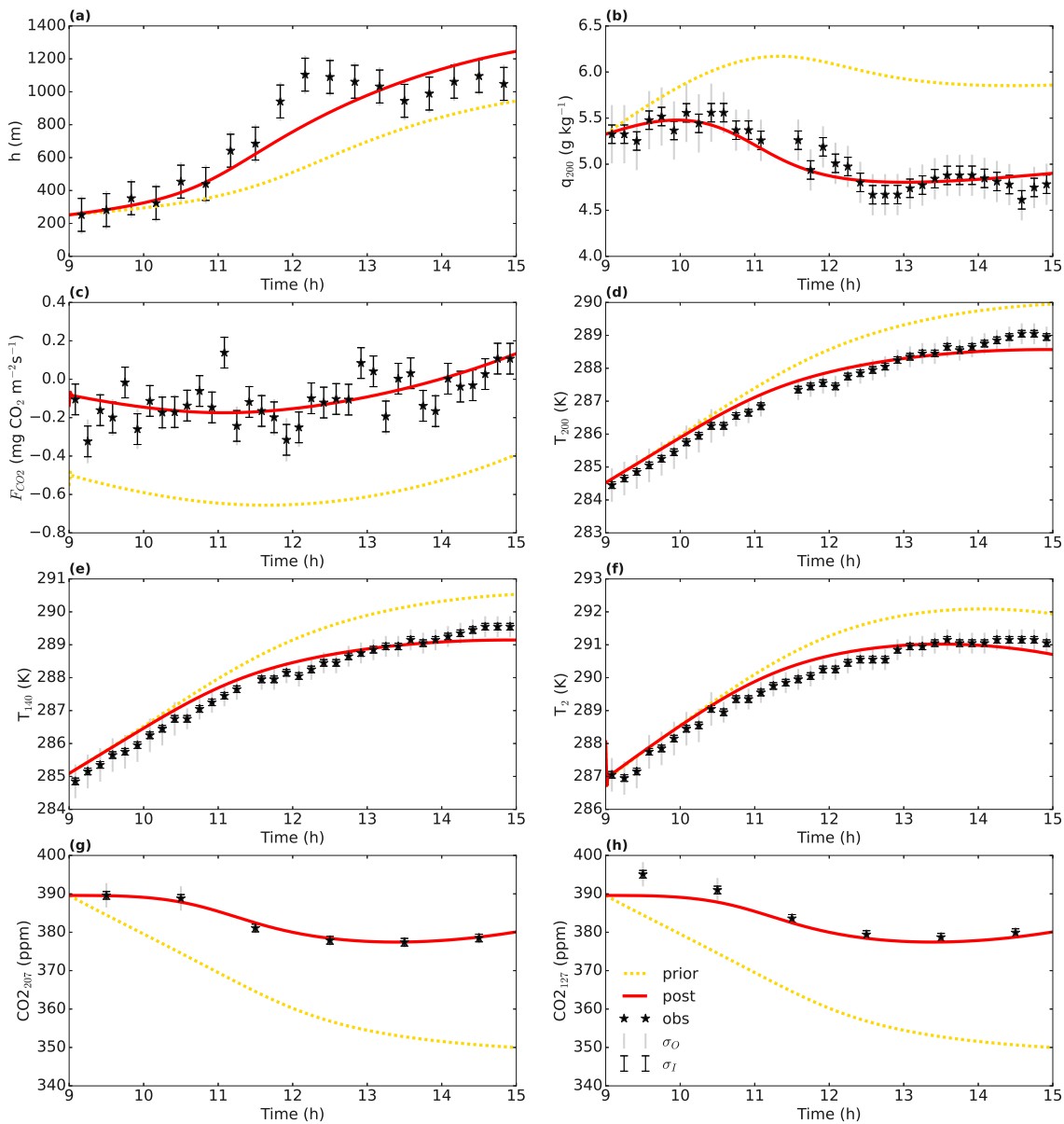

**Figure 5.** Optimisation for the Cabauw data. The full red line shows the optimised model ("post" meaning posterior), the dotted yellow line is the prior model. The observations are shown by a black star, the error bars show the measurement error standard deviation $\sigma_I$ and total error standard deviation $\sigma_O$ on the observations.

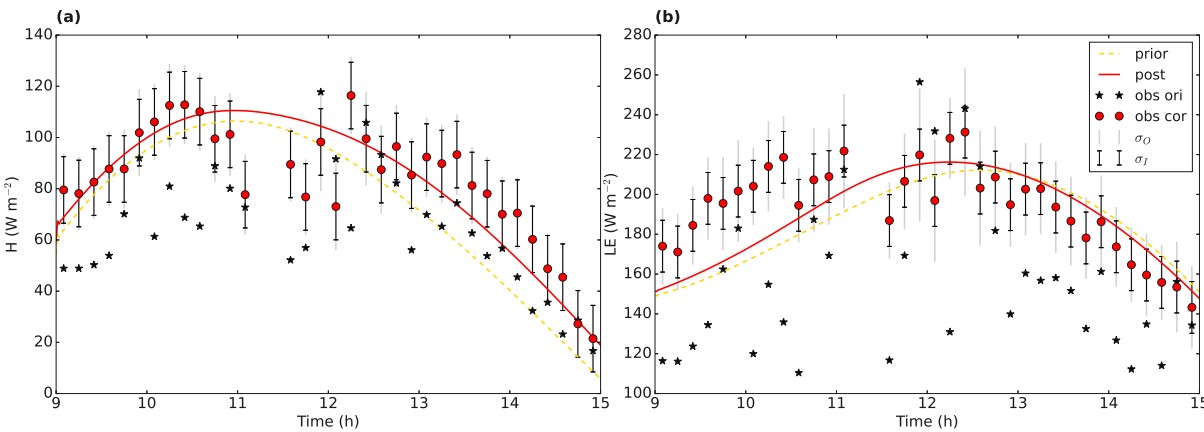

**Figure 6.** Energy balance data (left: sensible heat flux, right: latent heat flux) for the Cabauw case. The black stars are the original observations, the red dots the observations after forced closure of the energy balance. The error bars indicate the measurement error standard deviation $\sigma_I$ and the total observational error standard deviation $\sigma_O$. The full red line shows the optimised model ("post" meaning posterior), the dashed yellow line is the prior model.

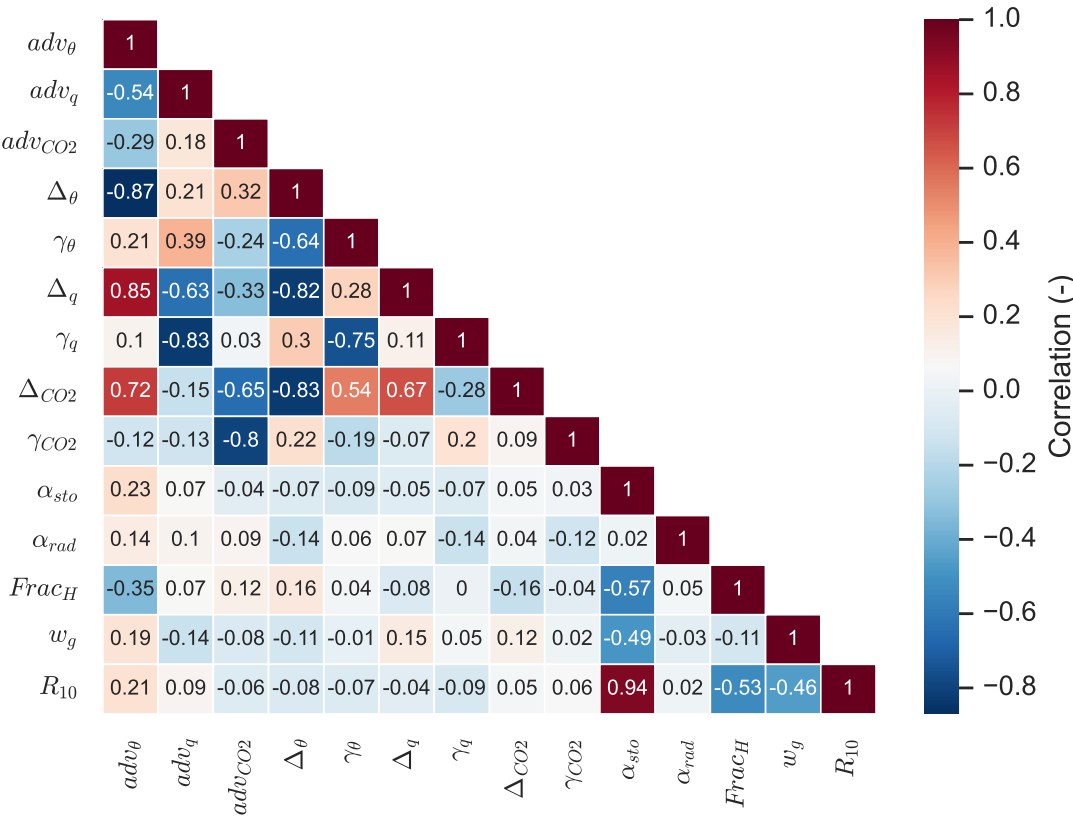

**Figure 7.** Correlation matrix for the optimised state parameters for the Cabauw case. The shown correlations are marginal correlations, not partial correlations.

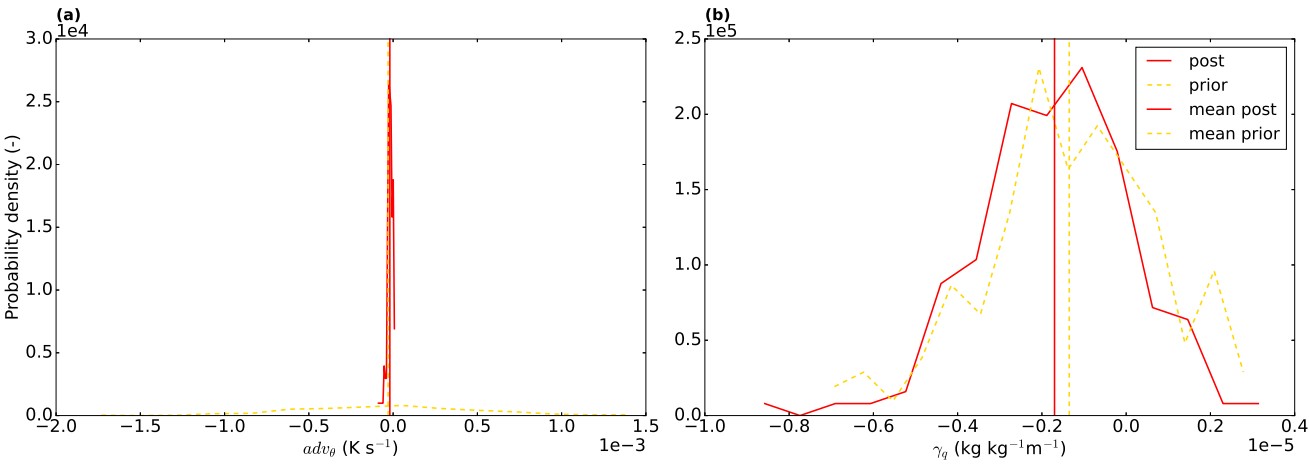

**Figure 8.** Probability density functions for the $adv_\theta$ (left) and $\gamma_q$ (right) parameters, the red full line is for the posterior and the yellow dashed line for the prior distribution. The vertical lines represent the mean of the distributions. 15 bins were used. Note that the prior distribution is determined from the sample of priors, which has a component of randomness.

**Table A1.** Used parameters in this paper.

| Name | Name in code | Description | Units |
|---|---|---|---|
| $adv_{CO2}$ | advCO2 | Advection of $CO_2$ | $\mathrm{ppm\,s^{-1}}$ |
| $adv_q$ | advq | Advection of moisture | $\mathrm{kg\,kg^{-1}s^{-1}}$ |
| $adv_\theta$ | advtheta | Advection of heat | $\mathrm{K\,s^{-1}}$ |
| $\alpha_{\mathrm{sto}}$ | sca_sto | Scaling factor for stomatal conductance | – |
| $\alpha_{\mathrm{rad}}$ | alpha | Surface albedo | – |
| $\Delta_{CO2}$ | deltaCO2 | Initial $CO_2$ jump at $h$ | ppm |
| $\Delta_q$ | deltaq | Initial specific humidity jump at $h$ | $\mathrm{kg\,kg^{-1}}$ |
| $\Delta_\theta$ | deltatheta | Initial temperature jump at $h$ | K |
| $\mathrm{Frac}_H$ | FracH | Fraction of energy balance gap partitioned to $H$ obs | – |
| $\gamma_{CO2}$ | gammaCO2 | Free atmosphere $CO_2$ lapse rate | $\mathrm{ppm\,m^{-1}}$ |
| $\gamma_q$ | gammaq | Free atmosphere specific humidity lapse rate | $\mathrm{kg\,kg^{-1}\,m^{-1}}$ |
| $\gamma_\theta$ | gammatheta | Free atmosphere potential temperature lapse rate | $\mathrm{K\,m^{-1}}$ |
| $h$ | h | Initial atmospheric boundary-layer height | m |
| $R_{10}$ | R10 | Respiration at 10 °C | $\mathrm{mg\,CO_2\,m^{-2}\,s^{-1}}$ |
| $w_g$ | wg | Volumetric water content top soil layer | – |
| $CO_2$ | CO2 | Mixed-layer $CO_2$ | ppm |
| $z_{0m}$ | z0m | Roughness length for momentum | m |
| $z_{0h}$ | z0h | Roughness length for scalars | m |
| $s_{CO2}$ | obs_sca_cf_CO2mh | Scaling factor for $CO2_{20}$ observations (Sect. 8) | – |

**Table A2.** Used observation streams in this paper. Note that variables such as $T_{200}$ and $T_{140}$ are called "Tmh" and "Tmh2" respectively in the model code for Sect. 9, which represents temperatures at the user-specified temperature measuring heights 1 and 2 respectively. For the $CO_2$ mixing ratio, we do not make a distinction in this paper between moist-air and dry-air mixing ratio.

| Name | Name in code | Description | Units |
|---|---|---|---|
| $T_{200}$ | Tmh (Sect. 9) | Temperature measured at 200 m height | K |
| $T_{140}$ | Tmh2 (Sect. 9) | Temperature measured at 140 m height | K |
| | $\vdots$ | | |
| $T_2$ | Tmh7 (Sect. 9), Tmh (Sect. 8) | Temperature measured at 2 m height | K |
| $q_{200}$ | qmh (Sect. 9) | Specific humidity measured at 200 m height | $\mathrm{kg\,kg^{-1}}$ |
| $q_{140}$ | qmh2 (Sect. 9) | Specific humidity measured at 140 m height | $\mathrm{kg\,kg^{-1}}$ |
| | $\vdots$ | | |
| $q_2$ | qmh7 (Sect. 9) | Specific humidity measured at 2 m height | $\mathrm{kg\,kg^{-1}}$ |
| $q$ | q | Mixed-layer specific humidity | $\mathrm{kg\,kg^{-1}}$ |
| $CO2_{207}$ | CO2mh (Sect. 9) | $CO_2$ mixing ratio measured at 207 m height | ppm |
| | $\vdots$ | | |
| $CO2_{27}$ | CO2mh4 (Sect. 9) | $CO_2$ mixing ratio measured at 27 m height | ppm |
| $CO2_{20}$ | CO2mh (Sect. 8) | $CO_2$ mixing ratio measured at 20 m height | ppm |
| $h$ | h | Boundary-layer height | m |
| $H$ | H | Surface sensible heat flux | $\mathrm{W\,m^{-2}}$ |
| $LE$ | LE | Surface latent heat flux | $\mathrm{W\,m^{-2}}$ |
| $F_{CO2}$ | wCO2 | Surface $CO_2$ flux | $\mathrm{mg\,CO_2\,m^{-2}\,s^{-1}}$ |
| $S_{out}$ | Swout | Outgoing shortwave radiation | $\mathrm{W\,m^{-2}}$ |

## Appendix B: Inverse-modelling variables

**Table B1.** Description, dimensions and units (of content) of inverse-modelling variables used in Sect. 3.1 and 3.4 of this paper.

| Variable | Description | Dimensions | Units (of content) |
|---|---|---|---|
| $J$ | Cost function | Scalar | – |
| $n$ | Number of state variables | Scalar | – |
| $\boldsymbol{x}$ | State vector ($i^{\text{th}}$ element indicated by $x_i$) | Vector, length $= n$ | Units of the respective state variables |
| $\boldsymbol{x_b}$ | Bias-parameters part of state vector | Vector, length = number of state variables belonging to bias correction | Units of the respective state variables: dimensionless |
| $\boldsymbol{x_m}$ | Model-parameters part of state vector | Vector, length = number of state variables that are model parameters | Units of the respective state variables |
| $\boldsymbol{x_m^{\{t\}}}$ | Model-parameters part of state vector with "true" parameter values | Vector, length = number of state variables that are model parameters | Units of the respective state variables |
| $\boldsymbol{x_A}$ | A-priori estimate of the state vector | Vector, length $= n$ | Units of the respective state variables |
| $m$ | Number of observations included in cost function | Scalar | – |
| $m_j$ | Number of observations from observation stream $j$ included in cost function | Scalar | – |
| $\boldsymbol{y}$ | Vector of observations included in cost function ($J$) | Vector, length $= m$ | Units of the respective observations |
| $\boldsymbol{y_H}$ | Vector of sensible heat flux observations included in $J$ | Vector, length = number of sensible heat flux observations in $\boldsymbol{y}$ | Units of each element: $\text{W m}^{-2}$ |
| $\boldsymbol{y_{LE}}$ | Vector of latent heat flux observations included in $J$ | Vector, length = number of latent heat flux observations in $\boldsymbol{y}$ | Units of each element: $\text{W m}^{-2}$ |
| $\boldsymbol{p}$ | Vector of model parameters that are not optimised | Vector, length = number of model parameters not included in state | Units of the respective parameters |
| $\boldsymbol{H(x_m, p)}$ | Vector of model output to be compared with observations | Same as vector $\boldsymbol{y}$ | Same units as vector $\boldsymbol{y}$ |
| $(\boldsymbol{\nabla_{x_m} H(x_m, p)})^T$ | Adjoint of model | Matrix, size = length of vector $\boldsymbol{x_m} \times$ length of vector $\boldsymbol{y}$ | Units of element on row $i$ and column $j$: units of $j^{\text{th}}$ element of $\boldsymbol{y}$ divided by units of $i^{\text{th}}$ element of $\boldsymbol{x_m}$ |

| | | | |
|---|---|---|---|
| $\mathbf{S_A}$ | A-priori error covariance matrix | Matrix with size $n \times n$ | Units of element on row $i$ and column $j$: units of $i^{\text{th}}$ state variable multiplied with units of $j^{\text{th}}$ state variable |
| $\mathbf{S_O}$ | Observational error covariance matrix | Matrix with size $m \times m$ (diagonal matrix) | Units of $i^{\text{th}}$ diagonal element: units of $i^{\text{th}}$ observation squared, other elements are 0 |
| $\sigma_{O,i}$ | Observational error standard deviation belonging to $i^{\text{th}}$ observation | Scalar | Unit of respective observation |
| $\sigma_{I,i}$ | Instrument (measurement) error standard deviation belonging to $i^{\text{th}}$ observation | Scalar | Unit of respective observation |
| $\sigma_{M,i}$ | Model error standard deviation for model output corresponding to $i^{\text{th}}$ observation | Scalar | Unit of respective observation |
| $\sigma_{R,i}$ | Representation error standard deviation for model output corresponding to $i^{\text{th}}$ observation | Scalar | Unit of respective observation |
| $s_i$ | Scaling factor for $i^{\text{th}}$ observation (constant within each observation stream) | Scalar | Dimensionless |
| $s_i^{\{t\}}$ | "True" scaling factor for $i^{\text{th}}$ observation (constant within each observation stream) | Scalar | Dimensionless |
| $w_i$ | Weight for $i^{\text{th}}$ observation | Scalar | Dimensionless |
| $\boldsymbol{F}$ | Forcing vector | Vector, length = length of vector $\boldsymbol{y}$ | Units of $i^{\text{th}}$ element: (units of $i^{\text{th}}$ observation)$^{-1}$ |
| $\boldsymbol{F_H}$ | Forcing vector for the sensible heat flux observations | Vector, length = length of vector $\boldsymbol{y_H}$ | Units of each element: $\text{m}^2\text{W}^{-1}$ |
| $\boldsymbol{F_{LE}}$ | Forcing vector for the latent heat flux observations | Vector, length = length of vector $\boldsymbol{y_{LE}}$ | Units of each element: $\text{m}^2\text{W}^{-1}$ |
| $\boldsymbol{\varepsilon}_{\text{eb}}$ | Energy balance closure residual | Vector, length = length of $\boldsymbol{y_H}$ or $\boldsymbol{y_{LE}}$, depending on equation | Units of each element: $\text{W m}^{-2}$ |
| $Frac_H$ | Fraction of energy balance closure residual added to the sensible heat flux | Scalar | – |

**Table B2.** Output variables defined in Sect. 5.

| Variable | Description | Unit |
|---|---|---|
| $\chi_r^2$ | Reduced chi-squared statistic | – |
| $\chi_{r,j}^2$ | Reduced chi-squared statistic for observation stream $j$ | – |
| $\chi_{r,b}^2$ | Reduced chi-squared statistic for background part cost function | – |
| $J_j$ | Partial cost function observation stream $j$ | – |
| $\varepsilon_{\text{bias},j}$ | Mean bias error $j^{\text{th}}$ observation stream | Units of respective observation stream |
| $\delta_{\text{nor},i}$ | normalised deviation of the posterior from the prior for $i^{\text{th}}$ state parameter | – |

*Author contributions.* MCK and PJMB designed the study. PJMB performed the majority of the coding and adjoint model construction, with help from MCK. PJMB performed the numerical optimisations, and wrote the manuscript with help from MCK.

*Competing interests.* The authors declare that they have no conflict of interest.

*Acknowledgements.* This work is part of the COS-OCS project (http://cos-ocs.eu/), a project that received funding from the European Research Council (ERC) under the European Union's Horizon 2020 research and innovation programme under grant agreement No 742798.
Part of this work was carried out on the Dutch national e-infrastructure with the support of SURF Cooperative. We are grateful to Jordi Vilà-Guerau De Arellano, Fred Bosveld, Arnoud Frumau and Henk Klein Baltink for helping us with and/or providing (directly or indirectly) the Cabauw data. We also like to thank the developers of the CLASS model. Thanks also to the three anonymous reviewers for helping us to improve the manuscript. We furthermore thank Chiel van Heerwaarden, Linda Kooijmans and Jordi Vilà-Guerau De Arellano for providing comments on our manuscript earlier.

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
