# Peer review of "ICLASS 1.1, a variational Inverse modelling framework for the Chemistry Land-surface Atmosphere Soil Slab model: description, validation and application"

_Geoscientific Model Development, 2022_

## Referee Comment (RC1)

**Review : ICLASS 1.0: a variational inverse modelling framework for the Chemistry land-surface Atmosphere Soil Slab model: description, validation and application**

**1 GENERAL COMMENTS**

The paper describes a novel variational inverse modelling framework for the one dimension Chemistry Land-surface Atmosphere Soil Slab model CLASS, which consists of a simplified land-surface model coupled to a model of atmospheric boundary layer. First, the paper shows a thorough presentation of the forward model and the associated inverse modelling framework, some illustrations of the adjoint coding and the description of error statistics. The adjoint model is validated technically using the tangent linear test and the adjoint test. Observation System Experiments show the strength of their approach to optimize land surface parameters. Eventually, the potential of ICLASS 1.0 at combining both large scale observations (e.g. CO2 mixing ratios) and local observations (e.g. CO2 flux) to constraint model parameters and to estimate bias in observations is demonstrated through an application for a grassland in the Netherlands.

This work can be expected to contribute significantly to improving our understanding of the terrestrial carbon cycle. Although briefly mentioned, the development of the inverse framework is a necessary step that presumingly aims to constrain the stomatal by combining observations of atmospheric mixing ratio and ecosystem fluxes of Carbonyl Sulfide at a specific location. The overall text is well written, and the authors very carefully present the framework, the experiments and discuss the results. The paper is within the scope of gmd and, I recommend its publication with minor amendments. Details of my comments will be found in the following.

**2 SPECIFIC COMMENTS**

Page 1, line 6: Replace "enables to estimate" by "enables the estimation of information".

Page 7, line 7: "free-tropospheric mixing ratios". I disagree, free tropospheric mixing ratios are not difficult to obtain by observations. There are numerous surface sites measuring greenhouse gases concentrations around the globe.

Page 1, line 19: Add an s after exchange.

Page 1, line 21; Strictly speaking, the second part of the sentence (the well known atm...) is false. The atmospheric boundary layer exists even though the daytime conditions are not sunny.

Page 2, line 30: Add after scalars (e.g. wind speed and temperature).

Page 2, line 30: For which time scale and horizontal resolution these assumptions are valid?

Page 2, line 35: Parenthesis within parenthesis. Use "for instance at Cabaw"..

Page 2, line 40: Here, you can mention the problems of equifinallity (Tang et al., 2008) and overfitting.

Page 2, line 42: Replace Inital by Initial.

Page 2, line 42: Replace e.g. by for instance.

5 Page 3, line 66: This is also illustrated in Ziehn et al. (2012) with the assimilation of atmospheric CO2 data in BETHY LSM.

Page 3, line 62: Above all, it is the iterative process that allows to find the local miminum of the cost function in case of linearity.

Page 3, line 65: The choice of using variational methods compared to other technics dealing with the non linearity (e.g. Particule filters) could be discussed here. The advantages of using an adjoint compared to a numerically computed gradient could

10 be also added. For instance, the adjoint model is a tool that allows to obtain the sensitivities of model outputs to land surface parameters with more efficiency. The adjoint computation is also less expensive than computing the cost function gradient.

Page 4, scheme: By storage flux, do you mean tendency of the scalar (e.g dc/dt)?

Page 4, line 109: It would be worth defining what is Jarvis-Stewart approach compared to the a-gs module.

Page 4, line 120: I disagree. Within a Bayesian framework, inverse modelling does not necessarily involve any prior informa-

15 tion.

Page 4, line 122: Delete others.

Page 5, line 125: Does it mean that the land surface model parameter are not optimised?

Page 5, line 138: The reference Chevallier et al., 2010 seems to me more appropriate than Chevallier et al. 2007 here. I would justify this assumption in an other sentence using .

20 Page 6, line 149: Add a coma after at this point.

Page 6, line 150: What is the point of adding some weights instead of inflating observational errors?

Page 6, line 156: Explain how si is distributed in Equation 5.

Page 6, line 165: Above all, this method is adapted for minimizing a non-linear cost function. Please specify the algorithm used. For instance, Raoult et al. 2016 used the L-BGFS-B algorithm as many others (see also Bastrikov et al., 2018; Kuppel et

25 al 2014; Bacour et al., 2015).

Page 7, Figure 2: The figure should be more illustrative. As such, it does not help to understand the framework. At least, add the formula in the box. The iterative process should be also illustrated. See Figure 1 for instance of Thanwerdas et al., 2021.

Page 8, line 1: Specify why you optimize $Frac_H$ instead of $\epsilon_{eb}$.

Page 8, line 229: It is well known that depending on prior parameters the optimisation can get stuck in a local minimum. Please

30 cite a textbook here. See also Santaren et al., 2014 and Bastrikov et al., 2018.

Page 8, line 236: Cite Tarantola after the word approach.

Page 11, line 284: Specifify that the adjoint is computed for each iteration.

Page 13, line 362: Specify what are the arguments $checkpoint_init$ and $model$.

Page 13, line 362: The optimized emission factor can become negative as well..

35 Page 14, line 390, Remove one of the two "to".

Page 16, line 425: Specify that the chi 2 is only an indicator that can be misleading in particular when off diagonal terms are involved in the observation error matrix (Chevallier , 2007).

Page 17, line 470: Remove in after reads.

Page 18, line 483: "similar to Honnorat et al. , 2007". This is a standard test, please cite a textbook here or more references.

Page 18, line 490: It would be nice to show in a tabular the values of $\alpha$ and the associated results for the left and right sides of the equation.

Page 19, line 523: RevealS.

Page 19, line 527: OSSEs are classic to test the ability of the system to properly estimate model parameters..

Page 19, line 530, Start a new sentence after complexity and remove the coma after experiments.

Page 19, line 535: ""In the cost function..true parameters". The sentence need to be explained as prior information means to avoid the parameters taking unrealistic values.

Page 19, line 543: Add a coma after experiment.

Page 20, table 1: Previously, you wrote that you removed prior information. What does the prior column correspond to?

Page 21, line 566: as many iterations WERE needed.

Page 21, line 570: Add a coma after setup.

Page 23, line 596: Are shallow clouds represented in the forward model?

Page 23: Combine Figures 3 and 4 .

Page 26: Combine Figures 5 and 6.

Page 22, line 590: On Figure 5, the height and relative humidity show a less good fit to observations around noon. Is it because of the formation of shallow clouds?

Page 28, line 644: "The use .. model" Please explain this sentence (this is done through the use of OSSE such as e.g. Stinecipher et al., 2022).

Page 28, line 657: "It avoids..." Please explain.

Page 30, line 672: Give an example of small scale processes which are not represented.

Page 30, conclusion: You could also emphasize that the inverse framework serves at determining which observations are needed through the use of OSSEs.

**3  Bibliography**

Bastrikov, V., MacBean, N., Bacour, C., Santaren, D., Kuppel, S., and Peylin, P.: Land surface model parameter optimisation using in situ flux data: comparison of gradient-based versus random search algorithms (a case study using ORCHIDEE v1.9.5.2), Geosci. Model Dev., 11, 4739–4754, https://doi.org/10.5194/gmd-11-4739-2018, 2018.

Kuppel, S., Peylin, P., Maignan, F., Chevallier, F., Kiely, G., Montagnani, L., and Cescatti, A.: Model–data fusion across ecosystems: from multisite optimizations to global simulations, Geosci. Model Dev., 7, 2581–2597, https://doi.org/10.5194/gmd-7-2581-2014, 2014.

Chevallier, F., et al., 2010: CO2 surface fluxes at grid point scale estimated from a global 21-year reanalysis of atmospheric measurements. J. Geophys. Res., 115, D21307, doi:10.1029/2010JD013887

Frederic Chevallier. Impact of correlated observation errors on inverted CO 2 surface fluxes from OCO measurements. Geophysical Research Letters, American Geophysical Union, 2007, 34 (24), ff10.1029/2007GL030463ff. ffhal-02948201f

5      Tang, J.,  Zhuang, Q. (2008). Equifinality in parameterization of process-based biogeochemistry models: A significant uncertainty source to the estimation of regional carbon dynamics. Journal of Geophysical Research, 113, G04010. https://doi.org/10.1029/2008JG

Ziehn, T., Scholze, M., and Knorr, W.: On the capability of Monte Carlo and adjoint inversion techniques to derive posterior parameter uncertainties in terrestrial ecosystem models, Global Biogeochem. Cy., 26, GB3025, https://doi.org/10.1029/2011GB004185, 2012.

10      Thanwerdas, J., Saunois, M., Berchet, A., Pison, I., Vaughn, B. H., Michel, S. E., and Bousquet, P.: Variational inverse modelling within the Community Inversion Framework to assimilate 13C(CH4) and CH4: a case study with model LMDz-SACS, Geosci. Model Dev. Discuss. [preprint], https://doi.org/10.5194/gmd-2021-106, in review, 2021.

Santaren, D., Peylin, P., Bacour, C., Ciais, P., and Longdoz, B.: Ecosystem model optimization using in situ flux observations: benefit of Monte Carlo versus variational schemes and analyses of the year-to-year model performances, Biogeosciences, 11,
15      7137–7158, https://doi.org/10.5194/bg-11-7137-2014, 2014.

Bacour, C., et al. (2015), Joint assimilation of eddy covariance flux measurements and FAPAR products over temperate forests within a process-oriented biosphere model, J. Geophys. Res. Biogeosci., 120, 1839–1857, doi:10.1002/ 2015JG002966.

---

## Referee Comment (RC3)

**Reviews to gmd-2022-63: ICLASS 1.0: a variational Inverse modelling framework for the Chemistry Land-surface Atmosphere Soil Slab model: description, validation and application**

Anonymous

The manuscript *ICLASS 1.0: a variational Inverse modelling framework for the Chemistry Land-surface Atmosphere Soil Slab model: description, validation and application* presents a new model iCLASS, based on the CLASS model to optimize variables and parameters in a simple Land-Surface Atmosphere model. The work behind the manuscript is impressive and very useful for the community. It will allow to better understand what drives interactions between the atmosphere and land surface.

5     However, despite the quality of the underlying work, including a detailed documentation, the full development of an adjoint, proper tests, etc., the manuscript in its present form is not suitable for publication in GMD.

The structure of the manuscript does not allow the reader to fully understand what has been done, in what context, and with what strengths and limitations. The manuscript should be extensively re-written and re-submitted when improved.

Below are axes of improvements and suggestions to improve the manuscript.

10 ## 1   Introduction and bibliography

The introduction is not well-balanced and lacks pieces of bibliography. The reader would expect an extensive "review" of what has been done in parameter estimation in land-atmosphere exchanges, and not only with simple models. For instance, there has been some work on parameter estimations with full-physics models, such as ORCHIDEE or JS-BACH. The advantages vs drawbacks of simple models such as CLASS, compared to full-physics models should be more thoroughly presented. The

15 scientific "ecosystem" of the present study should be better presented. There is a full field of studies using data assimilation, machine learning, etc.

The balance between giving only hints or extensive details is also clumpsy. For instance, in paragraph p.2 l. 34-48, the authors start giving information on the model itself compared to other models, but without going to the details. What is an "extensive set of observations"? What observations are better used than other models?

20 ## 2   Energy balance and conditions of applicability of CLASS

The CLASS model is a simplified model with all its benefits and drawbacks. In particular, what are the conditions of applicability of CLASS? The authors mention "golden days" several times in the text. What are these? How frequent are they? If there is only a few such days per year, the model is not really suitable for purpose...

About the energy balance and further assumptions, it is not fully clear what is the domain of applicability of such assumptions. In particular, the advection and entrainment in the model are extremely simplified. What values and variables are used to constrain the processes?

**3 Section 3.1 and mathematical notations**

Please make your mathematical notations consistent with the rest of the community.

- prior vector: $\mathbf{x}^b$: The author should explicitly write it somewhere, with all its sub-components (bias, parameters, inputs, etc.)

- posterior vector: $\mathbf{x}^a$

- full observation operator: $\mathcal{H}$

- adjoint sensitivities are usually noted as: $\delta S^*_{win}$

Overall, Section 3.1 is very hard to understand. It is not clear at all what is optimized or not. The section gives some general information about the inversion framework, but does not go to the necessary level of details about what exactly is in each mentioned vectors and operators. The dimension and content of all operators and matrices should be detailed.

The weights on observations or "regularization factors" are clumsy and not justified. If one observation is less worthy than another, then the uncertainty should just be scaled up, with no need for an extra complicated parameter.

Equation (6) is too implicit. The author should fully detail the "background" term, including what they optimize or not.

**4 Uncertainties and OSSEs**

Please provide extensive details on the uncertainties you specify for the inputs and parameters and some justification for the corresponding uncertainties. In particular, for parameters, the normal distributions are not necessary the most obvious choice. This should be justified and detailed.

The OSSEs are rather simple and do not fully allow to validate the model. More OSSEs should be made more systematically to show what is the influence of a given parameter in a given set-up. The author can perturb a parameter but not optimize it, etc. Besides, I may have missed the information, but I have the impression that the bias correction is not evaluated in the OSSEs. This should be added.

Regarding the posterior uncertainties, having truncated Normal distributions means that the minimum of the cost function is the node of the posterior distribution, which is not the mean or median, contrary to full normal distributions. Therefore, the authors should give further details on how the compute and analyze posterior distributions.

**5 Details on the model**

There is critical information missing about the CLASS and iCLASS models. Some of this information is given in the documentation of iCLASS, but not comprehensively. The reader cannot be expected to read the non-reviewed documentation to understand the article and how the adjoint is built.

In particular, there should be full details on the inputs and parameters of the CLASS models. What are the resolutions of each inputs? Where do they come from? Are they given by in-situ measurements? Meteorological forcing fields?

Similarly, what are the exact outputs of the model? How the output is compared to observations.

Finally, what is computed by the model? And what is given as inputs?

**6 Superfluous sections and elements**

The text is made hard to follow by numerous superfluous details.

For instance, section 4 is mainly made of a technical lecture on how to code an adjoint. This can be removed altogether.

**7 Technical comments**

1. p.1 l.9: replace "the core physics to model" by "the core physics to simulate"

2. p.3 l.63: The example is rather a negative feedback but not an obvious non-linearity. There are probably better examples.

3. p.3 l.66: "Analytical" is ill-chosen and refers to analytical inversions in the inversion framework. The adjoint is simply needed to compute explicitly and efficiently the gradient of the cost function, without relying on, e.g., finite-element estimations

4. Section 8: the validation of the adjoint using the gradient test and the test of the adjoint is really appreciated! The results of the test of the adjoint is generally reported as a N times the machine epsilon ($10^16$ in present machines)

5. p.15 eq.20: $x_A$ is modified in the Monte Carlo.

6. p.15 eq.22: $\chi^2$ formula is wrong for two reasons. First the chi-square diagnostics can be applied only with normal distributions. Truncated-Gaussians break the diagnostics; but for not so truncated Gaussians, it may still be valid.

Second, the authors mixed two versions of the chi-square diagnostics: one from, e.g., from Michalak et al. 2005 (doi:10.1029/2005JD005970), the other from, e.g., Zupanski et al. 2006 (https://doi.org/10.1175/MWR3125.1). In one version the chi-square has a mean of $n$ (nb obs) and in the other $n + m$ (nb obs + parameters). As written in eq.22, the expected mean is $n$, or the authors compute the other version, but should explain more clearly what is done.

---

## Author Comment (AC1)

Dear reviewers,

Thank you all very much for the time you have spent on reading our manuscript, and in particular for your constructive comments, which helped to improve our manuscript. Please find a point-to-point reply to each of your comments below, sorted per reviewer.

**Reply to reviewer 1:**

Page 1, line 6: Replace "enables to estimate" by "enables the estimation of information".

We have replaced this as suggested.

Page 7, line 7: "free-tropospheric mixing ratios". I disagree, free tropospheric mixing ratios are not difficult to obtain by observations. There are numerous surface sites measuring greenhouse gases concentrations around the globe.

Many surface sites indeed measure greenhouse gas concentrations, these measurements however often take place relatively close to the surface (e.g. measurement tower). With 'free-tropospheric' we want to indicate the concentration in the free troposphere above the boundary layer, a quantity that determines entrainment. However, measurement towers seldomly extend beyond the boundary layer. Therefore, the free-tropospheric concentrations are not always straightforward to obtain in our view.

Page 1, line 19: Add an s after exchange.

Adapted

Page 1, line 21; Strictly speaking, the second part of the sentence (the well known atm…) is false. The atmospheric boundary layer exists even though the daytime conditions are not sunny.

Indeed, we now changed the sentence into "Surface heating under sunny daytime conditions usually leads to the growth of a relatively well-mixed layer close to the land surface, the convective boundary layer (CBL)."

Page 2, line 30: Add after scalars (e.g. wind speed and temperature).

We have changed the line into "relatively strong vertical gradients of scalars (e.g. specific humidity and temperature) …"

Page 2, line 30: For which time scale and horizontal resolution these assumptions are valid?

Regarding the time scale, the model performs best during the convective daytime period, the assumptions on advection etc. should be valid for the whole modelled period. Regarding the horizontal scale: The model performs best on fair-weather days. The absence of deep convection etc. should ideally hold on a scale large enough that it does not influence the model simulation location. In practice, days are often not 'ideal', e.g. a time-varying advection can be present. This does not necessarily mean the model cannot be applied to that day, but, performance is likely to be worse.

We have added info about this to the introduction.

Page 2, line 35: Parenthesis within parenthesis. Use "for instance at Cabaw"..

We have adapted the sentence to avoid parenthesis within parenthesis

Page 2, line 40: Here, you can mention the problems of equifinallity (Tang et al., 2008) and overfitting.

We have added "The estimation of parameters is further complicated by possible over-fitting and the problem of parameter equifinality (Tang and Zhuang, 2008), the latter especially in case not enough types of observations are used"

Page 2, line 42: Replace Inital by Initial.

Thanks for spotting this typo, adapted

Page 2, line 42: Replace e.g. by for instance.

The sentence now reads "Some parameters can be obtained directly from observations (for instance initial mixed-layer humidity), but, for example, estimating free-tropospheric lapse rates or stomatal conductances is often more challenging."

Page 3, line 66: This is also illustrated in Ziehn et al. (2012) with the assimilation of atmospheric $CO_2$ data in BETHY LSM.

Around line 66 our manuscript has the following text: "The non-linearity causes numerically-calculated cost function gradients to deviate from the true analytical gradients, since the cost function can vary erratically with a changing model parameter value. This is hampering proper minimization of the cost function when using numerically calculated gradients."

The suggested reference is interesting, but we could not find the location in the paper of Ziehn et al. where these authors illustrate this point about numerical gradients.

Page 3, line 62: Above all, it is the iterative process that allows to find the local miminum of the cost function in case of linearity.

We have extended Figure 2 to make the iterative cycle clearer, see later in this document, as this cycle is indeed important.

Page 3, line 65: The choice of using variational methods compared to other technics dealing with the non linearity (e.g. Particule filters) could be discussed here. The advantages of using an adjoint compared to a numerically computed gradient could be also added. For instance, the adjoint model is a tool that allows to obtain the sensitivities of model outputs to land surface parameters with more efficiency. The adjoint computation is also less expensive than computing the cost function gradient.

We have added the following: "This approach furthermore allows to efficiently retrieve the sensitivity of model output to model parameters. Also, using an analytical gradient is generally computationally less expensive compared to using a numerical gradient (Doicu et al., 2010, p17)."

It is not our intention to provide an overview of possible methods here, as a proper overview would soon become quite extensive, and the paper is already quite substantial in length.

Page 4, scheme: By storage flux, do you mean tendency of the scalar (e.g dc/dt)?

Yes, We have adapted 'storage flux' into tendency now

Page 4, line 109: It would be worth defining what is Jarvis-Stewart approach compared to the a-gs module.

We have adapted the text as follows: "As an alternative for a-gs, a Jarvis-Stewart approach (Jarvis, 1976; Stewart, 1988) can also be used for calculating $H_2O$ exchange. The latter approach is more simple: stomatal conductance is calculated as a maximum conductance multiplied with a set of factors between 0 and 1 (Jacobs, 1994). In CLASS, there are 4 factors included, which

represent limitations due to the amount of incoming light, temperature, vapour pressure deficit and soil moisture."

Page 4, line 120: I disagree. Within a Bayesian framework, inverse modelling does not necessarily involve any prior information.

It can indeed be done without prior info, although adding the extra prior information often improves the solution or avoids ill-defined situations. We have slightly adapted the sentence: "Inverse modelling is based on using observations and, ideally, prior information to statistically optimise a set of variables driving a physical system (Brasseur and Jacob, 2017)."

Page 4, line 122: Delete others.

Deleted

Page 5, line 125: Does it mean that the land surface model parameter are not optimised?

No. Here we wanted to make a distinction between model parameters that are optimised and those that are not optimised (but still can have an influence on the model output). The first group are part of the state and thus vector $x_m$. The latter group of parameters are part of vector p. At this point in the paper we do not make a choice on which parameters to optimise and which not, that depends on the specific optimisation problem one wants to use ICLASS for, and can be chosen by the user. The full list of parameters that can be optimised is quite large (given in manual), and includes land surface model parameters as well.

Page 5, line 138: The reference Chevallier et al., 2010 seems to me more appropriate than Chevallier et al. 2007 here. I would justify this assumption in an other sentence using .

We have changed the Chevallier et al. 2007 reference into the Chevallier et al., 2010 reference. The remark "I would justify this assumption in an other sentence using ." was not fully clear to us.

Page 6, line 149: Add a coma after at this point.

Added

Page 6, line 150: What is the point of adding some weights instead of inflating observational errors?

Indeed identical changes can be made to the cost function by adapting weights or changing the observational errors. However, the observational error standard deviations are also used in the ensemble for estimating posterior errors (see section 5.2). When the observational errors are no longer realistic due to inflating/deflating these errors, the observations are not properly perturbed anymore. This problem is avoided when using weights. The latter can be used, for example, when you have 15 temperature observation streams, but only one $CO_2$ observation stream. In this case adding a weight of 1/15 to the temperature observation streams can make the observation streams more balanced, while keeping a realistic error for the observations. We have added an additional sentence to the text of the paper: "In principle, the observational error variances could also be adapted for this purpose, but by using weights we can keep realistic error estimations (important for Sect. 5.2)."

Page 6, line 156: Explain how si is distributed in Equation 5.

We have changed the sentence below eq 5 "These errors are assumed to be independent of each other." into "These errors are assumed to be independent of each other and normally distributed."

Page 6, line 165: Above all, this method is adapted for minimizing a non-linear cost function. Please specify the algorithm used. For instance, Raoult et al. 2016 used the L-BGFS-B algorithm as many others (see also Bastrikov et al., 2018; Kuppel et al 2014; Bacour et al., 2015).

The text now reads "The framework uses by default a truncated Newton method, the *tnc* algorithm (The SciPy community; Nash, 2000), for the optimisations. Truncated Newton methods are suitable for nonlinear optimisation problems (Nash, 2000). The chosen algorithm allows for specifying hard bounds…"

Page 7, Figure 2: The figure should be more illustrative. As such, it does not help to understand the framework. At least, add the formula in the box. The iterative process should be also illustrated. See Figure 1 for instance of Thanwerdas et al., 2021.

The figure was indeed very limited. The new figure:

[Figure]

**Figure 2.** Slightly simplified sketch of the workflow of the inverse modelling framework, when using the adjoint model for the derivative. Note that, for clarity of the figure, direct arrows between the parameters and the cost function and its gradients are not drawn. These arrows arise via the background part of the cost function (see equations in text). Everything within the shaded rectangle is part of the iterative cycle of optimisation. Model parameters that are not optimised are in vector $p$, this vector is used together with $x_m$ in every model simulation.

Page 8, line 1: Specify why you optimize Frac$_H$ instead of $\epsilon_{eb}$.

In our application example, $\epsilon_{eb}$ (the energy balance residual, see eq 8) is explicitly calculated from the observations, since we had radiation observations available. Optimising FracH ensures that the energy balance in the observations closes, as the difference between net radiation and the sum of all new heat fluxes becomes 0. If we would optimise $\epsilon_{eb}$ this would not be the case. We have slightly adapted the text below eq 10: "This implies that the energy balance closure residual is added partly to the sensible, partly to the latent heat flux." is changed into "This implies that the energy balance closure residual is added partly to the sensible, partly to the latent heat flux, thereby closing the energy balance in the observations."

Page 8, line 229: It is well known that depending on prior parameters the optimisation can get stuck in a local minimum. Please cite a textbook here. See also Santaren et al., 2014 and Bastrikov et al., 2018.

As we don't readily have a clear textbook example to cite, we added some more references, the text now reads: "The highly nonlinear nature of the optimisation problem can cause the optimisation to get stuck in a local minimum of the cost function (Santaren et al., 2014; Bastrikov

et al., 2018; Ziehn et al., 2012). This means that the resulting posterior state vector can depend on the prior starting point (Raoult et al., 2016), and the resulting posterior state can remain far from optimal."

Page 8, line 236: Cite Tarantola after the word approach.

Adapted

Page 11, line 284: Specify that the adjoint is computed for each iteration.

At line 162-165 in chapter 3, the following text is present: "In the statistical optimization, we attempt to find the values of the state vector x such that the function in Eq. (6) reaches its absolute minimum. This is done starting from an initial guess (x = $x_A$), after which the state vector is improved iteratively. The cost function and the gradient of the cost function (derivatives with respect to all parameters) are computed for different combinations of parameters in the state vector (Fig. 2)." We herein also refer to figure 2 (see higher up in this document), which we have extended, and wherein we made the iterative cycle clearer. Since line **284** belongs to a section that is more about illustrating the employed technique of adjoint coding, we prefer to not mention this in that section. The latter section will be moved to the supplementary material, in response to comments of other reviewers.

Page 13, line 362: Specify what are the arguments checkpoint‗init and model.

'model' is a forward model object passed as argument to the function, this is just a technical Python implementation, we have removed this argument in the example for simplicity. checkpoint_init[i] contains stored forward model variables, as explained in Sect. 4.3, We have added this info to the text.

Page 13, line 362: The optimized emission factor can become negative as well..

We assume this is about page 14, line 378? This is indeed true, but the emissions are not simply multiplied with a factor. Bergamaschi et al 2009 use the following formula for emissions (their eq 4):

$$e = e_{apri0} * \exp(x) \qquad for\ x < 0$$

$$e = e_{apri0} * (1 + x) \qquad for\ x > 0$$

The emission parameter (x) itself is unbounded, but the emissions (e) cannot become negative. To make it more clear, we have changed the text as follows:

"Their solution was to make the emissions a function of an emission parameter that is being optimised, instead of optimising the emissions themselves. By their choice of function, the emissions cannot become negative, even though the emission parameter is unbounded."

Page 14, line 390, Remove one of the two "to".

Removed

Page 16, line 425: Specify that the chi 2 is only an indicator that can be misleading in particular when off diagonal terms are involved in the observation error matrix (Chevallier , 2007).

We have added a similar statement to the text: "Note however that the $\chi 2$ statistic can be misleading, in particular when observational errors are correlated (Chevallier, 2007)"

Page 17, line 470: Remove in after reads.

The text now reads "… in this file observations are loaded, the state vector defined, etc."

Page 18, line 483: "similar to Honnorat et al. , 2007". This is a standard test, please cite a textbook here or more references.

The gradient test is indeed widely applied, but to our knowledge few papers give a detailed formula like Honnorat et al. (2007), that is similar to our formula. We now also refer at this place in the text to Elizondo et al. (2000).

Page 18, line 490: It would be nice to show in a tabular the values of α and the associated results for the left and right sides of the equation.

The paper now includes the following table:

| $\alpha$ (m) | 1 - ratio RHS and LHS (−) |
| --- | --- |
| 0.5 | $4.7 \times 10^{-1}$ |
| 0.2 | $3.2 \times 10^{-1}$ |
| 0.1 | $2.2 \times 10^{-1}$ |
| $1 \times 10^{-2}$ | $3.4 \times 10^{-2}$ |
| $1 \times 10^{-3}$ | $3.6 \times 10^{-3}$ |
| $1 \times 10^{-4}$ | $3.7 \times 10^{-4}$ |
| $1 \times 10^{-5}$ | $3.7 \times 10^{-5}$ |
| $1 \times 10^{-6}$ | $4.2 \times 10^{-6}$ |
| $1 \times 10^{-7}$ | $2.6 \times 10^{-6}$ |
| $1 \times 10^{-8}$ | $-8.2 \times 10^{-6}$ |
| $1 \times 10^{-9}$ | $5.9 \times 10^{-4}$ |
| $1 \times 10^{-12}$ | $-8.2 \times 10^{-2}$ |

Page 19, line 523: RevealS.

Adapted, thanks for spotting the typo

Page 19, line 527: OSSEs are classic to test the ability of the system to properly estimate model parameters..

We have added "This type of experiments is classic to test the ability of the system to properly estimate model parameters."

Page 19, line 530, Start a new sentence after complexity and remove the coma after experiments.

Adapted

Page 19, line 535: ""In the cost function..true parameters". The sentence need to be explained as prior information means to avoid the parameters taking unrealistic values.

This is specifically for the OSSEs. We first define 'true' parameters, which we use to create observations. Then, we start from a different prior state, and we want to try to find the true parameters back, using the observations we created earlier. Now, if we would include the background part of the cost function, i.e. a penalty for deviating from the prior, this would mean that we will not be able to find back the true state. This is because the true state would give the

best fit to the observations, but due to the penalty for deviating from the prior, this would normally not correspond to the minimum in the cost function. Therefore, we leave out the background part of the cost function.

We have added some info to the text: "In the cost function, we do not include the background part, to make sure that it is possible to find back the "true" parameters. This is because the background part of the cost function implies a 'penalty' for deviating from the prior state. This penalty implies that, when the model is run with the true parameters, the cost function would still not be zero, and the minimum of the cost function might be shifted."

Page 19, line 543: Add a coma after experiment.

Added

Page 20, table 1: Previously, you wrote that you removed prior information. What does the prior column correspond to?

The prior starting state. Even though the deviation from the prior is not included in the cost function (see our response about your comment about Page 19, line 535), the optimisation still needs a starting point.

Page 21, line 566: as many iterations WERE needed.

Sentence now reads "In this case, convergence is notably slower, e.g. more than six times as many iterations were needed to reduce the cost function to less than …"

Page 21, line 570: Add a coma after setup.

Added

Page 23, line 596: Are shallow clouds represented in the forward model?

In the configuration we used, the model does not take shallow (or any other) clouds into account. This can give rise to some deviation between observations and model, but we still expect the model grasps the main physics governing the boundary layer state. But see also our reply to your comment about Page 22, line 590.

Page 23: Combine Figures 3 and 4 .

Combined

Page 26: Combine Figures 5 and 6.

Combined

Page 22, line 590: On Figure 5, the height and relative humidity show a less good fit to observations around noon. Is it because of the formation of shallow clouds?

In radiation measurements of that day we see a reduction in incoming shortwave radiation for many data points around noon (see fig below). Earlier we wrote in the paper at line 604 about cumulus clouds. However, a colleague of us recently provided us with a satellite image of the day, the image suggests that high clouds were present instead. We have therefore adapted the text. The high clouds might play a role in the less good fit, although this issue is not easy to examine.

[Figure]

Page 28, line 644: "The use .. model" Please explain this sentence (this is done through the use of OSSE such as e.g. Stinecipher et al., 2022).

We have added the following: "This is done through the use of observation
system simulation experiments, similar to e.g. Ye et al. (2022)". We could not find a Stinecipher 2022 reference with OSSEs, therefore we used a different reference.

Page 28, line 657: "It avoids..." Please explain.

See also line 40-44, what we wanted to say here is that, with a framework like this, we avoid the need of manually fitting parameters of the forward model to obtain a good fit to observations (People using CLASS had to do this before this framework was built). Manually fitting parameters can be time-consuming and subjective. We have changed the sentence into "It avoids the need of manual trial-and-error in choosing parameter values for the model when fitting observations, thereby providing more objectivity."

Page 30, line 672: Give an example of small scale processes which are not represented.

The text now reads "... we cannot expect a relatively simple model to capture all small-scale processes playing a role in the atmospheric boundary layer and in land surface--atmosphere exchange (e.g. heterogeneous surface heating and evaporation, influence of individual thermals, ...)."

Page 30, conclusion: You could also emphasize that the inverse framework serves at determining which observations are needed through the use of OSSEs.

Thanks for this suggestion, we have added the following text to the concluding discussion: "ICLASS can also help in the planning of observational campaigns, to determine in advance which observation streams are needed to better constrain model processes."

**References**

Bergamaschi, P., Frankenberg, C., Meirink, J. F., Krol, M., Villani, M. G., Houweling, S., Dentener, F., Dlugokencky, E. J., Miller, J. B., Gatti, L. V., Engel, A., and Levin, I.: Inverse modeling of global and regional CH4 emissions using SCIAMACHY satellite retrievals,
Journal of Geophysical Research Atmospheres, 114, 1–28, https://doi.org/10.1029/2009JD012287, 2009.

Doicu, A., Trautmann, T., and Schreier, F.: Numerical Regularization for Atmospheric Inverse Problems, Springer Praxis Books in environmentral sciences, https://doi.org/10.1007/978-3-642-05439-6, 2010.

Elizondo, D., Faure, C., and Cappelaere, B.: Automatic- versus Manual- differentiation for non-linear inverse modeling, Tech. rep., INRIA (Institut National de Recherche en Informatique et en Automatique), https://hal.inria.fr/inria-00072666/document, 2000.

Honnorat, M., Marin, J., Monnier, J., and Lai, X.: Dassflow v1.0: a variational data assimilation software for 2D river flows, Tech. rep.,INRIA (Institut National de Recherche en Informatique et en Automatique), http://hal.inria.fr/inria-00137447, 2007.

Jacobs, C.: Direct impact of atmospheric CO2 enrichment on regional transpiration, Ph.D. thesis, Wageningen University, 1994.

Ye, H., You, W., Zang, Z., Pan, X., Wang, D., Zhou, N., Hu, Y., Liang, Y., and Yan, P.: Observing system simulation experiment (OSSE)-quantitative evaluation of lidar observation networks to improve 3D aerosol forecasting in China, Atmospheric Research, 270, 106 069,https://doi.org/10.1016/j.atmosres.2022.106069, 2022.

**Reply to reviewer 2**

**General comments**

The introduction to the paper is off the mark. It does not explain the links between ICLASS and the efforts of other models but contains a lot of more or less technical information (e.g. on the tangent-linear and adjoint). I think that readers interested in a variational inverse modelling framework may already know about the TL and adjoint. If the aim is to teach users of CLASS what is an inversion and how they can use it, it may not be best done with a paper in GMD.

To place the variational framework of this paper in comparison with other efforts in the scientific community, we now added a paragraph linking parameter estimation in land-surface models in other studies with ICLASS. Here, an important point we make is that the fully coupled land-atmosphere in ICLASS helps to infer land surface characteristics from atmospheric observations, something that is often not the focus of other variational frameworks.
The more technical text mentioning the adjoint in the introduction, is limited to one paragraph, discussing the challenge that non-linearity is posing.

The order for presenting the variables and various definitions is not always very logical or at least, easy to follow for the reader, particularly in Section 3. The whole of Section 4 and most of Section 8 are not relevant, as well as some theoretical paragraphs in Sections 3 and 5 (see Specific comments for more details).
In response to this valid comment, and a similar comment from another reviewer, the content of chapter 4 has been moved to the supplementary material. See specific comments for sections 3, 5 and 8.
The validation (Section 9) must deal with more relevant tests and show the uncertainties.
The simple OSSEs in the paper mainly focus on retrieving parameter values, prior uncertainties were not used. We will add a more sophisticated OSSE, including a test for the bias correction. See also specific comments. We will use an ensemble in the new OSSE, providing posterior uncertainties.

The same remark applies to the application example (Section 10): no posterior uncertainties are shown even though ICLASS can estimate them with its Monte-Carlo scheme.
Here we would like to point to table 3 and figure 9. In the last column of table 3, we show the posterior standard deviation of every parameter. In Figure 9 we picked out 2 parameters and show the full posterior pdfs.
Finally, some very practical information is missing, e.g. about the computation costs.
We will add info on the computation costs. Note that some brief info on computation costs relative to CLASS is already given on lines 647-653.

**Specific comments:**

*Introduction*

The introduction should be rewritten to include more of the general context surrounding ICLASS e.g. how is it linked to the efforts around other models. Nevertheless, in case they are useful, here are some remarks on specific points:
-p.2 l.31-34: what is the typical frequency of the "golden days" in a year? How are they

distributed? At least in the area where the example application is located.

The model performs best during the convective daytime period, the assumptions on advection etc. should be valid for the whole modelled period. Since the model performs best on fair-weather days, the absence of deep convection etc. should ideally hold on a spatial scale large enough that it does not influence the model simulation location. In practice, days are often not 'ideal', e.g. a time-varying advection can be present. This does not necessarily mean the model cannot be applied to that day, but, performance is likely to be worse. We have added info about this to the introduction. Determining the frequency of 'ideal' days is quite complex, as then advection etc. has to be known. Even though the model does not perform well in all meteorological situations, this and similar models have been successfully applied in numerous studies, see https://classmodel.github.io/publications.html.

-p.2 l.39-40: this is not true: neural networks or statistical models have no physics at all and their results can be consistent with measurements...

The results of those models can indeed be consistent with a set of measurements, but the point we want to illustrate here is the following: If you tune the parameters of the (CLASS) model using e.g. only $CO_2$ mixing ratio observations, you might easily manage to get a good fit to those observations. Several choices of parameter sets might give you similar results, as one parameter can compensate for another when only looking at one specific type of observations. But then, when keeping the same set of parameters chosen earlier, and comparing your model output also with humidity and temperature observations, likely your model will perform poorly. This means your model physics are not correct, but if you would only compare to $CO_2$ mixing ratios, this internal problem would remain hidden. If instead you fit model parameters using a wide range of different types of observations, you are likely to end up with model physics that are more correct, i.e.: it becomes less likely that one bad parameter can compensate for another. Of course, the essential physical processes should be well represented in the model, otherwise even the best set of parameters might not lead to a good fit.

In case of statistical models fitted with $CO_2$ mixing ratio observations, there will be no model output for variables other than $CO_2$ mixing ratio, they have no internal physics, so our statement in the paper "When model results are consistent with a diverse set of measurements, this gives more confidence that the internal physics are robust and the model has been adequately parameterised to reliably simulate reality" cannot be applied to those models.

-p.2 l.49 "capable of correcting observations for biases": this is a bit misleading as to what is done by ICLASS. Any inversion set-up can "correct observations for biases" if a control variable is created for it. The issue is whether the resulting corrections have any physical meaning.

The text reads "The above text illustrates the need for an objective optimisation framework, capable of correcting observations for biases. We therefore present here a description of ICLASS, an inverse modelling framework built around the CLASS model, including a bias-correction scheme."

It is indeed true that more complex bias patterns cannot be handled. There is however a capacity to physically correct observations for biases, and we would like to point to Figure 7 for this. The surface heat flux observations, which are often assumed to be prone to underestimation (see e.g. Foken 2008), are adapted in the direction one would expect.

We changed the text into "The above text illustrates the need for an objective optimisation framework, capable of correcting observations for biases.  We therefore present here a description of ICLASS, an inverse modelling framework built around the CLASS model, including a bias-correction scheme for specific bias patterns."

-p.3 l.64-65: beware, non-linear is not random (which I assume to be the meaning of "erratically" here).

"The non-linearity causes numerically-calculated cost function gradients to deviate

from the true analytical gradients, since the cost function can vary erratically with a changing

model parameter value."

What we wanted to say here is that in this case the cost function can (theoretically) change in a very non-linear way with a change in parameter value, e.g. increases and decreases of the cost function can alter with very small changes in the parameter, the shape of the cost function can be very irregular. We have changed erratically into irregularly.

*Forward model*

Please check which pieces of information are actually relevant for the inversion framework. If an option is not used in the tests or example application, it may not be explained here.
-p.4 l.96: how is the cloud mass flux included? Or is it not relevant here?
In the beginning and the end of the section we refer to Vilà-Guerau De Arellano et al (2015), where these details can be found. We do not include the cloud mass flux in the example, we shortly mentioned it here for completeness. Also, for readers who want to perform a study with a bigger focus on cumulus clouds, using ICLASS, it might be good to know it can be included.
-p.4 l.98: how are cloud effects on the BLH accounted for? Or is it not relevant here?
Idem to comment above
-p.4 l.101: do you use the option for the Monin-Obukhov similarity?
Yes, we consider this layer very important for correctly interpreting observations. We have now explicitly added 'we included the surface layer option in the model' to the section of the application example. Also for the OSSEs we now made clear that the surface layer was turned on.
-p.4 l.102-105: this very long sentence is not clear, please rephrase.
The two original sentences were "In the original CLASS surface layer, scalars, the zonal

wind speed and the meridional wind speed are evaluated at 2 m height. For some scalars, we

have extended this to multiple user-specified heights, as this allows to compare model output to

observations of chemical mixing ratios and temperatures at different heights (e.g. along tower)."

We changed it into "In the original CLASS surface layer, scalars such as temperature are

evaluated at 2 m height. For some scalars, we have extended this to multiple user-specified

heights. This allows to compare observations of chemical mixing ratios and temperatures at

different heights (e.g. along a tower) to model output."
-p.4 l.107: do you use this option?
Yes, both in the OSSEs and in the application example.
-p.4 l.107-108: "a-gs" module and big-leaf method are not defined/referenced anywhere.
Is it supposed to be commonly known methods?
Within the carbon community, these are relatively well known, but it is good to provide references for both. For a-gs we refer to (Jacobs, 1994; Ronda et al., 2001), for big-leaf approach we added a reference to Friend (2001).
-p.4 l.111: from which data does the model dynamically compute the long and short wave radiations?
We have added the following sentence: "In this module, shortwave radiation is calculated using the date and time, cloud cover and albedo. For longwave radiation the

calculation uses surface temperature and the temperature at the top of the surface layer."

We turned this feature on in both the OSSEs and application example.
-p.4 l.114: where do the surface temperatures come from?

The model calculates the surface temperature from solving the energy balance, the use of outgoing longwave radiation from the previous timestep makes this more simple (outgoing longwave radiation is a 4[th] power function of surface temperature).

We have adapted the referred sentence into: "The soil heat flux to the atmosphere is calculated based on the gradient between soil and surface temperature, the latter is obtained from a simplified energy balance."

*Inverse modelling framework*

-p.4 l.122-123: please clearly list the inputs and/or put them in Fig.1
We assume figure 2 was meant here (since figure 1 is about the forward model)? We have reworked figure 2 (also based on comments of another reviewer) into the following:

[Figure]

The prior input vectors $x_b$ and $x_m$ are shown in the figure. In case the reviewer meant Figure 1, the model has more than 50 parameters that could be optimised, more than can be properly shown in a figure.

-p.4 l.123: "[y]our bias correction scheme" has not yet been described. Moreover, the remaining parts of this subsection deals only with xm: please try to make the layout easier to follow for the reader.
Bias correction is elaborated in section 3.2 (which comes after), but it is already introduced in the introduction. In response to this comment we now refer forward to section 3.2.
-p.5 l.125-126: what are the "model parameters that are not part of the state"? If they don't, why are they in the model at all?
The model has more than 50 parameters that can be optimised. Usually, the user will only want to optimise a subset of all these parameters, to reduce the complexity of the optimisation problem. Thus, only a subset of all model parameters is in the state. The other parameters, even though they are kept constant, still have an influence on the model output and thus the cost function. If they were given other constant values, the model output might be different. Those parameters, that are not part of the state vector, but still have an influence on the model output, we place in a vector p. Brasseur and Jacob (2017) also use a vector p in their notation (see their eq 11.1).

-p.5 l.126seq: your notations are not conventional - at least, not from the atmospheric inversion conventions. We use R and B for the covariance matrices, for example.
Different communities prefer different notation. We based our notation on Brasseur and Jacob (2017), and their notation is to a large extent based on Rodgers (2000).
-p.5 l.132-p.6 l.148: all this is part of the general theory of the inversion, it is not particular to ICLASS so I think it must be omitted. Only the information that the observation errors are uncorrelated is relevant.
We understand the point of view of the reviewer, who wants to make this section more concise. We argue however that some of this information, like the splitting up of the observational error variance in different parts, is relevant for the ICLASS user, who has to provide values of $\sigma_I$ and optionally $\sigma_M$ and $\sigma_R$. Next to that, some of the potential users of ICLASS are not very experienced with inverse modelling, this extra information might be very helpful to them.
In response to this comment, we have moved the equation of the a-priori error covariance matrix and the accompanying text to the supplementary material, as this is common knowledge.
-p.6 l.154: how can these factors be optimised?
They can be optimised similarly to the other parameters, by iteratively calculating the gradient of the cost function (eq 13 gives the derivative with respect to a scaling factor) and the cost function itself for various values of the scaling factor. They are also part of the state when included in the optimisation.
-p.6 l.158-165: this is again part of the general theory of the inversion.
We consider Equation 7 non-standard since it contains an observation scaling factor
-p.7 3.2: put the definitions of xm before l.125. Maybe xb also.
Both are shortly introduced at lines 120-125. Moving the explanation from lines 175-180 to a location before line 125 is very difficult, since the observation scaling factors are not yet defined at that point.
-p.7 l.179: where do FracH appear in J? This is only indicated in Eq.11.
FracH influences part of vector y (see eq 9 and eq 10), which appears in J (eq 4). In principle we could write y in eq 4 as y(FracH). FracH is however not yet introduced at the moment the cost function is defined, and y is only a function of FracH if the user decides to include the energy balance closure bias-correction.
-p.7 l.180: "this is the topic of the next section": this is not a valid transition between sections. It is useless or may indicate that the sectionning and order of the sections is not logical enough.
The transition is altered, the text now reads "The second possible method of bias correcting (Sect. 3.3) is implemented specifically for the energy balance closure problem (Foken, 2008; Oncley et al., 2007; Renner et al., 2019), it involves a parameter "FracH" (-) that can be optimised."

Note that in this section we want to give an overview on what sorts of parameters can be optimised, the bias correction for energy balance closure is explained in the section that follows. We however include this one parameter from the next section, to be complete.
-p.7 l.186-197: why may the user desire to specify their own observational energy balance closure residual?
All the measurements appearing in Eq. 8 might not always be available for all studies
-p.8 l.193: can you conclude on the advantages and limitations of this bias correction?
We have added the following:
"Limitations of this approach are that we assume the radiation and soil heat flux measurements to be bias-free, and the FracH parameter constant during the day."
Regarding the advantages, we changed the following sentence "This implies that the energy balance closure residual is added partly to the sensible, partly to the latent heat flux" into
"This implies that the energy balance closure residual is added partly to the sensible, partly to the latent heat flux, thereby closing the energy balance in the observations."
-p.8 l.195-211: this is the general theory of the adjoint, it is not particular to ICLASS.

Equation 13 is the derivative to the observation scaling factor, which we think is not a standard equation. Eq 12 defines the forcing vector, which is used in eq 13 and 14 that deal with the bias correction.

-p.8 l.214: what are "forcing vectors"?

These are defined in eq 12, they contain the model-data mismatch, and are used as forcing for the adjoint (eq 11). See also Brasseur and Jacob (2017).

-p.8 l.215-217: this is not clear: what is the link between FracH, FH, the observation scaling factors? Please clarify the vocabulary.

It becomes indeed quite confusing with so many variables playing a role. FracH is specific for the energy balance closure problem, and explained in section 3.3. $F_H$ is a forcing vector for the H (sensible heat flux) observations, the definition of a forcing vector is given in eq 12. $F_H$ is used in the derivative of the cost function to the FracH parameter (eq 14). The observation scaling factors are introduced in eq 6, they are unrelated to FracH. Note that we have now added a table in the appendix describing all inverse modelling variables from section 3.

-p.9 l.226: what are the advantages and limitations of the numerical derivative compared to the analytical gradient?

An analytical gradient is generally computationally less expensive compared to using a numerical gradient (Doicu et al., 2010). In the case of ICLASS, we are not aware of any advantage of using the numerical derivative. Comparing the numerical and analytical derivative however can provide an extra check on the analytical derivative, and it can be interesting to see at which step size the differences become big. Also, in the OSSEs we use the numerical derivative at one point (line 565) to compare with our adjoint, so it might be useful to keep the employed formula in the paper

-p.9 l.228-230: general theory, remove.

This is indeed well-known within the inverse modelling community. It however serves here as the introduction of the section on convergence challenges, and as an argument on why the Monte-Carlo ensemble is useful.

-p.9 l.230-232: if the forward model crashes, aren't there any other issues than the inversion?

The forward model is very non-linear, certain combinations of input parameters lead to unphysical situations or numerical instabilities. Since CLASS is a simple model, it does not have advanced systems to prevent or deal with this kind of issues. Still, this and similar models have been successfully applied in numerous studies, see https://classmodel.github.io/publications.html.

-p.9 l.233: "on which state vectors are tested": a missing word?

Indeed a confusing sentence, now it reads "After starting from a user-specified prior state vector, the tnc algorithm autonomously decides which parameter values are tested during the rest of the optimisation."

-p.9 l.236-239: general explanation on the Monte-Carlo principle, not particular to this work.

This section is about what we have done to handle convergence challenges. The short explanation (4 lines) might indeed be quite general, but important to understand what is done. Furthermore, what is specific (not unique) to ICLASS is that we use the variational approach (our minimisation procedure) within the Monte Carlo approach (ensemble).

We would like readers to more or less understand how ICLASS works, without having to read other papers. We would like to keep this (in our opinion) important information in the paper.

Figure 1: please indicate also the inputs and outputs.

We assume this is about figure 2, we have reworked this figure (see higher up in this document).

*Adjoint model*

I appreciate the very pedagogical drive regarding the adjoint but I think that this section must be removed altogether since I don't think the reader of such a paper expects a lecture on the adjoint.

This section is moved to the supplementary material

*Error statistics*

-p.14 l.381-383: does it invalidate the approach not to keep in the normality assumption?
Why?
In the derivation of the commonly used general cost function equation, it is assumed that both prior and observational errors follow a (multivariate) normal distribution (Tarantola 2005). We however cannot keep the normality assumption, because we use hard bounds for state values via the tnc algorithm. This induces a certain inconsistency, and the degree of error will depend on the degree of truncation etc. However, there are more studies that apply hard bounds, e.g. Raoult et al. (2016). Even though the normality assumption is violated, we think the results can still be useful. We added a sentence: "For a parameter following a truncated normal prior distribution, the prior variance used in the cost function is not (fully) equal to the variance of the actual prior distribution. The extent to which this is the case, depends on the degree of truncation."
-p.14 l.384-392: make a graph? Also please check that you don't need to repeat information already given previously or to anticipate.
There is some repetition of earlier info at the beginning of this paragraph. One example: before the sentence "The instrument and representation error are taken from user input, the model error can either be specified by the user or estimated from a sensitivity analysis." we say
"Equation 5 states that the observational error consists of an instrument error, a model error and a representation error." This is intended to make the paper more readable, given the large amount of information in the paper, the reader might not remember everything from earlier sections. Moreover, this repetition does not take a lot of space.

It is not clear to us how a graph would clarify this portion of the text.

-p.14 l.395-p.15 l.419: "it will be shortly explained here": not necessary if it is the same as Chevallier et al. (2007), only detail the differences if any.

We understand the view of the reviewer, who wants to make the paper more concise. However, we would like readers to more or less understand how ICLASS works, without having to read other papers. This paper will also serve as a reference paper to which future studies using ICLASS can refer. We therefore would like to keep this crucial information in the paper.

*Output*

-p.15 l.422: "in ICLASS": what is the difference with the general definition of the chisquare?
The denominator differs depending on the situation, see Michalak et al. (2005). There is also difference between $X^2$ and $X^2_r$, e.g. compare Meirink et al. (2008) with Michalak et al. (2005). Our variable is $X^2_r$, we have adapted this.
-p.15 l.426: what does "default" mean? That the user can choose otherwise?
Yes, as is done in our OSSE example, in this case the cost function is only determined by the model observation fit
-p.16 l.412-452: a lot of this is generally known and used. Please keep to what is particular to ICLASS. Maybe also use tables.

We have now added a table in appendix with the output variables defined in this section. However, as the text also includes the employed formula and explanation, we cannot simply replace the text with this table.

-p.16 l.453- p.17 l.464: please use a graph or a list of a table.

Although this is in itself a good suggestion, there is a sequence here, with accompanying text in between. It is not clear to us how a graph, list or table would clarify and shorten this portion of the text.

*Technical details of the code*

-p.17 l.467seq: here again, please use a graph or a list or a table
In response to this comment, we have now used a list.
-p.17 l.477: "can easily be adapted": wouldn't netcdf be easier to use than pickle?

Thanks for this suggestion. In my (Peter) own experience netcdf is very useful for storing arrays with several dimensions (e.g. latitude, longitude,time). What we do with pickle here is merely to store the full Python objects so they can be loaded again later. Those objects are diverse, I think it might be more work to read/store these using netcdf.

*Adjoint model validation*

-p.17 l.480 - p.18 l.506: this is the general theory and must be removed.
See reply to next comment
-p.18 l.509 - p.19 l.519: same remark.
These sections seem important to us, as it provides a validation of the extensive adjoint code, with an example. The adjoint test and gradient tests are indeed common tests, yet the exact formula for the gradient test used here is, to our knowledge, not occurring in many places in literature. One other reviewer suggested us to extend this chapter with a table showing results of the gradient test. Presenting the results of the gradient test without including the formula of the test and a little explanation might not be the best solution. Given that this paper also serves as the reference paper for ICLASS, we think it can be useful to include the information on how the adjoint code was validated.
-p.19 l.522: how many is "the vast majority"? What about those that don't pass? What does "executed in this file" mean? How could you deal with numerical noise?

The file we talk about is a Python script, when running the file, a lot of tests are 'executed'. There is a default configuration of this file, but the user can adapt which sets of tests to run, as well as the model configuration and the number of time steps tested. The tests also involve random numbers used in formula 30, therefore the resulting output of the adjoint tests is slightly varying as well. Last time we ran the adjoint and gradient tests over multiple timesteps, we had two failing tests (on a total of more than 600 tests), one adjoint test that fails and one gradient test. The adjoint test resulted in a value in equation 30 of $3 \times 10^{-12}$. The part of the code tested involves a while loop, which might introduce extra numerical noise. The failing gradient test results in a value of -2.148466824970594e-97 using the tangent linear (LHS Eq. 28), while it results in a value of 0 using finite differences (RHS Eq. 28) with alpha=1e-5 or 1e-6 or 1e-7 or 1e-9 or 1e-12. Although this is labelled as a failure by our code, numerical noise is a likely explanation.

Additionally, besides the gradient and adjoint tests over multiple timesteps, we have tests for every separate module of CLASS, where we test more of the code. Some of these tests result in a reported failure when ran, they however require closer inspection. Looking at the following example output for testing a variable called 'fxdif_part1':

dfxdif_part1 :

7.847354016599084e-09 (finite difference output for first value of alpha)

7.844627725184239e-09 (finite difference output for second value of alpha)

7.845113447757512e-09 (…)

7.820133429703446e-09

4.163336342344337e-09

tl :7.844681884985882e-09 (this is the tangent linear output)

GRADIENT TEST FAILURE!! dfxdif_part1

Several increasingly smaller values for alpha (eq 28) are tested here consecutively. However, looking at this output it seems that it is merely numerical noise, since only for the smallest tested value of alpha (1e-8) the tangent linear output strongly diverts from the finite difference output.

Testing several values of alpha in the gradient tests (as we do) can be seen as a strategy to deal with numerical noise. Adjoint tests can also be ran multiple times with different random numbers.

*Invese modelling validation: OSSE*

The tests described in this section are useful but they are only very basic tests since, for example, I understand that four out of five are set-up without any perturbations of the observations. The error statistics are not described: are they the "true" ones or are they mis-specified in some tests? The convergence criteria are not discussed, which makes it difficult to compare the tests. Moreover, without the posterior uncertainties, the results are not complete nor comparable.
The tests are indeed basic, they were intended to show the capacity to fit observations and find good parameter values, not to test the statistics. Note that the posterior uncertainties are only estimated when performing an ensemble of optimisations (Monte Carlo approach). Given the focus of these basic OSSEs, we did not use an ensemble. We will add an OSSE that focuses more on statistics and the bias correction. This OSSE will have mis-specified error statistics. We also added a table that quantifies the fit for the OSSE with the perturbed obs, and lists the employed measurement error standard deviations.

The procedure for those simple OSSEs is described in the beginning of the section: "We first run the model with chosen values of a set of parameters we want to optimise. A set of model output data from this simulation then serve as the observations, while the parameters used to create these observations are referred to as the "true" parameters. Then we perform an optimisation using these observations, starting from a perturbed prior state vector.".

There is indeed only one experiment with perturbed obs, in this experiment we perturb the obs using the specified measurement error standard deviation, see line 540. The model and representation errors are set to 0 in all experiments. (we added this info to the paper now)
-p.19 l.527: what does "constructed adjoint" mean?
The adjoint they have constructed (coded). We left the word out now to avoid confusion.
-p.19 l.530: 5 experiments is a bit too small a number for actual validation of a code.
True, we will add another OSSE, focusing on statistics and bias correction. But note validation of the adjoint code is also done in chapter 8.
-p.19 l.535-536: keeping out the background makes them very basic tests.
Indeed, but in these first four tests, the goal is to test the capacity to find back the true parameters. We have added some info to the text: "In the cost function, we do not

include the background part, to make sure that it is possible to find back the "true" parameters. This is because the background part of the cost function implies a 'penalty' for deviating from the prior state. This penalty implies that, when the model is run with the true parameters, the cost function would still not be zero, and the minimum of the cost function might be shifted."
In the new, more complex OSSE that we will present, the background part will be included
-p.19 l.547-548: you can quantify the influence of a state parameter with the adjoint. Indeed, but given that the OSSEs are simple and do not require a lot of computation time there was no need to test this in advance
-p.20 l.549: what is "a very good fit"? How can it be defined without the uncertainties? The basic OSSEs were intended to show the capacity to fit observations and find good parameter values, not to test the statistics (we will add an OSSE with a stronger focus on statistics). When looking at figure 3 it is clear that the model matches the observations very well, the difference between observations and model output is very hard to see by eye. For the prior this is by far not the case. Even though we are not quantitatively describing the fit, one can call this "a very good fit" in our opinion. Adding a quantitative measure of fit such as the root mean square error does not add much here in our opinion, neither does the cost function (partly determined by observational error standard deviations that are simply chosen by us). For the OSSE with perturbed observations, we have added a table showing the prior and posterior RMSE.
-p.20 l.552: "a more complex problem": the problem is not well defined but is it complex?
What we mean here is more complex relative to the tests described before, because of an increased number of state parameters (all state parameters have a prior value different from the 'true' value).
-p.20 l.553-554: if the parameters have no influence on the cost function (which can be checked with the adjoint), then the inversion is useless.
Indeed, but given that the OSSEs are simple and do not require a lot of computation time there was no need to test this in advance. From the result of the OSSE in table 1, we can see that the optimised parameters are all different from the prior parameters, which indicates that the cost function is sensitive to all parameters, proving our hypothesis.

-p.20 l.554-55: the parameter interdependency issues are not the only ones which may arise in this case.
Since we can give a clear example of the interdependency issue that arises, we choose to mention that. But this might indeed not be the only possible issue that could have arisen.
-p.21 l.567-568: why does the analytical gradient perform better than the numerical calculation?
It is a very non-linear model, having exact gradient calculations seems to improve performance.
-p.21 l.573: the framework finds a minimum, not the minimum of the cost function.
The exact true parameter values would give a cost function of 0 (because of the set-up of the discussed OSSEs), since the framework approaches the true parameter values very well, the framework approaches the global minimum (in these simple OSSEs).
-p.21 l.575: what is "a good fit"?
Qualitatively, the fit is not as good as in Figure 3, since the observations are now perturbed and impossible to exactly reproduce with the model. We agree that some more quantitative info can be useful here, we therefore added a table that shows that the root mean squared error lies close to the prescribed measurement error standard deviation. Given that these measurement error standard deviations were used to create the random perturbations for the observations, this confirms the good fit quantitatively.
Figure 3: what about the uncertainties?

In this figure we did not include the specified observational error standard deviations. The first four simple OSSEs are about finding back the true parameters, in our opinion, the (artificial) observational error did not seem very important to include. In Figure 4 however, where observations are perturbed, we do include error bars.

*Application example*

-p.21 l.584-586: this is strangely put: observations are "derived" from other observations
What we want to say here is that we compute certain observation variables from other variables in the dataset. For instance, specific humidity is obtained using dew point temperature etc. This happens before assimilating the observations.
- it looks like you use the same word for actual observations i.e physical variables that are measured and "observations" in the modelling framework i.e. variables of which the model computes an equivalent for comparison.
Throughout the paper, we intended to use 'observations' for physical variables that are measured, irrespective of whether they are assimilated or not. The variables computed by the model, to be compared with observations, are in vector H(x), we never intended to indicate the contents of this vector as observations. If we have done so otherwise by mistake, please let us know the line number so we can adapt it.
-p.21 l.587: what are the "non-state parameters"? Put them in the table?
The CLASS model has over 50 parameters, putting them all in a table will take a lot of space. The user decides on which parameters to include in the state and which ones not.
-p.22 l.589: "the detailed settings on chosen model errors, etc" are crucial information, I think they should be put in the main text or at least in an appendix.
We will add the error specifications to either the tables in main text (prior), or to appendix (the others)
-p.22 l.591: 591-592: what about the uncertainties of the prior and posterior? Without them, "a much better fit" cannot be defined. Moreover, fitting the observations is not the reason why inversions are run. The aim is to reduce the uncertainty on the optimised parameters, which is not shown in the figures.
For the prior, we did not include the uncertainty in Table 3, we will add that. For the posterior uncertainty, we would like to point to table 3 and figure 9. In the last column of table 3, we show the posterior standard deviation of every parameter. In Figure 9 we picked out 2 parameters and show the full posterior pdfs. In figures 5 and 6, we included the observational error standard deviations of the shown subset of observations.
-p.22 l.594: what could be done about the non-optimal error specifications? Lacking information on the computing cost of the inversions, it is not possible to assess whether a number of error set-ups could be tested.
Here, we mention non-optimal error specifications as a possible reason why chi-squared is slightly low. As common in inverse modelling, exactly estimating all uncertainties is a difficult task. Testing different error set-ups is possible on an HPC-cluster (the application example uses an ensemble of 174 perturbed members), but is not the focus of this application example.
-p.22 l.595 - p.23 l.597: why are some observation streams different from the others with respects to the variance?
This is about the ratio of model and observation variance. There are some observation streams were this ratio is far from 1, and the model thus does not reproduce the variance well (this is also not always desired, the observations are influenced by measurement errors). There is no reason why this ratio would always be the same among observation streams, the model can have more difficulties reproducing one observation stream then another. Further analysis would be needed to determine why some observation streams are fitted better than others, but this section is just an application example.
-p.23 l.599-600: I don't understand the link between "the model also has a closed energy balance" and the "good fit".
This is indeed not very clear, our reasoning is as follows:

The energy balance equation is given by Eq 8 in the paper: residual = Rn – (H + LE + G)
From figure 7 it is clear that (generally) the model, both prior and posterior, has a higher sum of H+LE than what the uncorrected observations show.
The correction on the H and LE observations is based on measured net radiation (see eq 8, 9 and 10). The sum of H+LE in the model is also based on net radiation, which the model calculates. Thus, if we assume that the difference between measured and modelled soil heat flux (G) will be small in absolute numbers, and we assume measured and modelled net radiation to be comparable, it would mean that the sum of H+LE in the model would correspond to the sum of H+LE in the corrected observations quite well (although there is usually also a small linearisation error in the model fluxes, making energy balance closure imperfect). This explains the link between the closed energy balance and the good fit.

We now adapted the text to improve the clarity.
-p.23 l.601-602: this sentence is not clear.
We want to say here that the error in the energy balance in the measurements is relatively large, by comparing the errors (LHS equation 8) to the measured sensible heat flux. The term 'measured sensible heat flux' is however slightly ambiguous because we 'correct' observations, so we added between brackets 'without applying Eq. 9'.
-p.23 l.603 - p.24 l.604: aren't there any data available to check the cumulus clouds or the drop in net radiation?
We have adapted the text here, a colleague provided us with a MODIS satellite image, showing that high clouds are a more likely explanation. We also see a drop in incoming shortwave radiation around noon of that day:

[Figure]

-p.24 l.608-609: is this assumption very limiting?
We left out the sentences 'Such a bias can be accounted for in the framework, by adding a scaling factor for the surface CO2 flux observations to the state. This however implies the assumption that the bias takes the form of a fixed fraction of the observed surface CO2 flux.'
Regarding the question on whether this assumption is very limiting, this question cannot be readily answered by us, see e.g. Liu et al. (2006) and Deventer et al. (2021) for a discussion of CO2 flux biases.
-p.25 l.614: "we shortly return to this later in this section": avoid this with a more explicit division in subsections?
Thanks for this suggestion to improve readability. We have now divided the section about the application example into several subsections, and refer to the specific subsection.
-p.26 l.627-628: this sentence calls for a discussion on the impacts of the misspecification of prior errors.
This analysis is about correlations between posterior parameters. Concerning the importance of correctly specifying the prior errors: we think that this is a well-known problem in inversions.
The impact of the prior in this example will be relatively modest, given that the nr of observations (multiplied with their respective weights) is about 10 times larger than the number of state parameters (although of course this also depends on the specified error variances).

-p.26 l.630: what does "relatively strongly" mean?

We have changed the sentence into "it can be noted that the advCO2 parameter is relatively strongly correlated with both the $\Delta CO2$ and $\gamma CO2$ parameters (Fig. 6: corr. = -0.65 and -0.8 respectively)"

-p.26 l.632: what are these differences?

This is about differences in how entrainment is handled. From the paper of Casso-Torralba et al (2008): "Observations of thermodynamic variables and CO2 mixing ratio as well as vertical profiles of the turbulent fluxes are used to retrieve the contribution of the budget terms in the scalar conservation equation. On the basis of the daytime evolution of turbulent fluxes, we calculate the budget terms by assuming that turbulent fluxes follow a linear profile with height"

Their estimate of advection we compare with (their Figure 9), is obtained as a residual budget term. The other terms in their budget are storage and flux divergence. The latter one includes entrainment, although they do not explicitly calculate it for Figure 9.

In our case, the entrainment fluxes are calculated as follows: First, the buoyancy entrainment flux is taken as a fixed fraction of the surface flux of this quantity, to which entrainment driven by shear can optionally be added. From this virtual heat entrainment flux, an entrainment velocity is calculated. The entrainment flux for a specific scalar (e.g. CO2) is than obtained by multiplying the entrainment velocity with the size of the (inversion-layer) discontinuity for the respective scalar.

-p.27 l.637: is 0.05 the average?

Indeed, the text states "The average absolute value of difference between the non-diagonal matrix entries when using the subsample and the non-diagonal matrix entries when using the full successful perturbed ensemble amounts to 0.05"

To explain the text: This is about the differences in the correlation matrix when using the full successful perturbed ensemble compared to when using a subsample. We take the absolute value before averaging, otherwise positive and negative differences can compensate each other. We only look at non-diagonal entries of the matrix, since the correlations on the diagonal are always 1.

-p.27 l.638: what does "reasonably robust" mean?

It is difficult to exactly pinpoint the number of members needed to get a good estimate of the correlation matrix. But we showed here that, when using only 75 of the 150 successful members, the non-diagonal matrix entries change on average by only 0.05 (in abs value), which is not a lot. This gives a certain level of confidence that 150 is enough, but hard to exactly quantify how much confidence.

-p.27 l.642: is there is "no clear reduction in uncertainty", then the inversion was useless. It may not have failed mathematically but its results are not interesting as such. (The fail may be interesting to ask for more observations.)

We partly agree with this statement. One could say that, for the $\gamma_q$ parameter, the inversion was useless, as the posterior is about as uncertain as the prior. This is however just one parameter, in the example 14 parameters are optimised simultaneously, e.g. the $adv_\theta$ parameter in Figure 9 does show a clear reduction in uncertainty.

-p.27 l.642 - p.28 l.643: this is not clear to me.

The sentence reads "The wide posterior pdf implies that similar results can be obtained over a relatively wide range of $\gamma_q$, possibly by perturbing other parameters with a similar effect".

It is important to realize here how the posterior uncertainties were obtained. This was done by running an ensemble in which both the prior and the model-data mismatch was perturbed. This results in ensemble of posterior states, from which uncertainties were derived (using only members with post chi² <=2).

A wide posterior pdf means that there was quite some spread in the posterior values of $\gamma_q$. Each posterior ensemble member obtaining a good chi² can be seen as providing a similar result (in terms of its fit). Thus, similar results can be obtained over quite a range

of $\gamma_q$ values. Next to that, as the correlation matrix has shown us, there are correlations among parameters, also involving $\gamma_q$. Thus, e.g. a large value of $\gamma_q$ can be largely compensated by a small value of another parameter, explaining the last part of the sentence.

-p.27 l.647-654: this should come sooner in the text.

We will mention this earlier in the application example, in a separate subsection.

Tables 1 and 3: what about the convergence criteria? What about the uncertainties (prior and posterior)?

Table 1 is about the simple OSSEs. Prior uncertainties were not used here, and posterior uncertainties not calculated (we did not run an ensemble), as the focus was on the capacity to obtain good parameter estimates.

We will add the prior uncertainty to Table 3 (the application example), the posterior uncertainty is already included (column Post. st. dev.)

Regarding convergence criteria, this is rather complex: There are multiple ways in which the optimisation can come to a stop. The SciPy algorithm optimize.fmin_tnc can consider an optimisation as converged (we use the default tolerances, see https://docs.scipy.org/doc/scipy/reference/generated/scipy.optimize.fmin_tnc.html). The ICLASS user can however specify a desired threshold of the cost function. In case the optimize.fmin_tnc considers an optimisation as converged and the threshold is not yet reached, the optimize.fmin_tnc algorithm will then be restarted from the best state so far, the maximum number of times a restart will be performed is also given by the user.

It can also happen that a maximum number of function evaluations is reached within the optimize.fmin_tnc algorithm, before an optimisation is considered as converged by the algorithm.

In case an optimisation shows very little change in the cost function over a certain number of iterations, the optimize.fmin_tnc algorithm can be terminated (depending on a setting) and possibly restarted (criteria as above).

A model crash can also lead to an early termination, in this case no restart is attempted. The user can control the convergence criteria of the optimization to a certain extent, through settings in the standard tnc routines and by specifying an optional desired cost function threshold and the maximum number of restarts.

Table 2: how much are the sensible and latent heat flux observations corrected?

Here we would like to refer to Figure 7, which shows the original and corrected observations.

Figures 4,5 and 6,7: what about the uncertainties?

The observational error and measurement error standard deviations are shown with error bars in these figures (for Figure 4 measurement errors equal observational errors). For the application example we also estimate posterior uncertainties on the optimised parameters using a Monte-Carlo approach, shown in table 3 and (for some params) in Figure 9. The uncertainty in the state parameters leads to an uncertainty in model output, but this is not readily available in ICLASS. In principle, one could run the model using the obtained posterior parameter values of a successful ensemble member, and repeat this for all successful ensemble members. From this ensemble of model output, uncertainty estimates on the model output could be made.

Figure 9: what about the Gaussian assumption?

We only assume the prior to be a (truncated) Gaussian, we do not make any assumptions on the shape of the posterior pdfs (nonlinear model), except that we place hard outer bounds on some parameters. Regarding the prior, note that the prior distribution is determined from the sample of priors in the ensemble, which has a component of randomness. This explains why the **prior** pdfs do not have a perfect (truncated) Gaussian shape.

*Concluding discussion*

-p.28 l.657-658: general theory of inversions.
Indeed rather general, but in our opinion it is useful to indicate these advantages/limitations of the framework, especially for those less familiar with inverse modelling.
-p.28 l.659: what could the more advanced error estimation methods be?
For instance, the measurement error could be more based on instrument errors belonging to the used devices and representation error could take spatial variability in measurements into account. For e.g. $CO_2$ mixing ratio errors, the residual standard deviation of flask samples around a smooth curve fit could be used (Michalak et al., 2005). Model error variance estimations might possibly be obtained from analysing the model behaviour compared to precise observations in specific situations, but in practice this might prove very hard. In literature, more methods can be found; e.g. Michalak et al., 2005.
-p.30 l.679-680: the correction of biases is a very complex topic. It is often done outside the inversion framework. a bias correction scheme such as tested here probably cannot be expected to deal completely with the issue.
We fully agree with this statement, the bias correction scheme is useful but cannot correct for all possible complex bias patterns. We have added the following to the concluding discussion: "The correction of biases is however a very complex topic. There are limitations to the level of complexity that our scheme can handle, ICLASS cannot be expected to deal completely with all bias issues."
- please add information on the computation costs.

The section on the computation costs in the application example will be extended, it is also turned into a separate subsection.

**Technical comments:**

-p.3 l.76 and others: why is the term "adjoint" in italics?
We removed the italics at line 76, we now only keep the very first occurrence of adjoint in the introduction in italics, for emphasis.
-p.6 l.156: what are the (-)? Also found elsewhere.
Between brackets we indicate the units of the variable, in this case the variable is dimensionless. In the latex source code we wrote (\unit{-})
-p.16 l.436: "similar to" instead of "similar as"?
We have adapted the sentence

**References**

Brasseur, G. and Jacob, D.: Inverse Modeling for Atmospheric Chemistry, in: Modeling of Atmospheric Chemistry, pp. 487–537, Cambridge University Press, Cambridge, https://doi.org/10.1017/9781316544754.012, 2017.

Casso-Torralba, P., de Arellano, J. V. G., Bosveld, F., Soler, M. R., Vermeulen, A., Werner, C., and Moors, E.: Diurnal and vertical variability of the sensible heat and carbon dioxide budgets in the atmospheric surface layer, Journal of Geophysical Research Atmospheres, 113, https://doi.org/10.1029/2007JD009583, 2008.

Deventer, M.J.; Roman, T.; Bogoev, I.; Kolka, R.K.; Erickson, M.; Lee, X.; Baker, J.M.; Millet, D.B.; Griffis, T.J. Biases in open-path carbon dioxide flux measurements: Roles of instrument surface heat exchange and analyzer temperature sensitivity. Agric. For. Meteorol. 2021, 296.

Doicu, A., Trautmann, T., and Schreier, F.: Numerical Regularization for Atmospheric Inverse Problems, Springer Praxis Books in environmentral sciences, https://doi.org/10.1007/978-3-642-05439-6, 2010.

Foken, T.: The Energy Balance Closure Problem : An Overview, Ecological Applications, 18, 1351–1367, http://www.jstor.org/stable/810 40062260, 2008.

Friend, A. D.: Modelling Canopy CO2 Fluxes: Are 'Big-Leaf' Simplifications Justified?, Global Ecology and Biogeography, 10, 603–619, http://www.jstor.org/stable/3182690, 2001.

Jacobs, C.: Direct impact of atmospheric CO2 enrichment on regional transpiration, Ph.D. thesis, Wageningen University, 1994.

Liu, H., Randerson, J. T., Lindfors, J., Massman, W. J., and Foken, T.: Consequences of incomplete surface energy balance closure for CO2 fluxes from open-path CO2/H2O infrared gas analysers, Boundary-Layer Meteorology, 120, 65–85, https://doi.org/10.1007/s10546-005-9047-z, 2006.

Meirink, J. F., Bergamaschi, P., and Krol, M. C.: Four-dimensional variational data assimilation for inverse modelling of atmospheric methane emissions: Method and comparison with synthesis inversion, Atmospheric Chemistry and Physics, 8, 6341–6353, https://doi.org/10.5194/acp-8-6341-2008, 2008.

Michalak, A. M., Hirsch, A., Bruhwiler, L., Gurney, K. R., Peters, W., and Tans, P. P. (2005), Maximum likelihood estimation of covariance parameters for Bayesian atmospheric trace gas surface flux inversions, J. Geophys. Res., 110, D24107, doi:10.1029/2005JD005970.

Raoult, N. M., Jupp, T. E., Cox, P. M., and Luke, C. M.: Land-surface parameter optimisation using data assimilation techniques: The adJULES system V1.0, Geoscientific Model Development, 9, 2833–2852, https://doi.org/10.5194/gmd-9-2833-2016, 2016.

Rodgers C. D. (2000) Inverse Methods for Atmospheric Sounding, World Sci., Tokyo.

Ronda, R. . J. ., de Bruin, H. . A. . R., and Holtslag, A.: Representation of the Canopy Conductance in Modeling the Surface Energy Budget for Low Vegetation, American Meteorological Society, 40, 1431–1444, https://www.jstor.org/stable/10.2307/26184869, 2001.

Tarantola, A.: Inverse problem theory and methods for model parameter estimation, Society for Industrial and Applied Mathematics (siam), Philadelphia, USA, https://doi.org/10.1137/1.9780898717921, 2005.

Vilà-Guerau De Arellano, J., Van Heerwaarden, C. C., Van Stratum, B. J., and Van Den Dries, K.: Atmospheric boundary layer: Integrating air chemistry and land interactions, Cambridge University Press, 2015.

**Reply to reviewer 3**

**Introduction and bibliography**

The introduction is not well-balanced and lacks pieces of bibliography. The reader would expect an extensive "review" of what has been done in parameter estimation in land-atmosphere exchanges, and not only with simple models. For instance, there
has been some work on parameter estimations with full-physics models, such as ORCHIDEE or JS-BACH. The advantages vs drawbacks of simple models such as CLASS, compared to full-physics models should be more thoroughly presented. The scientific "ecosystem" of the present study should be better presented. There is a full field of studies using data assimilation, machine learning, etc.

To place the variational framework of this paper in comparison with other efforts in the scientific community, we now added a paragraph linking parameter estimation in land-surface models in other studies with ICLASS. We briefly discuss advantages/disadvantages of CLASS vs models with more complex physics. An important point we make is that the fully coupled land-atmosphere in ICLASS helps to infer land surface characteristics from atmospheric observations, something that is often not the focus of other variational frameworks.

The balance between giving only hints or extensive details is also clumpsy. For instance, in paragraph p.2 l. 34-48, the authors start giving information on the model itself compared to other models, but without going to the details. What is an "extensive set of observations"? What observations are better used than other models?

About this example: CLASS has both a land-surface representation and a mixed-layer representation, which is an advantage compared to other uncoupled models. This also means that it can use information from a variety of observation types, as CLASS models both fluxes and mixing ratios. We cannot list all possible obs types here, but think of temperatures at multiple heights, humidity at multiple heights, $CO_2$ mixing ratios, heat fluxes and $CO_2$ fluxes, … We have changed the text into "A model like CLASS, containing both a mixed-layer and land-surface part, can be used to fit an extensive set of observation streams simultaneously." We are not claiming that CLASS uses observations in a better way than other models would do, but we indicate that many studies only use a small part of the available observations. The example study we refer to applies CLASS without an inverse modelling framework, which makes it difficult to include a lot of observation types.

**Energy balance and conditions of applicability of CLASS**

The CLASS model is a simplified model with all its benefits and drawbacks. In particular, what are the conditions of applicability of CLASS? The authors mention "golden days" several times in the text. What are these? How frequent are they? If there is only a few such days per year, the model is not really suitable for purpose…

About the energy balance and further assumptions, it is not fully clear what is the domain of applicability of such assumptions. In particular, the advection and entrainment in the model are extremely simplified. What values and variables are used to constrain the processes?

At line 32 we define golden days: "Those are days in which advection is either absent or uniform in time and space, deep convection is absent, and sufficient incoming shortwave radiation heats the surface allowing for the formation of a prototypical convective boundary layer."

We understand that this raises questions about the frequency of these days etc. We therefore added the following: "The model performs best during the convective daytime period, the assumptions on advection etc. should be valid for the whole modelled period. Since the model performs best on fair-weather days, the absence of deep convection etc. should ideally hold on a spatial scale large enough that it does not influence the model simulation location. In practice, days are often not 'ideal', e.g. a time-varying advection can be present. This does not necessarily mean the model cannot be applied to that day, but, performance is likely to be worse."

We want to stress also that it is not our intention to provide a complete detailed description of the CLASS model itself, we already included about 1 page of info on CLASS itself in the paper. CLASS is an existing model, successfully used in several studies. For details about the model, we refer to Vilà-Guerau De Arellano et al. (2015). In the introduction, we also include the following text "This and similar models have been applied frequently, e.g. for understanding the daily cycle of evapotranspiration (van Heerwaarden et al., 2010), studying the effects of aerosols on boundary layer dynamics (Barbaro et al., 2014), studying the effects of elevated $CO_2$ on boundary layer clouds (Vilà-Guerau De Arellano et al., 2012) or for studying the ammonia budget (Schulte et al., 2021).".

See also https://classmodel.github.io/publications.html. There has also been a 2019 GMD paper employing (an adapted version of) CLASS: (Wouters et al., 2019, https://doi.org/10.5194/gmd-12-2139-2019)

Regarding the question "In particular, the advection and entrainment in the model are extremely simplified. What values and variables are used to constrain the processes?":

Advection is indeed represented in the model in a simple way. Advection of e.g. temperature is given by a single parameter. To constrain this parameter, traditionally the model is tuned by hand to available observations such as temperature and possibly mixed layer height. This is where ICLASS offers a great improvement, as it allows to more objectively use all the available observations to optimise this parameter.

Regarding entrainment, there was a mistake in the text, we now write:
"Above the mixed layer a discontinuity occurs in the scalar quantities, representing an infinitely small inversion layer. Above the inversion, the scalars are assumed to follow a linear profile with height in the free troposphere (Fig. 1). The entrainment fluxes are calculated as follows: First, the buoyancy entrainment flux is taken as a fixed fraction of the surface flux of this quantity (Stull, 1988, p 478), to which entrainment driven by shear can optionally be added. From this virtual heat entrainment flux, an entrainment velocity is calculated. The entrainment flux for a specific scalar (e.g. $CO_2$) is than obtained by multiplying the entrainment velocity with the size of the (inversion-layer) discontinuity for the respective scalar."

Section 3.1 and mathematical notations
Please make your mathematical notations consistent with the rest of the community.

- prior vector: $\mathbf{x}^b$: The author should explicitly write it somewhere, with all its sub-components (bias, parameters, inputs, etc.)

- posterior vector: $\mathbf{x}^a$

- full observation operator: $\mathcal{H}$

- adjoint sensitivities are usually noted as: $\delta S_{win}^*$

Different communities prefer different notation. We based our notation on Brasseur and Jacob (2017), and their notation is to a large extent based on Rodgers (2000).

The components of the state vector are described in section 3.2, there is also a table added now describing all inverse-modelling variables included in chapter 3, including those relating to the state vector. There are more than 50 parameters that can be optimised, we cannot list them all in the paper, this is done in the manual. Choosing which parameter to optimise and which ones to keep fixed (and thus what is in the state) is eventually up to the user, this varies with the study to be performed with ICLASS.

Overall, Section 3.1 is very hard to understand. It is not clear at all what is optimized or not.

In section 3.2 we give an overview of the types of parameters that can be optimised in ICLASS, thereby splitting the state vector into a bias-correction part and a model parameter part. There are more than 50 parameters that can be optimised, we cannot list them all in the paper, this is done in the manual. Choosing which parameter to optimise and which ones to keep fixed is eventually up to the user, this varies with the study to be performed with ICLASS.

The section gives some general information about the inversion framework, but does not go to the necessary level of details about what exactly is in each mentioned vectors and operators. The dimension and content of all operators and matrices should be detailed.

We now added a long table in the appendix that list the dimensions, units and a short description of most variables of this chapter. We try to describe the vectors in the main text as well where they are introduced.

The weights on observations or "regularization factors" are clumsy and not justified. If one observation is less worthy than another, then the uncertainty should just be scaled up, with no need for an extra complicated parameter.

Indeed identical changes can be made to the cost function by adapting weights or changing the observational error variances. However, the observational error standard deviations are also used in the ensemble for estimating posterior errors (see section 5.2). When the observational errors are no longer realistic due to inflating/deflating these errors, the observations are not properly perturbed anymore. This problem is avoided when using weights. The latter can be used, for example, when you have 15 temperature observation streams, but only one CO2 observation stream. In this case adding a weight of 1/15 to the temperature observation streams can make the observation streams more balanced, while keeping a realistic error for the observations. We have added an additional sentence to the text of the paper: "In principle, the observational error variances could also be adapted for this purpose, but by using weights we can keep realistic error estimations (important for Sect. 5.2)."

Equation (6) is too implicit. The author should fully detail the "background" term, including what they optimize or not.

Equation 6 gives the cost function as used in ICLASS. The first term of this equation is the background term, wherein vector x is the state, containing the variables to be optimised (and $x_A$ is the prior state). In section 3.2 we give an overview of the types of parameters that can be optimised in ICLASS, thereby splitting the state vector into a bias-correction part and a model parameter part. There are more than 50 parameters that can be optimised, we cannot list them all in the paper, this is done in the manual. Choosing which parameter to optimise and which ones to keep fixed is eventually up to the user, this varies with the study to be performed with ICLASS. Similarly, for the a-priori error covariance matrix $S_A$, the user chooses the variances/covariances, and the size of this matrix varies with the chosen state vector size.

Uncertainties and OSSEs

Please provide extensive details on the uncertainties you specify for the inputs and parameters and some justification for the corresponding uncertainties. In particular, for parameters, the normal distributions are not necessary the most obvious choice. This should be justified and detailed.

The tests are indeed basic, they were intended to show the capacity to fit observations and find good parameter values, not to test the statistics. The prior information is not used in these simple OSSEs, thus the prior uncertainty is irrelevant in these simple tests. We will add an OSSE focusing more on statistics, were we will provide the uncertainties. The employed observational error standard deviations for the OSSE with perturbed observations is shown in Figure 4, and we added these now in a table as well. Note that the form of the cost function does not allow for using e.g. uniform priors. However, as is mentioned in section 5.1, it is possible to perturb parameters that are not part of the state, using a "normal", "bounded normal", "uniform" or "triangular" distribution.

The OSSEs are rather simple and do not fully allow to validate the model. More OSSEs should be made more systematically to show what is the influence of a given parameter in a given set-up. The author can perturb a parameter but not optimize it, etc.

Since the forward model CLASS is an existing model, successfully used in other studies, we do not intend to validate the CLASS model itself, or test its sensitivities to parameter values. The OSSEs are rather intended to focus on the parameter optimisation framework. But the OSSEs are indeed rather basic, and we will add a more involved OSSE, taking also posterior uncertainties and bias correction into account.

Besides, I may have missed the information, but I have the impression that the bias correction is not evaluated in the OSSEs. This should be added.

The bias correction was indeed missing (although the bias-correction was to some extent tested in the application example), we will add an OSSE that tests the bias correction.

Regarding the posterior uncertainties, having truncated Normal distributions means that the minimum of the cost function is the node of the posterior distribution, which is not the mean or

median, contrary to full normal distributions. Therefore, the authors should give further details on how the compute and analyze posterior distributions.

We only assume the prior to be a (truncated) Gaussian, we do not make any assumptions on the shape of the posterior pdfs (nonlinear model), except that we place hard outer bounds on some parameters. We use a Monte Carlo technique to sample the posterior pdfs, see section 5.2 and Fig. 9. We have slightly adapted section 5.2 to make this more clear.

Details on the model

There is critical information missing about the CLASS and iCLASS models. Some of this information is given in the documentation of iCLASS, but not comprehensively. The reader cannot be expected to read the non-reviewed documentation to understand the article and how the adjoint is built. In particular, there should be full details on the inputs and parameters of the CLASS models. What are the resolutions of each inputs? Where do they come from? Are they given by in-situ measurements? Meteorological forcing fields?

Similarly, what are the exact outputs of the model? How the output is compared to observations. Finally, what is computed by the model? And what is given as inputs?

We understand that, without background knowledge on CLASS, these questions arise. However, as mentioned earlier in this document, CLASS is an existing model, successfully used in several studies, although we made some changes to the model (listed in the manual). We give about 1 page of information on the model itself in the paper, for details about the model itself, we refer to Vilà-Guerau De Arellano et al. (2015). In the introduction, we also include the following text "This and similar models have been applied frequently, e.g. for understanding the daily cycle of evapotranspiration (van Heerwaarden et al., 2010), studying the effects of aerosols on boundary layer dynamics (Barbaro et al., 2014), studying the effects of elevated CO2 on boundary layer clouds (Vilà-Guerau De Arellano et al., 2012) or for studying the ammonia budget (Schulte et al., 2021).".

CLASS requires a set of input parameters to be chosen, e.g. free-tropospheric lapse rates of temperature, specific humidity, initial CO2 mixing ratio in mixed layer, but also land-surface-model parameters such as roughness length for momentum, leaf area index, and initial soil moisture content of top layer. Where the user obtains these inputs from is up to the user, this does not matter for ICLASS itself. The inputs can come from in-situ measurements, but e.g. reanalysis data might also be used. Note that the model is a slab model, it has no horizontal resolution, this simplifies the required inputs. The full list of input variables that can be included in the state is given in the ICLASS manual, the list is too long to include in the main text. We give a few examples of input parameters in section 3.1 and section 3.2.

We do not transform any model output into observation space, we directly compare the model output to observations. With the in-situ observations we used in the application example this was well possible, in case the user uses different observation types, he/she should take care to perhaps make the observations suitable for comparison to the model output.
Model output includes time-series of mixed-layer potential temperature, specific humidity, CO2 mixing ratio,…, but also heat fluxes, CO2 fluxes, Inversion strength,… The full list of output variables that can be compared to observations is given in the manual, it is too long to give in the main text. We give one example in section 3.1.

6 Superfluous sections and elements

The text is made hard to follow by numerous superfluous details.

For instance, section 4 is mainly made of a technical lecture on how to code an adjoint. This can be removed altogether.

In response to this valid comment, and a similar comment from another reviewer, section 4 is moved to the supplementary material.

Technical comments
1. p.1 l.9: replace "the core physics to model" by "the core physics to simulate"
Adapted
2. p.3 l.63: The example is rather a negative feedback but not an obvious non-linearity. There are probably better examples.
The example itself is indeed a negative feedback. A negative feedback can only occur in a non-linear model, proving the non-linearity. We tried to make the non-linearity more clear now in the text: "An important challenge for the optimisation framework is the strong non-linearity of the model. As an example, the change in mixed-layer specific humidity (q) with time is a function of q itself: a stronger evapotranspiration flux leads to an increased specific humidity in the mixed layer, which in turn reduces the evapotranspiration flux again (van Heerwaarden et al., 2009)." Another example we could think of is e.g. $CO_2$ uptake being a non-linear function of incoming radiation.
3. p.3 l.66: "Analytical" is ill-chosen and refers to analytical inversions in the inversion framework. The adjoint is simply needed to compute explicitly and efficiently the gradient of the cost function, without relying on, e.g., finite-element estimations
We understand the confusion with 'analytical inversions', we however talk about an analytical **gradient** of the cost function, not an analytical solution to the minimisation problem. The adjoint is a tool that helps us obtain an analytical gradient of the cost function. In our view, the two classes of methods for computing a gradient of any function is either 'analytically' or 'numerically', i.e. involving finite differences. The term 'analytical gradient' is also used in Raoult et al. (2016), see also Doicu et al. (2010).
4. Section 8: the validation of the adjoint using the gradient test and the test of the adjoint is really appreciated! The results of the test of the adjoint is generally reported as a N times the machine epsilon ($10_16$ in present machines)
We have updated the sentence: "When we evaluated Eq. (30) on this part of the code, the result was less than $1 \times 10{-15}$ (which corresponds to $10 \times$ machine precision), meaning that the test passes"
5. p.15 eq.20: $x_A$ is modified in the Monte Carlo.

Thanks for spotting this, we had not indicated this in the equation. We replaced $x_A$ now with $x_A^{\{p\}}$
, the p indicating perturbed.

6. p.15 eq.22: $\chi^2$ formula is wrong for two reasons. First the chi-square diagnostics can be applied only with normal distributions. Truncated-Gaussians break the diagnostics; but for not so truncated Gaussians, it may still be valid.
We indeed allow truncated Gaussians distributions for the prior parameters. In some cases this might indeed have a significant impact on the validity of the calculated $\chi^2_r$, we have added the following text to the paper: "Furthermore, as mentioned in Sect. 5.1, prior parameters can follow a

truncated normal distribution, violating the normality assumption. The impact of this depends on the degree of truncation, but also on the number of observations etc. It can lead to an ideal $\chi^2_r$ value diverting from 1."

Note that we call the variable the *reduced* chi-squared statistic now.

Second, the authors mixed two versions of the chi-square diagnostics: one from, e.g., from Michalak et al. 2005 (doi:10.1029/2005JD005970), the other from, e.g., Zupanski et al. 2006 (https://doi.org/10.1175/MWR3125.1). In one version the chi-square has a mean of $n$ (nb obs) and in the other $n+m$ (nb obs + parameters). As written in eq.22, the expected mean is $n$, or the authors compute the other version, but should explain more clearly what is done.

It is optional to include a background part of the cost function, usually the background part is included, but e.g. in the simple OSSEs it was not. When the background part is included in the cost function, we expect a posterior cost function of size (approximately) $n+m$ (see **), if it is not included we expect a posterior cost function of size (approximately) $m$. Therefore, as in both cases we want an optimal value of 1 for $\chi^2_r$, the denominator in eq 22 is taken as $n+m$ when the background part is included, and $m$ if it is not included, as mentioned in the paper.

The expected value of $m+n$ for a cost function with background part included corresponds to the case described in paragraph 20 of Michalak et al (2005).: *"the residuals are expected to follow the statistical distributions specified in the covariance matrices R and Q."*

**Our reasoning is given here. In a simple case where all weights are 1 and the prior errors are uncorrelated, the posterior cost function of size $n+m$ can be understood as follows: The average value of the $i^{th}$ posterior observation residual squared, $(H(x_{m,post},p)_i - s_i y_i)^2$, should be close to $\sigma^2_{O,I}$, and the average value of the $i^{th}$ posterior data residual squared, $(x_{post,i} - x_{A,i})^2$, should be close to the $i^{th}$ diagonal element of the a-priori error covariance matrix when the optimisation converges well and errors and prior parameters are properly specified. We have $m$ observation residuals, and $n$ data residuals (if background part included). In this example with a diagonal $S_A$ matrix, the residuals are assumed to be independent of each other. Each squared residual contributes on average a value of approximately 1 to the cost function, summing to approximately $m+n$, and thus $\chi^2_r \approx 1$. If e.g. we have 15 uncorrelated parameters and all posterior parameters would deviate a lot more then $\sigma_A$ from the prior, the prior parameters and/or errors are very likely not properly specified. This can be understood from the following: The prior distribution specifies that the *true* value of a parameter $x_i$ (which is approximated by the posterior value) should in approx. 68% of the cases be located at $x_{A,i} +/- \sigma_{A,i}$ (normal distribution, although truncated normal distributions might deviate from this), if e.g. all 15 parameters are outside this range, there is a very unlikely situation.

We have added more explanation to the text in the paper.

**References**

Brasseur, G. and Jacob, D.: Inverse Modeling for Atmospheric Chemistry, in: Modeling of Atmospheric Chemistry, pp. 487–537, Cambridge
University Press, Cambridge, https://doi.org/10.1017/9781316544754.012, 2017.

Doicu, A., Trautmann, T., and Schreier, F.: Numerical Regularization for Atmospheric Inverse Problems, Springer Praxis Books in environmentral sciences, https://doi.org/10.1007/978-3-642-05439-6, 2010.

Michalak, A. M., Hirsch, A., Bruhwiler, L., Gurney, K. R., Peters, W., and Tans, P. P. (2005), Maximum likelihood estimation of covariance parameters for Bayesian atmospheric trace gas surface flux inversions, J. Geophys. Res., 110, D24107, doi:10.1029/2005JD005970.

Raoult, N. M., Jupp, T. E., Cox, P. M., and Luke, C. M.: Land-surface parameter optimisation using data assimilation techniques: The adJULES system V1.0, Geoscientific Model Development, 9, 2833–2852, https://doi.org/10.5194/gmd-9-2833-2016, 2016.

Rodgers C. D. (2000) Inverse Methods for Atmospheric Sounding, World Sci.,
Tokyo.

Stull, R. B.: An introduction to boundary layer meteorology, Kluwer Academic Publishers, Dordrecht, 1988.

van Heerwaarden, C. C., Vilà-Guerau de Arellano, J., Moene, A. F., and Holtslag, A. A. M.: Interactions between dry-air entrainment, surface evaporation and convective boundary-layer development, Quarterly Journal of the Royal Meteorological Society, 135, 1277–1291, https://doi.org/10.1002/qj.431, 2009.

Vilà-Guerau De Arellano, J., Van Heerwaarden, C. C., Van Stratum, B. J., and Van Den Dries, K.: Atmospheric boundary layer: Integrating air chemistry and land interactions, Cambridge University Press, https://doi.org/10.1017/CBO9781316117422, 2015.

Wouters, H., Petrova, I. Y., van Heerwaarden, C. C., Vilà-Guerau de Arellano, J., Teuling, A. J., Meulenberg, V., Santanello, J. A., and Miralles, D. G.: Atmospheric boundary layer dynamics from balloon soundings worldwide: CLASS4GL v1.0, Geoscientific Model Development,
12, 2139–2153, https://doi.org/10.5194/gmd-12-2139-2019, 2019.

---

## Author Response (AR2)

Dear Reviewer,

Thanks a lot for your efforts to further improve the manuscript. Please find a point-to-point reply to your comments below.

Peter Bosman and Maarten Krol

p 2, line 35: The justification is too vague and not quantitative enough. It would be more relevant to specify the time scale and the horizontal resolution for which the assumption is valid based on specific case studies.
We have slightly adapted the text to make it more clear:
"The best model performance is during the convective daytime period. Since the CBL-model physics are relatively simple and only include the essential boundary layer processes, the model performs best on what might be called "golden days". Those are days in which advection is either absent or uniform in time and space, deep convection and precipitation are absent, and sufficient incoming shortwave radiation heats the surface allowing for the formation of a prototypical convective boundary layer. When these assumptions are met, the evolution of the budgets of heat, moisture, and gases is to a large extent determined by local land-atmosphere interactions. The aforementioned assumptions should ideally be valid for the whole modelled period. They should ideally hold on a spatial scale large enough that violations of the assumptions in the region do not influence the model simulation location. In practice, days are often not "ideal", e.g. a time-varying advection can be present. This does not necessarily mean the model cannot be applied to that day, but, performance is likely to be worse."

The assumptions underlying mixed-layer theory are rarely (or never) fully met. Placing more specific numbers on the scales over which the assumptions should hold is therefore very hard. Mixed-layer theory is however a powerful theory to understand the essential boundary layer behaviour, even if these assumptions are not fully met. For studies using mixed-layer theory, see e.g. Ouwersloot et al. (2012), Vilà-Guerau De Arellano et al (2012), van Heerwaarden et al. (2010), Pietersen et al. (2015), Pino et al. (2006).
p 3, line 74: Add ..with processes not included..."
It is not fully clear to us where this should be added. In the sentence before we write about ORCHIDEE and JSBACH: "These models have more complex physics not included in the CLASS model, which can be advantageous in accurately simulating land-atmosphere exchange". Therefore we assume the message of CLASS having more simple physics to be clear to the reader.
p 3, line 73: "This enables to include the information" Write instead "This enables the inclusion of information..".
Adapted, we now wrote "This facilitates the inclusion of atmospheric observations…"
p 3, line 74: You can specify that others groups have already coupled a land surface model to a transport model to assimilate both atmospheric observations (e.g., CO2 mixing ratio) and terrestrial observations. See the MPI-CCDAS with Schurmann et al. (2016), the ORCHIDEE-CCDAS with Peylin et al. (2016) and the BETHY-CCDAS (Rayner et al. (2005); Schloze et al. (2007); Ziehn et al. (2012) ; Kaminski (2013).
You could mention the technics used in these systems.

We added some information to the text: "Kaminski et al. (2012) and Schürmann et al. (2016) also assimilate both land-surface-related and atmosphere-related observations. In those studies a land-surface model is coupled to an atmospheric transport model. Meteorology is not simulated in those studies. In ICLASS, meteorology adds an additional set of observation streams, that can be used to optimise land-surface-related parameters that are linked both to gas fluxes and meteorology."

p 3, line 84: The BETHY Land Surface Model disposes also of an adjoint that is used to optimize the land surface parameters. See Ziehn et al. (2012).

Thanks for the suggestion, We adapted the text and incorporated the reference to Ziehn et al.: "An adjoint has been used in the past to optimise parameters, e.g. for land-surface models (Raoult et al., 2016; Ziehn et al., 2012)."

p28, line 705: When mentioning the CPU time consumed by an experiment with ICLASS, it would be relevant to compare with a Land Surface Model (e.g., SIB4) coupled to a transport model. How much faster is ICLASS compared to a full LSM coupled to a transport model?

We agree that it could be interesting to compare ICLASS to e.g. SIB4 coupled with a chemical transport model (CTM). However, we do not have a coupled SIB4-CTM run available for e.g. the Cabauw case. The result will also strongly depend on the configuration of the coupled model, e.g. which parameters are optimised, resolution etc.

**References**

Ouwersloot, H. G., Vilà-Guerau De Arellano, J., Nàlscher, A. C., Krol,M. C., Ganzeveld, L. N., Breitenberger, C.,Mammarella, I.,Williams,J., and Lelieveld, J.: Characterization of a boreal convective boundary layer and its impact on atmospheric chemistry during HUMPPACOPEC-2010, Atmospheric Chemistry and Physics, 12, 9335–9353, https://doi.org/10.5194/acp-12-9335-2012, 2012.

Pietersen, H. P., Vilà-Guerau de Arellano, J., Augustin, P., van de Boer, A., de Coster, O., Delbarre, H., Durand, P., Fourmentin, M., Gioli, B., Hartogensis, O., Lohou, F., Lothon, M., Ouwersloot, H. G., Pino, D., and Reuder, J.: Study of a prototypical convective boundary layer observed during BLLAST: contributions by large-scale forcings, Atmos. Chem. Phys., 15, 4241–4257, https://doi.org/10.5194/acp-15-4241-2015, 2015.

Pino, D., de Arellano, J. V. G., & Kim, S. W. (2006). Representing sheared convective boundary layer by zeroth-and first-order-jump mixed-layer models: Large-eddy simulation verification. *Journal of applied Meteorology and Climatology*, *45*(9), 1224-1243.

van Heerwaarden, C. C., Vilà-Guerau de Arellano, J., Gounou, A., Guichard, F., and Couvreux, F.: Understanding the daily cycle of evapotranspiration: A method to quantify the influence of forcings and feedbacks, Journal of Hydrometeorology, 11, 1405–1422, https://doi.org/10.1175/2010JHM1272.1, 2010.

Vilà-Guerau De Arellano, J., Van Heerwaarden, C. C., and Lelieveld, J.: Modelled suppression of boundary-layer clouds by plants in a CO2-rich atmosphere, Nature Geoscience, 5, 701–704, https://doi.org/10.1038/ngeo1554, 2012.

---

## Author Response (AR3)

Dear editor,

Thank you for handling our manuscript. Hereby your comment and our response to it:

*For the comments of a reviewer and the authors' response:*
*p 3, line 74: Add ..with processes not included..." It is not fully clear to us where this should be added.*
*In the sentence before we write about ORCHIDEE and JSBACH: "These models have more complex physics not included in the CLASS model, which can be advantageous in accurately simulating land-atmosphere exchange". Therefore we assume the message of CLASS having more simple physics to be clear to the reader.*
*Could you mention what kinds of physics and processes are not included in the model?*

We have adapted the text:

"These models simulate additional processes not included in the CLASS model, which enables to calculate additional land-surface variables. For example, in contrast to ORCHIDEE, CLASS cannot simulate leaf phenology or the allocation of carbon to different biomass pools."